# Detecting local genetic correlations with scan statistics

Hanmin Guo [1,2], James J. Li[3,4], Qiongshi Lu [5,7✉] & Lin Hou [1,2,6,7✉]

Genetic correlation analysis has quickly gained popularity in the past few years and provided insights into the genetic etiology of numerous complex diseases. However, existing approaches oversimplify the shared genetic architecture between different phenotypes and cannot effectively identify precise genetic regions contributing to the genetic correlation. In this work, we introduce LOGODetect, a powerful and efficient statistical method to identify small genome segments harboring local genetic correlation signals. LOGODetect automatically identifies genetic regions showing consistent associations with multiple phenotypes through a scan statistic approach. It uses summary association statistics from genome-wide association studies (GWAS) as input and is robust to sample overlap between studies. Applied to seven phenotypically distinct but genetically correlated neuropsychiatric traits, we identify 227 non-overlapping genome regions associated with multiple traits, including multiple hub regions showing concordant effects on five or more traits. Our method addresses critical limitations in existing analytic strategies and may have wide applications in post-GWAS analysis.

[1] Center for Statistical Science, Tsinghua University, Beijing, China. [2] Department of Industrial Engineering, Tsinghua University, Beijing, China. [3] Department of Psychology, University of Wisconsin-Madison, Madison, WI, USA. [4] Waisman Center, University of Wisconsin-Madison, Madison, WI, USA. [5] Department of Biostatistics and Medical Informatics, University of Wisconsin-Madison, Madison, WI, USA. [6] MOE Key Laboratory of Bioinformatics, School of Life Sciences, Tsinghua University, Beijing, China. [7] These authors jointly supervised: Qiongshi Lu, Lin Hou. ✉email: qlu@biostat.wisc.edu; houl@tsinghua.edu.cn

Genome-wide association studies (GWASs) have been carried out for numerous cosmplex traits and diseases, identifying tens of thousands of single-nucleotide polymorphisms (SNPs) associated with these phenotypes. However, our understanding of most traits' genetic basis remains incomplete, in part due to the limited power and interpretability of the traditional GWAS approach that correlates one trait with one SNP at a time. Recently, statistical methods that jointly model multiple phenotypes have quickly gained popularity in human genetics research[1–3]. Leveraging pervasive pleiotropy in the human genome, these methods enhanced the statistical power to identify genetic associations[1,4–7], improved the accuracy of genetic risk prediction[8,9], revealed novel genetic sharing across diverse phenotypes[10–12], and provided great insights into the genetic basis of a variety of diseases and traits[13,14].

Genetic similarity between traits can be modeled at different scales. Methods that identify SNPs associated with multiple phenotypes have achieved some success[15–17]. However, most complex human traits and their genetic overlaps are highly polygenic, with top SNPs showing weak to moderate effects[18–20]. Thus, single SNP-based methods modeling pleiotropy effects may not be sufficient to characterize the full landscape of genetic similarity of complex traits. An alternative approach is to estimate the genetic correlation between different traits[10,12,21,22]. These methods effectively utilize genome-wide genetic data, including SNPs that do not reach statistical significance in GWAS, to quantify the overall genetic sharing between two traits. In addition, recent methodological advances have enabled estimation of genetic correlation with GWAS summary statistics[10,11,23], making these approaches widely applicable to a large number of complex phenotypes. With these advances, genetic correlation analysis has become a routine procedure in post-GWAS analysis and was implemented in almost all large-scale GWASs published in the past few years.

However, despite improved statistical power and wide applications, genetic correlation approaches fail to provide detailed, mechanistic insights due to its oversimplification of complex genetic sharing into a single metric. Two recent methods improved genetic correlation analysis by providing local[12] and annotation-stratified estimates[11]. However, these methods rely on strong prior evidence about which local region or functional annotation to investigate. When applied to hypothesis-free scans, statistical power is reduced. In this work, we introduce LOGO-Detect (LOcal Genetic cOrrelation Detector), a method that uses scan statistics to identify genome segments harboring local genetic correlation between two complex traits. Compared to other methods, LOGODetect does not pre-specify candidate regions of interest, and instead, automatically detects regions with shared genetic components with great resolution and statistical power. In addition, LOGODetect only uses GWAS summary statistics as input and is robust to sample overlap between GWASs. We demonstrate its performance through extensive simulations and analysis of well-powered GWASs for seven distinct but genetically correlated neuropsychiatric traits[24,25]. Our analysis implicates a collection of hub regions (small genome segments harboring local genetic correlations for multiple trait pairs) in the genome that underlie the risk for several of these traits.

## Results

**Method overview**. Our goal is to identify genome segments showing consistent association patterns with two different traits. Here, we provide an overview of our approach and the technical details are discussed in the "Methods" section. We propose the following scan statistic:

$$Q(R) = \frac{\sum_{i \in R} z_{1i} z_{2i}}{\left(\sum_{i \in R} l_i\right)^{\theta}} \qquad (1)$$

to quantify the extent of local genetic similarity in a genome region, where $R$ is the index set for all SNPs in the region, $z_{1i}$ and $z_{2i}$ are the association z-scores for the $i$th SNP with two traits, $l_i$ is the linkage disequilibrium (LD) score for the $i$th SNP[10], and $\theta$ controls the impact of LD. $Q(R)$ extends the scan statistic proposed for single trait analysis[26,27] to the framework of detecting local genetic correlation. The scan statistic $Q(R)$ is a LD score-weighted inner product of local z-scores from two GWASs and is conceptually similar to local genetic correlation—regions with high absolute values of $Q(R)$ show concordant association patterns across multiple SNPs in the region and the sign of $Q(R)$ shows if the correlation is positive or negative. Of note, when the candidate region is the whole genome and $\theta$ is equal to 1, the scan statistic is an estimator for the global genetic covariance[11]. In our framework, we do not assume that per-SNP genetic covariance is the same for all SNPs across genome, but assume that genetic covariance is localized in some small genome regions. Therefore, we use the scan statistic in a local region, as a metric to detect significant local genetic sharing. A full discussion of the functional form of the scan statistic $Q(R)$ is provided in Supplementary Notes. We search for genome segments with the highest $|Q(R)|$ values by scanning the genome while allowing the segment size to vary (Fig. 1). Since we assume that the global genetic covariance can be solely attributed to some small regions, thus, the identified segments should collectively recapitulate a large proportion of genetic covariance of two traits. Therefore we select the optimal tuning parameter $\theta$ by maximizing the aggregated genetic covariance of all the identified regions. Statistical evidence of genetic sharing is assessed using a Monte Carlo approach.

**Simulation results**. We conducted simulations to compare the performance of LOGODetect with three existing methods: ρ-HESS[12], coloc[28], and gwas-pw[17]. ρ-HESS estimates local genetic correlation in pre-specified genomic regions based on a fixed-effect model, and coloc and gwas-pw are Bayesian approaches that estimate the posterior probability of colocalization for two traits. Note that the definition of genetic covariance (correlation) in our study is consistent with the traditional definition of covariance (correlation) of additive genetic effects under fixed-effect model[10].

We used HAPGEN2[29] to simulate genotypes for 100,000 samples based on 503 individuals with European ancestry from the 1000 Genomes Project Phase 3 data[30], and assessed the type I error of the four approaches under a variety of settings (see the "Methods" section; Supplementary Notes). First, we simulated phenotypes under an infinitesimal model in which genetic effects were assumed to be the same for all SNPs. We evaluated our method across a range of heritability combinations for two traits. We then compared different methods in two additional model settings representing diverse genetic architecture: a heritability-enrichment model where 3% of randomly selected SNPs explain 30% of trait heritability and the LDAK model[31] with MAF-dependent and LD-dependent architecture. In addition, we investigated if overlapping samples between two studies, mis-specified models with non-normal effects, and binary phenotypes would bias the inference. The family-wise type I error rate of our method was well-calibrated in all simulation settings with varying heritability values, extent of sample overlap, and genetic architecture (Supplementary Tables 1–8), showcasing the statistical robustness of LOGODetect. Type I errors for ρ-HESS were too conservative when heritability varies from 0.01 to 0.05 but

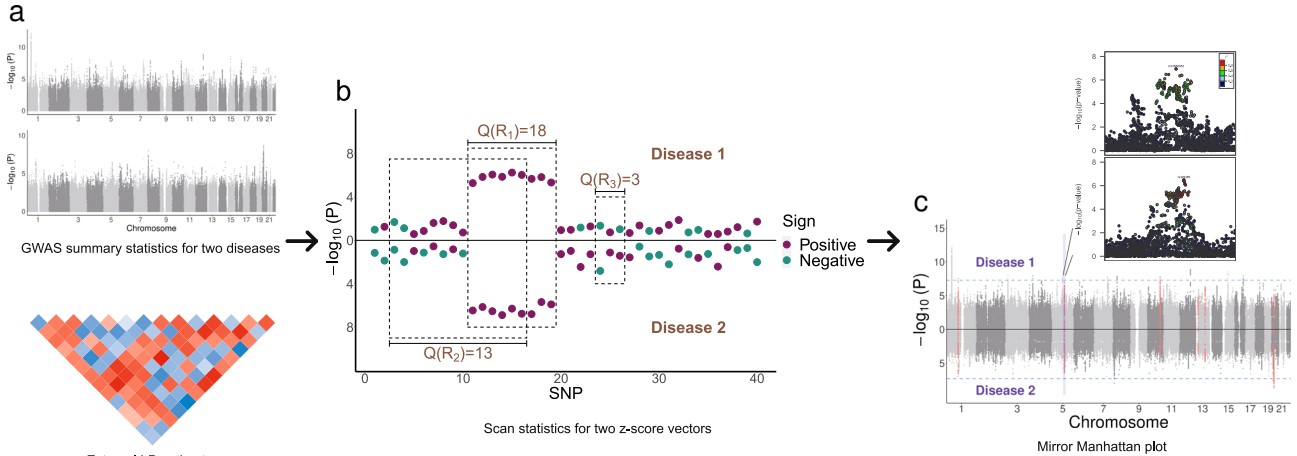

**Fig. 1 LOGODetect workflow. a** The inputs of LOGODetect include GWAS summary statistics for two traits and a reference panel for LD estimation. **b** Scan statistic is defined over a region, as the LD-weighted inner product of two z-score vectors in this region. A large absolute value of the scan statistic would hint at local genetic correlation. **c** LOGODetect identifies genome segments showing consistent associations with two different traits.

showed substantial inflation when heritability was large (e.g. 0.2) (Supplementary Table 9).

We also assessed the statistical power of LOGODetect under various settings. Three different metrics (i.e. point detection rate, segment detection rate, and $G$-score) were used to quantify the statistical power (see the "Methods" section). Signal points detection rate and signal segments detection rate measure the sensitivity at the SNP level and segment level, respectively. However, they do not reflect specificity of the method, as both metrics will be 1 if the identified region is the entire genome. $G$-score is a more informative alternative, which can jointly quantify specificity and sensitivity. First, we evaluated the power of LOGODetect with different $\theta$ under a heritability enrichment model, where a higher level of heritability was attributed to correlated regions (Supplementary Fig. 1). LOGODetect with adaptive $\theta$ achieved universally higher statistical power in three measures compared to the fixed $\theta$ approach, which demonstrated that maximizing aggregated genetic covariance of the identified regions could lead to a reasonable estimate of $\theta$.

Further, we compared different methods under a heritability enrichment model. As heritability increases, LOGODetect showed improvements in all three measures of statistical power without inflating the type I error (Fig. 2a–c). LOGODetect achieved greater signal points detection rates compared to ρ-HESS when heritability is low to moderate, compared to gwas-pw when heritability is moderate to high, and compared to coloc in all heritability settings (Fig. 2a). Moreover, LOGODetect showed almost universally higher signal segments detection rates and $G$-scores compared to the other three methods (Fig. 2b, c). LOGODetect achieved only slightly lower signal segments detection rates than ρ-HESS in one exceptional case when heritability is 0.05. We obtained consistent results under the heritability enrichment model with varing proportion of heritability (Fig. 2d–f). The gain of $G$-score can be attributed to the fact that LOGODetect flexibly and precisely identifies true signal regions, while ρ-HESS, coloc, and gwas-pw pre-specify candidate regions, which in general are much larger than the true signal regions, regardless of the disease phenotype. We also investigated if sample overlaps or binary phenotypes would affect the performance of our method. In addition, we compared statistical power of different approaches under mis-specified models, including LDAK model[31] with MAF-dependent and LD-dependent effect sizes, non-infinitesimal models with sparse effects, and infinitesimal models with heavy-tailed effect distributions. Details of simulation settings are shown in the Supplementary

Notes. We obtained consistent results under all simulation settings (Supplementary Figs. 2–10). Finally, the presence of correlated signal regions in the genome would not inflate type I error in non-signal regions (Supplementary Tables 10 and 11).

**Application to seven neuropsychiatric traits**. Previous studies have revealed pervasive pleiotropy[32–34] and genetic covariance[35–38] among neuropsychiatric traits. However, there is limited understanding of the specific genetic loci contributing to multiple traits. We applied LOGODetect to study the pairwise local genetic correlation between seven neuropsychiatric traits (Supplementary Table 12): bipolar disorder (BIP; $n = 51,710$), schizophrenia (SCZ; $n = 105,318$), major depressive disorder (MDD; $n = 173,005$), neuroticism (NEU; $n = 390,278$), attention-deficit/hyperactivity disorder (ADHD; $n = 53,293$), autism spectrum disorder (ASD; $n = 46,350$), and intelligence (IQ; $n = 269,867$), using summary statistics from the latest GWASs[39–45]. We adaptively selected the best $\theta$ by maximizing the genetic covariance in all identified regions (Supplementary Table 13). In total, we identified 410 regions (merged into 227 non-overlapping segments) showing concordant associations with multiple traits (FDR < 0.05; Fig. 3a and Supplementary Figs. 11–28). 274 of the 410 regions showed positive correlations (Supplementary Data 1). Size of the identified genome segments varied from 4 KB to 1.6 MB (Supplementary Fig. 29). The number of significant segments identified in our analysis is proportional to the absolute value of genetic correlation between each trait pair (Supplementary Fig. 30; correlation $r = 0.23$). We identified 56 shared genomic regions for BIP and SCZ (Fig. 3b; genetic correlation $r_g = 0.68$, $p = 9.14e{-}87$), 53 regions for SCZ and IQ (Supplementary Fig. 18; genetic correlation $r_g = -0.23$, $p = 4.36e{-}28$), 40 regions for MDD and NEU (Supplementary Fig. 26; $r_g = 0.78$, $p = 6.38e{-}41$), and 261 regions for 16 other trait pairs, which is consistent with the strong genetic overlap between these traits[46–49]. Overall, we identified strong genetic sharing (higher genetic correlation and more shared genome segments) among BIP, SCZ, MDD, and NEU and among MDD, ADHD, ASD, and IQ. Sharing between these two clusters was relatively weaker, which is consistent with previous reports[50].

**LOGODetect identifies precise regions with genetic sharing**. To benchmark the performance of LOGODetect with existing approaches, we also applied ρ-HESS, coloc, and gwas-pw to the

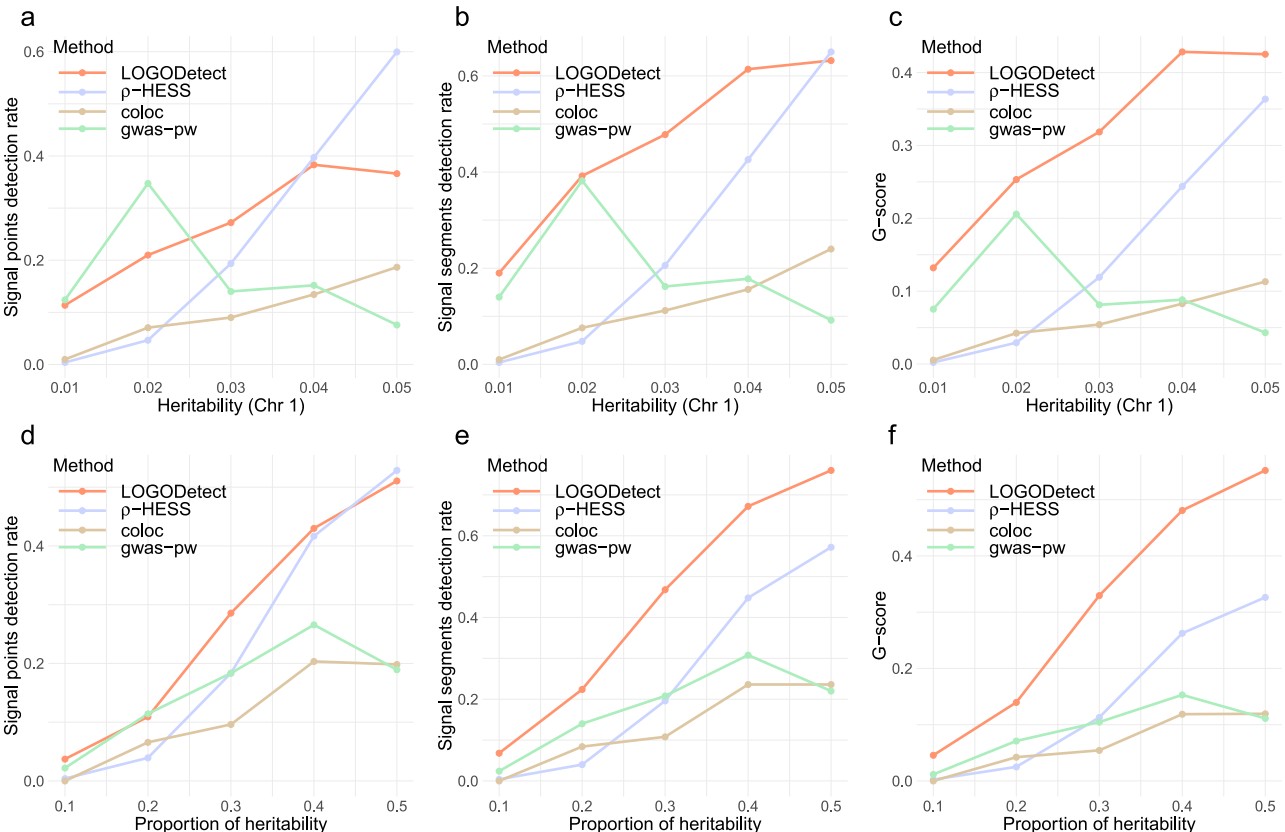

**Fig. 2 Assessment of statistical power under a heritability-enrichment model with varying trait heritability and heritability enrichment.** The *Y*-axis shows the statistical power assessed by three different metrics: **a**, **d** signal points detection rate measures sensitivity at the SNP level, **b**, **e** signal segments detection rate measures sensitivity at the segment level, and **c**, **f** G-score jointly measures specificity and sensitivity. The heritability represents the trait heritability on chromosome 1 and the proportion of heritability represents the proportion of the trait heritability explained by the signal regions. Significance cutoffs for gwas-pw are adjusted so that the empirical type I error rate is controlled at 0.05. Details on the above three metrics and the adjustment procedure for the significance cutoff are discussed in the "Methods" section. Source data are provided as a Source Data file.

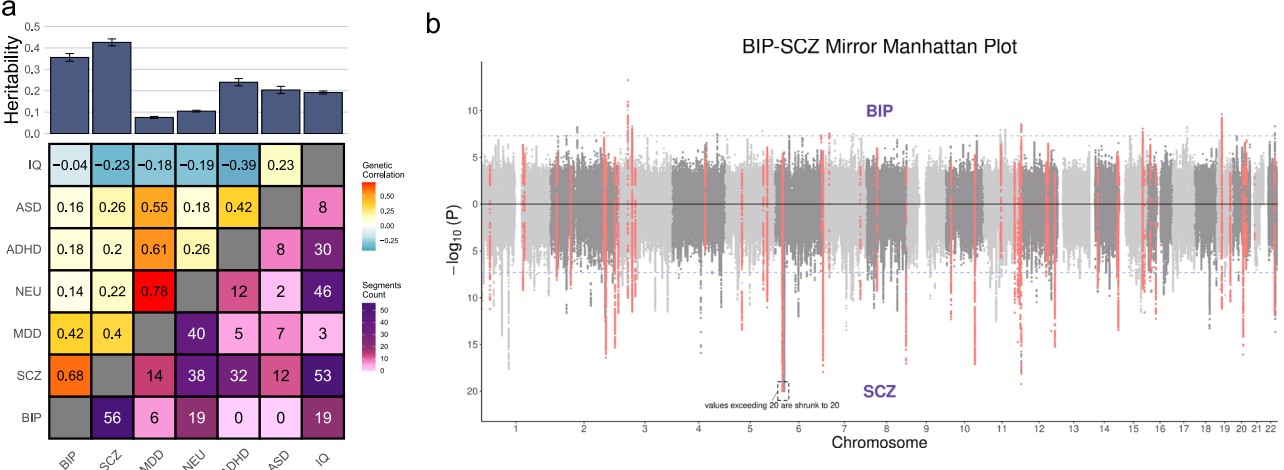

**Fig. 3 LOGODetect identifies genome regions contributing to multiple neuropsychiatric traits. a** Heatmap shows the genetic correlation estimates (upper triangle) and the number of segments with local genetic correlation identified by LOGODetect (lower triangle) between the seven neuropsychiatric traits; Barplot shows the observed scale heritability estimates and standard errors of the seven traits using LDSC[10]. **b** Mirrored Manhattan plot for BIP and SCZ. The 56 shared genome regions identified by LOGODetect are highlighted in red. One locus on chromosome 6 with −log₁₀ *P*>20 in SCZ is truncated at 20 for visualization purpose.

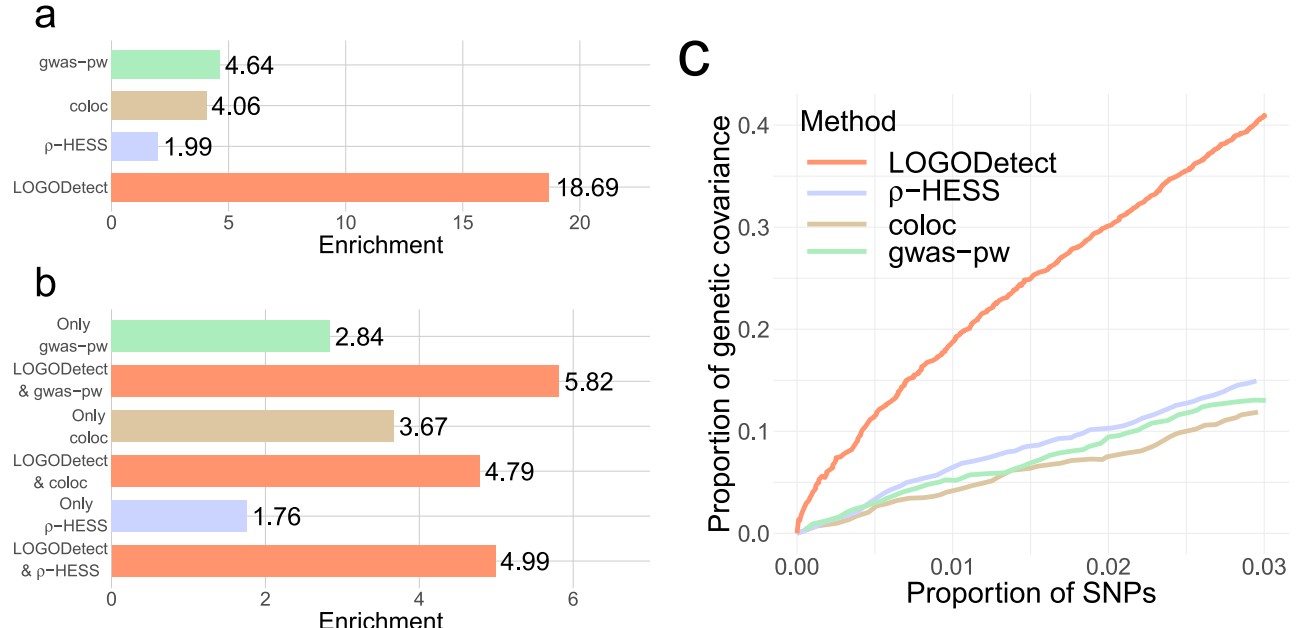

**Fig. 4 LOGODetect identifies precise genomic regions harboring local genetic correlations.** Genetic covariance and its corresponding enrichment were calculated using stratified-LDSC[10]. **a** Genetic covariance fold enrichment (i.e. the ratio between the proportion of total genetic covariance and the proportion of the total SNP counts) in regions identified by LOGODetect, ρ-HESS, coloc, and gwas-pw, respectively. **b** Genetic covariance fold enrichment in regions identified by ρ-HESS, coloc, and gwas-pw that also overlapped with LOGODetect findings, and regions identified by ρ-HESS, coloc, and gwas-pw alone. **c** Genetic covariance explained and proportion of SNPs covered by regions identified by LOGODetect, ρ-HESS, coloc, and gwas-pw.

same seven neuropsychiatric traits. We first assumed full sample overlaps as suggested in the original paper that introduced ρ-HESS. In total, ρ-HESS detected 778 regions for BIP and SCZ, and 304 regions for SCZ and IQ (FDR < 0.05; Supplementary Table 14). It only detected three regions for MDD and NEU, and failed to detect any significant local genetic correlation for any disorder pairs of MDD, ADHD, and ASD. Additionally, we also estimated the shared sample sizes based on the reported size of cohorts included in multiple studies (Supplementary Table 15), and used these approximated values as inputs for ρ-HESS to correct for sample overlap bias. The results remained consistent (Supplementary Table 14). The colocalization methods also detected strong genetic sharing between BIP and SCZ, between SCZ and NEU, and between SCZ and IQ (Posterior probability > 0.95; Supplementary Table 14).

We used the analysis of BIP and SCZ as an example to further illustrate the performance of LOGODetect. We used genetic covariance enrichment to quantify the precision of identified signal regions (Supplementary Notes). First, regions identified by LOGODetect showed the highest enrichment of genetic covariance compared to other methods (Fig. 4a). Although ρ-HESS identified more shared regions between BIP and SCZ, the enrichment of genetic covariance was 9.4-fold higher in the regions identified by LOGODetect, which is concordant with the simulation results based on *G*-scores. Second, we broke down the regions identified by ρ-HESS, coloc, and gwas-pw into two subsets: regions that overlap and do not overlap with regions identified by LOGODetect. The regions overlapping with LOGODetect results showed a higher enrichment for genetic covariance while enrichment in the regions identified by other methods alone were substantially lower (Fig. 4b). Third, to avoid comparison at an arbitrary significance cutoff, we ranked the regions identified by LOGODetect, ρ-HESS, coloc, and gwas-pw, by the corresponding *p*-values and posterior probability separately, and evaluated the proportion of explained genetic covariance at various thresholds. LOGODetect substantially outperformed other methods, explaining more genetic covariance with

the same proportion of SNPs (Fig. 4c; Supplementary Figs. 31–50). We also used estimated overlapping sample sizes to de-bias ρ-HESS estimates and results remained consistent (Supplementary Fig. 51).

There are two reasons why our method showed improved performance compared to the other methods. First, ρ-HESS and the colocalization methods pre-specify regions of interest, which are generally much larger than the signal regions harboring true genetic sharing (Supplementary Fig. 52), while our scanning approach is data-adaptive and can precisely identify the boundaries for signal regions. Second, both BIP (heritability $h^2$ = 0.35) and SCZ ($h^2$ = 0.43) have high SNP heritability. As demonstrated in the simulations, regions identified by ρ-HESS may include a non-negligible proportion of false positive findings.

Further, we evaluated the identified regions in an independent replication cohort. We tested whether the significantly correlated regions between BIP and SCZ can be replicated in the UK Biobank (UKBB). The summary statistics of BIP ($n_{case}$ = 1064, $n_{control}$ = 365,476) and SCZ ($n_{case}$ = 571, $n_{control}$ = 365,476) in the UKBB were collected (Supplementary Table 16). Due to the unbalanced case-control ratio and limited effective sample size, we used aggregated genetic covariance to evaluate the replication (Supplementary Notes). Stratified-LDSC was not applicable due to the imbalanced sample sizes of cases and controls, therefore we applied GNOVA[11] for stratified genetic covariance analysis of the regions identified by four methods in the UKBB data, respectively. The regions identified by LOGODetect and ρ-HESS both showed significant genetic covariance, but the regions identified by LOGODetect have a 6.7-fold higher genetic covariance enrichment than that of ρ-HESS, which demonstrates again that LOGODetect can more precisely detect the true signal regions (Table 1; Supplementary Fig. 53). Regions identified by gwas-pw showed no significant genetic covariance, while regions identified by coloc showed significant genetic covariance with the opposite sign.

We also replicated findings for body-mass index (BMI) and height, for which independent replication cohorts of large sample

**Table 1 Stratified genetic covariance analysis on UKBB replication cohorts.**

|  | Genetic Cov[a] | s.e. | *p*-value | Proportion of genetic cov (%) | Proportion of SNPs (%) | Fold enrichment |
|---|---|---|---|---|---|---|
| LOGODetect | 2.18e−4 | 6.65e−5 | 1.04e−3 | 11.50 | 1.15 | 10.02 |
| ρ-HESS | 6.62e−4 | 3.16e−4 | 3.61e−2 | 30.00 | 20.12 | 1.49 |
| coloc | −5.70e−5 | 2.30e−5 | 1.33e−2 | −2.85[b] | 0.34 | −8.36[b] |
| gwas-pw | 3.84e−5 | 6.54e−5 | 5.57e-1 | 1.92 | 1.61 | 1.20 |

[a]Genetic Cov represents estimated genetic covariance of the identified regions using GNOVA.
[b]Genetic covariance of regions identified by coloc has opposite sign compared to global genetic covariance, therefore the corresponding proportion of genetic covariance and fold enrichment are negative.

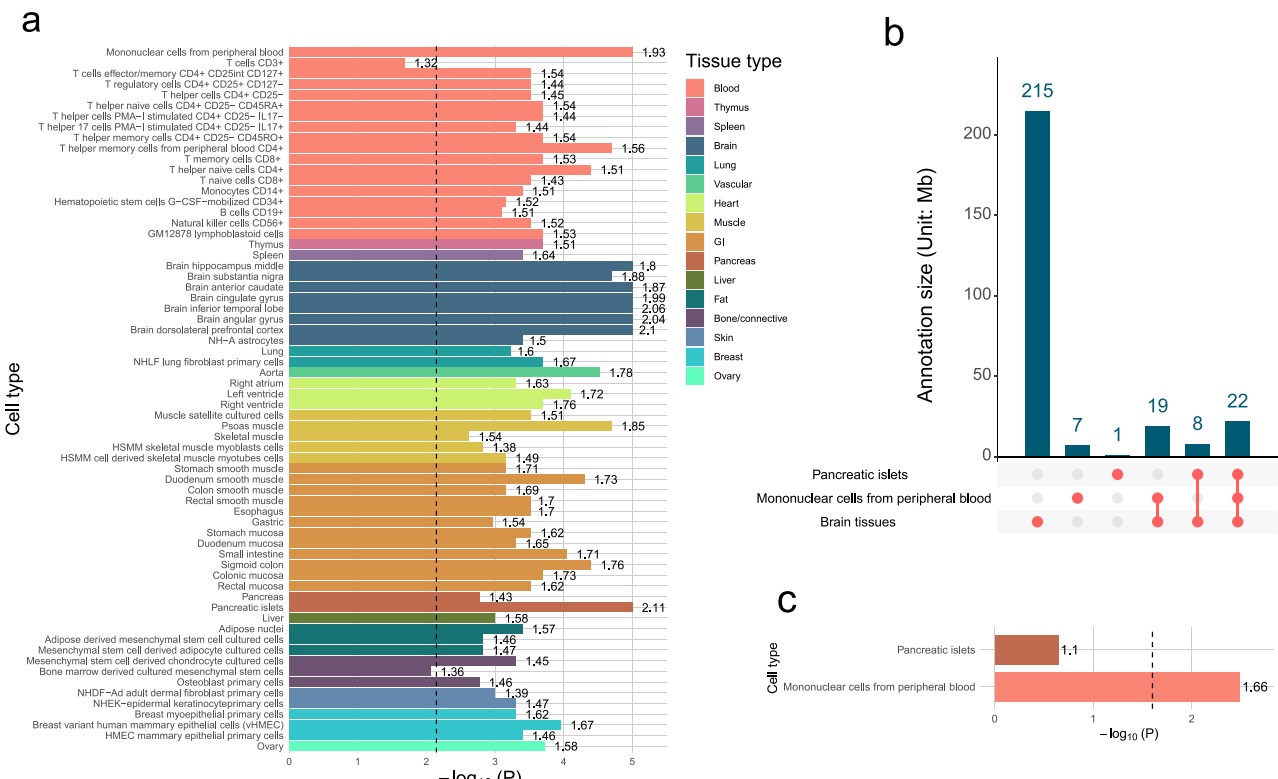

**Fig. 5 Tissue-specific enrichment of genome regions conferring risk for multiple neuropsychiatric traits. a** Permutation test results over 66 cell-type-specific annotations. Fold enrichment is labeled next to each bar. **b** The overlap of predicted functional regions in pancreatic islets, mononuclear cells from peripheral blood, and eight brain regions. **c** Enrichment in the predicted functional regions in pancreatic islets and mononuclear cells from peripheral blood after conditioning on the annotation overlap with brain regions.

size are available (Supplementary Notes). We identified 24 regions with significant local genetic correlation in the discovery analysis. 17 of 24 regions identified in the discovery stage were successfully replicated, suggesting the effectiveness of LOGODetect to identify replicable genomic regions with local genetic correlations (Supplementary Table 17).

**Tissue enrichment of hub regions shared by neuropsychiatric traits.** We used 66 GenoSkyline-Plus tissue-specific functional annotations[51] to investigate the functional relevance of the genomic regions found to harbor local genetic correlations among seven neuropsychiatric traits (Supplementary Table 18). We used permutation tests to assess the enrichment of genome regions shared by multiple traits in these annotation tracks. Genome regions identified by LOGODetect were significantly enriched in eight brain regions (minimum enrichment = 1.50, *p* = 4.00e−4)

(Fig. 5a). In addition to brain tissues, regions shared by neuropsychiatric traits were also strongly enriched in mononuclear cells from peripheral blood (enrichment = 1.93, *p* = 1.00e−5) and pancreatic islets (enrichment = 2.11, *p* = 1.00e−5). Of note, annotated functional regions in mononuclear cells and pancreatic islets have substantial overlaps with annotations of brain tissues (Fig. 5b). After conditioning on functional regions in the brain, the enrichment in pancreatic islets was substantially reduced (enrichment = 1.1, *p* = 0.224; Fig. 5c), while enrichment in mononuclear cells remained significant (enrichment = 1.66, *p* = 3.55e−3).

To further assess whether enrichments are truly tissue-specific, we performed conditional analysis on six generic annotations (i.e., coding regions, enhancers, introns, promoters, 5′UTRs, and 3′UTRs, extended by a 500-bp window around each annotation) in Finucane et al.[52]. After conditioning on these annotations, the enrichment in brain tissues remained significant (minimum

enrichment = 1.37, $p = 1.98e{-}3$), suggesting that the observed enrichment in functional genome in these brain tissues were not driven by generic annotations alone (Supplementary Fig. 54).

We also ran Gene Ontology-enrichment analysis using FUMA[53]. The 968 genes in regions detected by LOGODetect were significantly enriched for 83 GO terms (Supplementary Table 19) after multiple testing correction, including RNA metabolic process ($p = 5.36e{-}13$), nucleolus ($p = 9.30e{-}6$), and protein arginine deiminase activity ($p = 7.35e{-}9$).

**Hub regions contributing to multiple neuropsychiatric traits.** Next, we investigated hub regions shared by five or more traits. Among the 227 non-overlapping genome regions identified in our analysis, 91 regions were identified in two or more different trait pairs (Supplementary Data 2). The five regions identified in at least seven pair-wise analyses are summarized in Supplementary Table 20. Notably, LOGODetect consistently identified these hub regions in more trait pairs compared to other methods. These hub regions show consistent associations with multiple neuropsychiatric traits and can potentially reveal key mechanisms and pathways underlying the shared genetics across traits.

The region showing significant correlation in nine pair-wise analyses is a locus spanning 711 KB on chromosome 11 (Fig. 6). Interestingly, two independent peaks were identified in this region between SCZ and NEU and between MDD and NEU. SNPs in this region have previously reached genome-wide significance in the SCZ[40] (lead SNP rs2514218; $p = 2.42e{-}12$), NEU[45] (lead SNP rs35738585; $p = 2.47e{-}17$), and IQ GWAS[44] (lead SNP rs2885208; $p = 4.58e{-}8$). Additionally, SNPs at this locus showed consistent associations with BIP (lead SNP rs10502165; $p = 3.90e{-}5$). More importantly, this genome region covers the NTAD (*NCAM1-TTC12-ANKK1-DRD2*) gene cluster. Multiple variants of *DRD2* and *NCAM1* are reported to be associated with BIP, SCZ, MDD, and NEU[54–56]. Also, multiple eQTLs for *DRD2* (lead SNP rs6589381; $p = 1.10e{-}14$) and *NCAM1* (lead SNP rs1079021; $p = 9.20e{-}16$) are located in the region.

Another region on chromosome 11 spans 488 KB and shows significant correlations in seven pair-wise analyses (Supplementary Fig. 55). *IGSF9B*, a potential risk gene for SCZ[40] and IQ[44], and its multiple eQTLs (lead SNP rs558709; $p = 1.80e{-}13$) are located in this genomic region. The third hub region is located on chromosome 14 spanning 694 KB and shows significant correlations in seven trait pairs (Supplementary Fig. 56). Gene *PRKD1* largely overlaps with this locus, and *FOXG1*, which is an associated gene for SCZ[40] and IQ[44], lies about 200 KB away from the locus. In addition, multiple eQTLs for *PRKD1* (lead SNP rs80019464; $p = 6.40e{-}5$) and *FOXG1* (lead SNP rs138384350; $p = 6.10e{-}7$), are located in the region. The fourth region on chromosome 3 spans 258 KB and was identified in seven pairs (Supplementary Fig. 57). Notably, most parts of this genomic region are covered by the gene *FOXP1*, which is an implicated risk gene for SCZ[40] and IQ[44]. The final region spans 450 KB on chromosome 10. This region was identified in seven trait pairs (Supplementary Fig. 58) and largely overlaps with *SORCS3*, a previously implicated risk gene for MDD and ADHD[42,57,58].

## Discussion

Through simulations and analyses of GWAS data, we demonstrated that our method effectively identified genetic regions that show concordant associations across multiple complex traits with high resolution and statistical power. Compared to existing

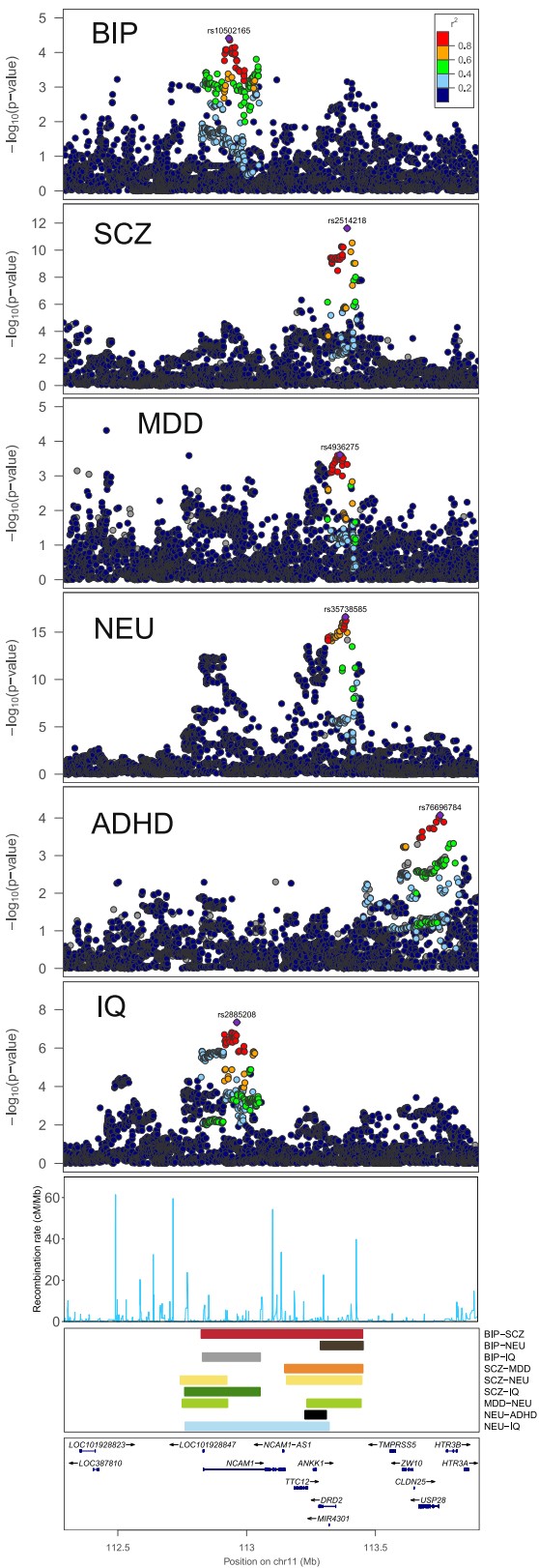

**Fig. 6 Putative target genes for the hub region in chr11 shared by nine neuropsychiatric trait pairs.** Locuszoom plot, recombination rate, and the gene names are provided. The colored band denote the location of the significant region and which trait pair is detected in.

approaches, LOGODetect has greater statistical power and is robust across various heritability settings and in existence of sample overlaps. Applied to well-powered GWASs for seven phenotypically distinct but genetically correlated neuropsychiatric traits, LOGODetect identified numerous shared genomic regions including hub regions that showed consistent effects for more than four traits. The regions identified by LOGODetect explain a larger portion of genetic covariance than existing approaches. Furthermore, the enrichment holds true in independent replication studies.

Two genes (i.e. *DRD2* and *NCAM1*) are located in the hub region on chromosome 5 (Fig. 6). *DRD2*, also known as dopamine receptor D2, encodes the D2 subtype of the dopamine receptor. The dopamine hypothesis of schizophrenia suggests that dopaminergic pathways are overactive in schizophrenia[59]. In addition, multiple variants are reported to be associated with psychiatric disorders[54]. *NCAM1*, short for neural cell adhesion molecule 1, plays an important role in formation of plexiform layers, neurite fasciculation, nerve–muscle interactions and other aspects of neural development[60]. Expression of *PSA-NCAM* is increased in antidepressant treatment, while in animal models of depression or in depressed patients *PSA-NCAM* is reduced[56]. Notably, *NCAM1* was identified by LOGODetect as implicated gene for MDD, but it cannot be identified by other three methods.

Other identified hub regions also included a handful of interesting candidate genes. *IGSF9B* (Supplementary Fig. 55) encodes a brain-specific cell adhesion molecule which is highly expressed in GABAergic interneurons, concentrated to hippocampal and cortical inhibitory synapses for their development into interneurons[61]. Interestingly, promotion of *IGSF9B* for inhibitory synapses development is coupled with *NLGN2*, loss of function variants of which were found in autism and schizophrenia patients[62,63]. *PRKD1* (Supplementary Fig. 56) encodes a serine/threonine protein kinase which is important in many cellular processes, and regulates neuronal polarity, synapse formation, and synaptic plasticity[64–66]. *FOXG1* (Supplementary Fig. 56) encodes the fork-head box protein G1 which is strongly expressed in neural tissues, operates as a transcriptional repressor essential in brain development[67]. It was suggested that *PRKD1* locus regulates *FOXG1* in a cis-acting way, and is associated with the FOXG1 syndrome including mental retardation, absent language, and dyskinesia[67]. *FOXP1* (Supplementary Fig. 57) is one of the FOXP transcription factor subfamily. It is expressed in cerebral cortex, striatum, and spinal cord of the central nervous system, and is shown to regulate striatum development, motor neuron migration, and midbrain dopamine neuron differentiation[68]. *FOXP1* is associated with ASD, speech delay, and intellectual disability[69,70]. *SORCS3* (Supplementary Fig. 58) is highly expressed in the CA1 region of the hippocampus, and is involved in synaptic depression and spatial learning ability[71,72]. It is also known to play an important role in protein networks associated with *PICK1*, *NGF*, and *PDGF-BB*[73,74], which have been implicated in ADHD, ASD, MDD, and SCZ[75–78].

Our method still has some limitations. First, the goal of LOGODetect is to identify genomic regions harboring local genetic correlations. We do not give explicit estimation of local genetic correlation, but the sign of the correlation can be inferred. Although local genetic correlation in identified regions can be estimated by other methods (e.g., ρ-HESS) in principle, this remains a statistically challenging problem. As shown in our simulations, the estimation is inaccurate. Under the null hypothesis that local genetic correlation is zero, ρ-HESS underestimates the standard error of local genetic covariance when heritability is high and leads to inflated type I error rates, and it overestimates the standard error of local genetic covariance when heritability is low and leads to deflated statistical power. We note

that the deflation of type I error observed for ρ-HESS is not contradictory to results published in ρ-HESS paper[12]. ρ-HESS was formulated as an estimation problem instead of the hypothesis testing problem in our manuscript. In their paper, they have shown simulation results to demonstrate that the local genetic correlation can be accurately estimated when the true parameter is 0. However, the evidence shown in the ρ-HESS paper could not rule out deflation when the method is used for inference. These problems are further exacerbated when ρ-HESS is applied to very small local genomic regions identified by LOGODetect. Second, LOGODetect scans a large number of genomic regions to search for local regions where the scan statistic significantly deviates from the null distribution. We currently do not have an analytical solution to derive or approximate the theoretical null distribution. Instead, a Monte Carlo approach is employed to quantify the null distribution of the maximal scan statistic, which is computationally expensive. Third, we proposed an empirical method to select the tuning parameter based on the aggregated genetic covariance of the identified regions. Although there is no theoretical guarantee, we have conducted extensive simulations to demonstrate that the empirical strategy to estimate θ works well with frequently used genetic models and leads to superior performance of LOGODetect compared to competitive methods, in terms of error control and statistical power. Fourth, our simulations are conducted with simulated genotypes based on the European ancestry individual data in the 1000 Genomes Project. It would be interesting to simulate data with various population structures to test our method. In real data applications, GWAS summary statistics are usually corrected for the genomic control factor, thus we expect population structure to have minor impact on the performance of LOGODetect. Fifth, several recent methods have been proposed to jointly model more than two GWAS traits to infer the structure of shared genetics across multiple phenotypes[14,47,79]. A future direction is to generalize our method to search for genomic regions shared by more than two traits. Finally, LOGODetect studies genetic correlation from GWAS data, which uses a bivariate random effect model and defines the genetic correlation as the correlation between SNPs[10,18,21,80]. Under this model, the definition of genetic correlation is consistent with the traditional definition of correlation of additive genetic effects[10]. Yet the concept should be distinguished from the additive genetic correlation, since the estimation could be biased to the partial effects of tag SNPs, and the causal effects of untagged SNPs would be absorbed to effect of random error term[81].

Taken together, we have introduced LOGODetect, a scan statistic method to identify local genetic regions showing correlated effects with multiple neuropsychiatric traits. Complementary to single SNP-based approaches for pleiotropy mapping[17,28] and genetic correlation estimation methods utilizing genome-wide data[10,21], our method elucidates the shared genetic architecture between two traits by identifying local genomic segments that are concordant. The candidate genes and regions we identified may be tapping into a set of transdiagnostic mechanisms that underlie all of psychopathology (i.e., the "*p*" or general factor[47]). In practice, LOGODetect can be used in combination with other methods to further improve statistical power and biological interpretability. For example, it may be of interest to first screen the genome by identifying larger genetic regions[12,82] or certain functional annotations[11] enriched for the shared genetics between two traits. Then, LOGODetect can be applied to these candidate regions to identify the precise genetic segments that explain such sharing. Since high-dimensional sampling remains a challenge, a multi-tier analytical strategy would improve the statistical power and computational burden in the analysis. We believe that LOGODetect has addressed some key limitations in the current

practice of cross-trait genetic correlation analysis and will benefit complex trait genetics research.

## Methods

**Genetic model.** Suppose two standardized traits $\mathbf{y}_1$ and $\mathbf{y}_1$ follow the linear model with random effects:

$$\mathbf{y}_1 = \mathbf{X}\beta + \boldsymbol{\epsilon},$$
$$\mathbf{y}_2 = \mathbf{Z}\gamma + \boldsymbol{\delta}, \tag{2}$$

where $\mathbf{X}$ and $\mathbf{Z}$ are fixed and standardized genotype matrices with $M$ columns (i.e. the number of SNPs is $M$); $\boldsymbol{\epsilon}$ and $\boldsymbol{\delta}$ are non-genetic effects; $\beta$ and $\gamma$ are $M$-dimensional vectors denoting genetic effects. They follow the multivariate normal distribution:

$$\begin{bmatrix} \boldsymbol{\beta} \\ \boldsymbol{\gamma} \end{bmatrix} \sim N\left( \begin{bmatrix} 0 \\ 0 \end{bmatrix}, \begin{bmatrix} \frac{h_\beta^2}{M}\mathbf{I}_M & \frac{\rho_g}{K}\tilde{\mathbf{I}}_M \\ \frac{\rho_g}{K}\tilde{\mathbf{I}}_M & \frac{h_\gamma^2}{M}\mathbf{I}_M \end{bmatrix} \right),$$

where $h_\beta^2$ and $h_\gamma^2$ denote the heritability for two traits; $\rho_g$ is the global genetic covariance between two traits; $\tilde{\mathbf{I}}_M$ is a diagonal matrix whose $i$th diagonal element equals 1 if the effects of the $i$th SNP on two traits (i.e. $\beta_i$ and $\gamma_i$) are correlated and equals 0 if otherwise; $K$ is the number of SNPs such that $\beta_i$ and $\gamma_i$ are correlated, i.e., $K = \mathrm{tr}[\tilde{\mathbf{I}}_M]$. $\beta$ and $\gamma$ are independent from non-genetic effects $\boldsymbol{\epsilon}$ and $\boldsymbol{\delta}$. The statistical model described here is similar to the polygenic model used in genetic correlation estimation[10]. The difference is that we allow local genetic sharing and do not assume the global genetic covariance to be equally attributed to all SNPs in the whole genome. Compared to the local genetic correlation estimation method in the literature[12], we do not assume genetic effects to be fixed. Instead, our framework is a direct generalization of the model developed for global genetic correlation estimation[10,11]. Under the alternative hypothesis, we denote the non-overlapping genetic regions that contribute to multiple traits to be $R_1, \ldots, R_r$ and the union set as $\mathcal{R} = \cup_{j=1}^{r} R_j$ such that $\tilde{\mathbf{I}}_M[i,i] = 1$ if and only if $i \in \mathcal{R}$. While under the null hypothesis, two traits share no genetic covariance, i.e., $\mathcal{R} = \emptyset$.

**Scan statistic and scanning procedure.** We use a scan statistics approach to identify regions showing correlated effects between different traits. This type of approach has been used for burden test in a single-trait setting[83]. Suppose $n_1, n_2$ are the sample sizes for two GWASs, respectively, and we first consider the simpler case that there is no sample overlap between two GWASs. Additionally, we denote the association $z$-scores for two traits as

$$\mathbf{z}_1 = [z_{11}, z_{12}, \ldots, z_{1M}]^T = \frac{1}{\sqrt{n_1}}\mathbf{X}^T\mathbf{y}_1,$$
$$\mathbf{z}_2 = [z_{21}, z_{22}, \ldots, z_{2M}]^T = \frac{1}{\sqrt{n_2}}\mathbf{Z}^T\mathbf{y}_2. \tag{3}$$

Then, we can define the scan statistic as:

$$Q(R) = \frac{\sum_{i \in R} z_{1i}z_{2i}}{\left(\sum_{i \in R} l_i\right)^\theta}, \tag{4}$$

where $R$ is the index set for SNPs in a genome region, $l_i$ is the LD score[80] for the $i$th SNP computed within a 1 MB window, and $\theta$ is a tuning parameter that controls the strength we penalize over the LD structure. If SNPs in the region $R$ show strong, concordant effects on both traits, then the inner product $\sum_{i \in R} z_{1i}z_{2i}$ will tend to have a larger absolute value and therefore yield a larger scan statistic. On the contrary, if two traits are genetically independent in the local region, then the corresponding scan statistic would be close to 0. Therefore, the scan statistic is informative to detect local genetic correlation. The purpose of the LD score term in the denominator is to normalize the effect of LD. The expected absolute value of $\sum_{i \in R} z_{1i}z_{2i}$ is larger in regions with strong LD (Supplementary Fig. 59; Supplementary Notes). Without the normalization term on the denominator, the method will favor regions with large LD that may not be of biological interest.

Finally, we use the maximal scan statistic over all possible regions as the test statistic:

$$Q_{max} = \max_{|R| \le C} |Q(R)|, \tag{5}$$

where $C$ is a pre-specified parameter that defines the upper boundary of the SNPs count in a region. In practice, $C$ can be set based on the number of SNPs in the dataset (e.g. the average number of SNPs in 1 million bases). LOGODetect takes advantages of the flexible framework to scan local regions with varying sizes. Compared to a sliding-window approach based on a pre-specified window size, our method is more appealing since the size of signal region could vary substantially by locus and by trait. We use a Monte Carlo type approach to assess the distribution of $Q_{max}$ under the null hypothesis. We draw 5000 pseudo-samples $\begin{bmatrix} \mathbf{z}_1 \\ \mathbf{z}_2 \end{bmatrix}$ under the null distribution using a procedure detailed in the next section. Then, we estimate the empirical null distribution of $Q_{max}$ and its 95% upper quantile, $Q_{0.95}$. Taken together, the scanning procedure works as follows. We scan the genome to find $R_1$ such that $|Q(R_1)|$ reaches the maximum. If $|Q(R_1)| \ge Q_{0.95}$, we claim that $R_1$ is a significant signal region and remove these SNPs from the analysis. Then, we repeat

the procedure on the remaining SNPs until no region is declared significant. This procedure controls the family-wise type I error rate. Calculating $Q(R)$ over all possible candidate regions is indeed computationally expensive, so we constrain $|R|$ to be a multiple of 10 in practice, which reduced the computation burden by ~10 folds, with minimal reduction in accuracy. Finally, regions that are no more than 100 KB away from each other are merged into a single region.

**Choice of parameter $\theta$.** Parameter $\theta$ affects the size of identified regions. A relatively long segment may not have a large absolute value of scan statistic, due to the penalty in the denominator $\left(\sum_{i \in R} l_i\right)^\theta$. A larger $\theta$ implies stronger penalty, henceforth is more likely to detect smaller signal segments. In particular when $\theta$ equals 1, $|Q(R)|$ will attain local maximum with $R$ containing only one variant. A reasonable range for $\theta$ is between 0 and 1. In practice, it is important to consider the "best" $\theta$ adaptive to the data. We used the proportion of genetic covariance of the identified regions as the metric. We varied the value of $\theta$ in the candidate set, and chose the best $\theta$ such that the corresponding identified regions have the largest genetic covariance. In general, one can use any subset of [0, 1] as the candidate set of $\theta$ values. However, extensively searching for $\theta$ substantially increases the computation time. In practice, we suggest the set of {0.4, 0.45, 0.5, 0.55, 0.6, 0.65, 0.7} would be sufficient. Denote the regions detected by LOGODetect under parameter $\theta$ as $\hat{R}_1^\theta, \ldots, \hat{R}_m^\theta$. We denote their union as $\hat{\mathcal{R}}^\theta$ and denote the genetic covariance in $\hat{\mathcal{R}}^\theta$ as $\rho(\hat{\mathcal{R}}^\theta)$. In theory, $\rho(\hat{\mathcal{R}}^\theta) = \sum_{i=1}^{m} |\hat{R}_i^\theta \cap \mathcal{R}|\frac{\rho_g}{K}$, where $\mathcal{R}$ is union set of true signal regions, $\rho_g$ is the global genetic covariance, and $K = |\mathcal{R}|$ is the number of SNPs in $\mathcal{R}$. In practice, the true signal regions $\mathcal{R}$ is unknown. $\rho(\hat{\mathcal{R}}^\theta)$ can be estimated using the stratified-LDSC[10]. Let $\pi(\theta) = \frac{\rho(\hat{\mathcal{R}}^\theta)}{\rho_g}$ be the proportion of genetic covariance explained by $\hat{\mathcal{R}}^\theta$ to the global genetic covariance. We assume that $\rho(\hat{\mathcal{R}}^\theta) = 0$ and $\pi(\theta) = 0$ if $\hat{\mathcal{R}}^\theta = \emptyset$. We calculate $\pi(\theta)$ for a candidate set of $\theta$ values, and then we determine $\theta$ adaptive to data via the following optimization problem:

$$\hat{\theta} = \arg\max_\theta |\pi(\theta)|. \tag{6}$$

**Monte Carlo simulation of pseudo-$z$-score vectors.** In order to simulate the null distribution of $Q_{max}$, we need to generate pseudo-$z$-score vectors. When two GWASs do not have sample overlap, it can be verified that

$$\mathrm{var}[\mathbf{z}_1] = \frac{1}{n_1}\left[\frac{h_\beta^2}{M}\mathbf{X}^T\mathbf{X}\mathbf{X}^T\mathbf{X} + (1 - h_\beta^2)\mathbf{X}^T\mathbf{X}\right], \tag{7}$$

$$\mathrm{cov}[\mathbf{z}_1, \mathbf{z}_2] = \frac{\rho_g}{\sqrt{n_1 n_2}K}\mathbf{X}^T\mathbf{X}\tilde{\mathbf{I}}_M\mathbf{Z}^T\mathbf{Z}. \tag{8}$$

And similarly for $\mathrm{var}[\mathbf{z}_2]$. Therefore, under $H_0$, the combined $z$-score vector

$$\begin{bmatrix} \mathbf{z}_1 \\ \mathbf{z}_2 \end{bmatrix} \sim N\left( \begin{bmatrix} 0 \\ 0 \end{bmatrix}, \begin{bmatrix} \frac{1}{n_1}\left[\frac{h_\beta^2}{M}\mathbf{X}^T\mathbf{X}\mathbf{X}^T\mathbf{X} + (1-h_\beta^2)\mathbf{X}^T\mathbf{X}\right] & 0 \\ 0 & \frac{1}{n_2}\left[\frac{h_\gamma^2}{M}\mathbf{Z}^T\mathbf{Z}\mathbf{Z}^T\mathbf{Z} + (1-h_\gamma^2)\mathbf{Z}^T\mathbf{Z}\right] \end{bmatrix} \right),$$

asymptotically. Note that in practice individual genotype data is hard to obtain due to privacy, it is meaningful to analyze based only on summary statistics. Here by using reference panel (e.g. the 1000 Genomes Project Phase 3 data[30]), $\frac{1}{n_1}\mathbf{X}^T\mathbf{X}$ and $\frac{1}{n_2}\mathbf{Z}^T\mathbf{Z}$ can be estimated as $\mathbf{V}$, $\frac{1}{n_1^2}\mathbf{X}^T\mathbf{X}\mathbf{X}^T\mathbf{X}$ and $\frac{1}{n_2^2}\mathbf{Z}^T\mathbf{Z}\mathbf{Z}^T\mathbf{Z}$ can be estimated as $\widetilde{\mathbf{V}^2} = \frac{n-1}{n-2}\mathbf{V}^2 - \frac{M-2}{n-2}\mathbf{V}$, where $n$ is the sample size of the reference panel and $\mathbf{V}$ is the LD matrix of the reference panel. And the genetic heritability for two traits $h_\beta^2, h_\gamma^2$ can be estimated through LDSC[80]. After plugging in the reference LD matrix, we have

$$\begin{bmatrix} \mathbf{z}_1 \\ \mathbf{z}_2 \end{bmatrix} \sim N\left( \begin{bmatrix} 0 \\ 0 \end{bmatrix}, \begin{bmatrix} \frac{n_1 h_\beta^2}{M}\widetilde{\mathbf{V}^2} + (1-h_\beta^2)\mathbf{V} & 0 \\ 0 & \frac{n_2 h_\gamma^2}{M}\widetilde{\mathbf{V}^2} + (1-h_\gamma^2)\mathbf{V} \end{bmatrix} \right),$$

asymptotically under the null.

The random multivariate normal vectors have complex covariance structure, which is computationally challenging as the dimension of the vector can be as high as $10^7$ in GWAS. We developed a computationally tractable method that leverages the LD structure in the genome. First, we split the high-dimensional vector $\mathbf{z}$ into sub-vectors $\mathbf{z} = \begin{bmatrix} \mathbf{z}_{(1)}, \mathbf{z}_{(2)}, \ldots, \mathbf{z}_{(m)} \end{bmatrix}$. These sub-vectors are defined by the genome positions, each spanning 1 MB genome block, i.e. chr1: 0–1 MB, chr1: 1–2 MB, etc. We denote the variance matrix of $\mathbf{z}$ as $\boldsymbol{\Sigma}$ and it can be written as the block matrix form. Denote $\boldsymbol{\Sigma}_{i,j} = \mathrm{cov}[\mathbf{z}_{(i)}, \mathbf{z}_{(j)}]$ as the submatrix of $\boldsymbol{\Sigma}$, with rows indexed by the $i$th block $\mathbf{z}_{(i)}$ and columns indexed by the $j$th block $\mathbf{z}_{(j)}$. Then we use a block-wise tridiagonal matrix to approximate $\boldsymbol{\Sigma}$ by shrinking $\boldsymbol{\Sigma}_{i,j}$ to 0 if $|i - j| \ge 2$. This approximation is reasonable in the context of GWAS since SNPs should be independent if they are physically apart. Then, we can use an iterative approach to generate each block $\mathbf{z}_{(i)}$ by conditioning on the previous block $\mathbf{z}_{(i-1)}$ via the

conditional normal distribution:

$$(\mathbf{z}_i | \mathbf{z}_{i-1} = \mathbf{a}) \sim \mathrm{N}\left(\boldsymbol{\Sigma}_{i,i-1}\boldsymbol{\Sigma}_{i-1,i-1}^{-1}\mathbf{a}, \boldsymbol{\Sigma}_{i,i} - \boldsymbol{\Sigma}_{i,i-1}\boldsymbol{\Sigma}_{i-1,i-1}^{-1}\boldsymbol{\Sigma}_{i-1,i}\mathbf{a}\right).$$

In practice, $\boldsymbol{\Sigma}_{i,i}$ may be rank deficient and therefore not invertible. We adopt the truncated singular value decomposition (TSVD) method[84] and use the top $q$ singular values and their corresponding singular vectors to calculate the inverse matrix. For numerical stability, we choose $q$ to be as large as possible such that the conditional number is <1000[85]. Finally, we standardize each pseudo $z$-score vector so that it has the same mean and variance as the $z$-score vector in real data.

**Application to binary traits.** So far, we have based the derivation on the setting that the both input traits are continuous. This is a common approach to introducing genetic correlation methodology[10,11]. However, most genetic correlation methods, including LOGODetect, can be directly applied to GWAS summary statistics of binary outcomes[10,11]. It is known that under the liability threshold model, the following formulas hold[10]:

$$h_{\beta,\mathrm{obs}}^2 = \frac{h_\beta^2 \phi(\tau_1)^2 S_1(1-S_1)}{P_1^2(1-P_1)^2}, \tag{9}$$

$$\rho_{\mathrm{g,obs}} = \rho_\mathrm{g} \frac{\sqrt{\phi^2(\tau_1)\phi^2(\tau_2)S_1(1-S_1)S_2(1-S_2)}}{P_1(1-P_1)P_2(1-P_2)}, \tag{10}$$

where $h_{\beta,\mathrm{obs}}^2$ and $\rho_{\mathrm{g,obs}}$ denote heritability and genetic covariance on the observed scale, respectively; $P_1$ and $P_2$ denote population prevalence for two traits; $S_1$ and $S_2$ denote sample prevalence for two traits; $\tau_1 = \Phi^{-1}(1-S_1)$, $\tau_2 = \Phi^{-1}(1-S_2)$, $\phi$ and $\Phi$ denote the standard normal distribution density and its cumulative distribution function, respectively. When applying LOGODetect to binary traits, we replace $h_\beta^2, h_\gamma^2$ (i.e., heritability on the liability scale) with $h_{\beta,\mathrm{obs}}^2, h_{\gamma,\mathrm{obs}}^2$ (i.e., heritability on the observed scale).

**Extension for sample overlaps.** Suppose there are $n_s$ shared samples in the two GWASs, then the linear models can be restated as

$$\begin{bmatrix} \mathbf{y}_{1,\mathrm{ns}} \\ \mathbf{y}_{1,\mathrm{s}} \end{bmatrix} = \begin{bmatrix} \mathbf{X}_{\mathrm{ns}} \\ \mathbf{X}_{\mathrm{s}} \end{bmatrix} \boldsymbol{\beta} + \begin{bmatrix} \boldsymbol{\epsilon}_{\mathrm{ns}} \\ \boldsymbol{\epsilon}_{\mathrm{s}} \end{bmatrix},$$
$$\begin{bmatrix} \mathbf{y}_{2,\mathrm{ns}} \\ \mathbf{y}_{2,\mathrm{s}} \end{bmatrix} = \begin{bmatrix} \mathbf{Z}_{\mathrm{ns}} \\ \mathbf{Z}_{\mathrm{s}} \end{bmatrix} \boldsymbol{\gamma} + \begin{bmatrix} \boldsymbol{\delta}_{\mathrm{ns}} \\ \boldsymbol{\delta}_{\mathrm{s}} \end{bmatrix}, \tag{11}$$

where $\begin{bmatrix} \mathbf{y}_{1,\mathrm{ns}} \\ \mathbf{y}_{1,\mathrm{s}} \end{bmatrix}$, $\begin{bmatrix} \mathbf{y}_{2,\mathrm{ns}} \\ \mathbf{y}_{2,\mathrm{s}} \end{bmatrix}$ are the standardized phenotypes of all individuals in each GWAS. $\begin{bmatrix} \mathbf{X}_{\mathrm{ns}} \\ \mathbf{X}_{\mathrm{s}} \end{bmatrix} = \mathbf{X}$, $\begin{bmatrix} \mathbf{Z}_{\mathrm{ns}} \\ \mathbf{Z}_{\mathrm{s}} \end{bmatrix} = \mathbf{Z}$ are standardized genotypes of all individuals in each GWAS. $\boldsymbol{\epsilon}_{\mathrm{ns}}, \boldsymbol{\epsilon}_{\mathrm{s}}, \boldsymbol{\delta}_{\mathrm{ns}}, \boldsymbol{\delta}_{\mathrm{s}}$ are the non-genetic effects where $\mathrm{cov}[\boldsymbol{\epsilon}_{\mathrm{s}}, \boldsymbol{\delta}_{\mathrm{s}}] = \rho_e \mathbf{I}_{n_s}$. It can be shown that

$$\mathrm{cov}[\mathbf{z}_1, \mathbf{z}_2] = \frac{\rho_g}{\sqrt{n_1 n_2}K} \mathbf{X}^\mathrm{T}\mathbf{X}\tilde{I}_M\mathbf{Z}^\mathrm{T}\mathbf{Z} + \frac{\rho_e}{\sqrt{n_1 n_2}}\mathbf{X}_\mathrm{s}^\mathrm{T}\mathbf{Z}_\mathrm{s}, \tag{12}$$

While $\mathrm{var}[\mathbf{z}_1]$ and $\mathrm{var}[\mathbf{z}_2]$ have the same form as no sample overlaps setting. By using reference panel, $\frac{1}{n_s}\mathbf{X}_\mathrm{s}^\mathrm{T}\mathbf{Z}_\mathrm{s}$ can be replaced by $\mathbf{V}$. Therefore, under $H_0$, the combined $z$-score vectors asymptotically follows multivariate normal distributions

$$\begin{bmatrix} \mathbf{z}_1 \\ \mathbf{z}_2 \end{bmatrix} \sim \mathrm{N}\left( \begin{bmatrix} 0 \\ 0 \end{bmatrix}, \begin{bmatrix} \frac{n_1 h_\beta^2}{M}\widetilde{\mathbf{V}^2} + (1-h_\beta^2)\mathbf{V} & \frac{\rho_e n_s}{\sqrt{n_1 n_2}}\mathbf{V} \\ \frac{\rho_e n_s}{\sqrt{n_1 n_2}}\mathbf{V} & \frac{n_2 h_\gamma^2}{M}\widetilde{\mathbf{V}^2} + (1-h_\gamma^2)\mathbf{V} \end{bmatrix} \right)$$

Note that the variance matrix can be split into two terms as

$$\mathrm{Var}\begin{bmatrix} \mathbf{z}_1 \\ \mathbf{z}_2 \end{bmatrix} = \begin{bmatrix} \frac{n_1 h_\beta^2}{M}\widetilde{\mathbf{V}^2} + (1-h_\beta^2)\mathbf{V} & \frac{\rho_e n_s}{\sqrt{n_1 n_2}}\mathbf{V} \\ \frac{\rho_e n_s}{\sqrt{n_1 n_2}}\mathbf{V} & \frac{n_2 h_\gamma^2}{M}\widetilde{\mathbf{V}^2} + (1-h_\gamma^2)\mathbf{V} \end{bmatrix}$$
$$= \begin{bmatrix} \frac{n_1 h_\beta^2}{M}\widetilde{\mathbf{V}^2} + (1-h_\beta^2 - \frac{\rho_e n_s}{\sqrt{n_1 n_2}})\mathbf{V} & 0 \\ 0 & \frac{n_2 h_\gamma^2}{M}\widetilde{\mathbf{V}^2} + (1-h_\gamma^2 - \frac{\rho_e n_s}{\sqrt{n_1 n_2}})\mathbf{V} \end{bmatrix} + \begin{bmatrix} \frac{\rho_e n_s}{\sqrt{n_1 n_2}}\mathbf{V} & \frac{\rho_e n_s}{\sqrt{n_1 n_2}}\mathbf{V} \\ \frac{\rho_e n_s}{\sqrt{n_1 n_2}}\mathbf{V} & \frac{\rho_e n_s}{\sqrt{n_1 n_2}}\mathbf{V} \end{bmatrix}, \tag{13}$$

if $\rho_e n_s$ is positive, and can be split into two terms as

$$\mathrm{var}\begin{bmatrix} \mathbf{z}_1 \\ \mathbf{z}_2 \end{bmatrix} = \begin{bmatrix} \frac{n_1 h_\beta^2}{M}\widetilde{\mathbf{V}^2} + (1-h_\beta^2 + \frac{\rho_e n_s}{\sqrt{n_1 n_2}})\mathbf{V} & 0 \\ 0 & \frac{n_2 h_\gamma^2}{M}\widetilde{\mathbf{V}^2} + (1-h_\gamma^2 + \frac{\rho_e n_s}{\sqrt{n_1 n_2}})\mathbf{V} \end{bmatrix} + \begin{bmatrix} -\frac{\rho_e n_s}{\sqrt{n_1 n_2}}\mathbf{V} & \frac{\rho_e n_s}{\sqrt{n_1 n_2}}\mathbf{V} \\ \frac{\rho_e n_s}{\sqrt{n_1 n_2}}\mathbf{V} & -\frac{\rho_e n_s}{\sqrt{n_1 n_2}}\mathbf{V} \end{bmatrix}, \tag{14}$$

if $\rho_e n_s$ is negative. We can independently simulate pseudosamples following the normal distribution with mean 0 and each variance term, respectively. Finally, by adding up two vectors simulated with respect to different variance terms, we get the pseudo $z$-score vector of interest. In particular, the parameters $\sigma_\beta^2, \sigma_\gamma^2, \rho_e n_s$ appearing in the $z$-score null distribution are not of our interest, but we need their values while doing Monte Carlo sampling of $\begin{bmatrix} \mathbf{z}_1 \\ \mathbf{z}_2 \end{bmatrix}$. We adopt LDSC[10] to estimate

them. Note that LDSC is based on random effect random design model setup, which is incompatible with our model assumption, yet we believe it should yield little consequence.

**Genome partition and FDR control.** We separated the genome into 204 LD blocks using ldetect[86]. Each LD block spans 15 MB on average. We applied LOGODetect to each LD block separately and identified the local regions with $p$-value < 0.05 under a family-wise type I error control. We aggregated all the candidate regions across different LD blocks, and applied Benjamini–Hochberg procedure[87] to control FDR with a cutoff of 0.05, accounting for the multiple testing problem concerning all LD blocks.

**Computation time.** The major computation step in LOGODetect is to compute the maximal scan statistic in real data and in Monte Carlo samples. The computation time depends on the number of SNPs in GWAS. For a typical GWAS with 6 million SNPs, it takes about 12 h on a 2.5GHz cluster with 22 computation cores.

**Simulation settings.** Based on 503 individuals with European ancestry from the 1000 Genomes Project Phase 3 data, we simulated genotype data for 100,000 individuals with minor allele frequency (MAF) > 5% on chromosome 1 using HAPGEN2[29]. 336,532 variants remained in the dataset after removing strand-ambiguous SNPs. Samples were randomly divided into two subsets with equal sample size, each with 50,000 individuals. We used each subset to simulate the phenotype data.

First, we performed simulations under the null hypothesis to see whether our approach would produce false positive findings. We follow the infinitesimal model, where the effect size level of all the normalized SNPs are the same, and the per-normalized-SNP genetic effect was drawn from a normal distribution $\mathrm{N}(0, \frac{h^2}{336,532})$ for both traits. To realistically model the polygenic genetic architecture with different levels of genetic effects, we attributed 30% of the trait heritability to 5000 randomly chosen SNPs, while the remaining SNPs explain 70% of the trait heritability. The per-SNP genetic effect was drawn from a normal distribution $\mathrm{N}(0, 0.3 * \frac{h^2}{5000})$ for SNPs with high heritability enrichment, and from $\mathrm{N}(0, 0.7 * \frac{h^2}{331,532})$ for SNPs with low heritability enrichment. The trait heritability $h^2$ was set to vary from 0.01 to 0.05 in each scenario. Note that a heritability value of 0.01 or 0.05 on chromosome 1 will approximately correspond to heritability values of 0.12 or 0.60 in the whole genome, which are realistic values for typical GWAS traits. Each simulation setting was repeated for 100 times.

Next, we performed simulations to assess the statistical power under a heritability enrichment model. We randomly selected $N = 5$ segments, each containing $L = 1000$ SNPs, as the signal regions shared between two traits. We attributed $p = 0.3$ of trait heritability to the signal regions. The genetic effect size for the SNPs in the signal regions follows a multivariate normal distribution

$$\begin{bmatrix} \beta_i \\ \gamma_i \end{bmatrix} \sim \mathrm{N}\left( \begin{bmatrix} 0 \\ 0 \end{bmatrix}, \begin{bmatrix} \frac{ph^2}{NL} & \frac{ph^2\rho}{NL} \\ \frac{ph^2\rho}{NL} & \frac{ph^2}{NL} \end{bmatrix} \right).$$

The genetic effect size for the SNPs outside the signal regions follows a different multivariate normal distribution without local genetic correlation.

$$\begin{bmatrix} \beta_i \\ \gamma_i \end{bmatrix} \sim \mathrm{N}\left( \begin{bmatrix} 0 \\ 0 \end{bmatrix}, \begin{bmatrix} \frac{(1-p)h^2}{336532-NL} & 0 \\ 0 & \frac{(1-p)h^2}{336532-NL} \end{bmatrix} \right).$$

The trait heritability $h^2$ was set to vary from 0.01 to 0.05, and the correlation of genetic effect size of two traits $\rho$ was set to 0.9. Each simulation setting was repeated for 100 times. Further simulation settings are described in detail in the Supplementary Notes.

We adjusted the significance cutoff of different approaches to achieve the same type I error. For coloc and gwas-pw, in those heritability settings with empirical type I error >0.05, we increased the cutoff of the posterior probabilities so that the empirical type I error is controlled at 0.05.

**Evaluate model performance.** We use three different metrics to quantify the performance of our approach. Denote the true signal segments as $R_1, \ldots, R_J$, and the segments detected by LOGODetect as $\hat{R}_1, \ldots, \hat{R}_K$. We define the signal points detection rate as the number of true signal SNPs detected by LOGODetect divided by the number of true signal SNPs, that is $\frac{\sum_{j=1}^J |R_j \cap (\cup_{k=1}^K \hat{R}_k)|}{\sum_{j=1}^J |R_j|}$. Similarly, we define signal segments detection rate as the number of true signal segments detected by LOGODetect divided by the number of true signal segments, namely $\frac{\sum_{j=1}^J I\{R_j \cap (\cup_{k=1}^K \hat{R}_k) \neq \emptyset\}}{J}$, where we call a segment true positive if it overlaps with a true signal segment. Signal points detection rate and signal segments detection rate aim to measure the sensitivity at the SNPs level and segments level, respectively. To take the extent of the overlap into consideration, we also followed[88] to define $S(R_j)$, the $G$-score with respect to a signal region $R_j$, as $\max_{1 \le k \le K} \frac{|\hat{R}_k \cap R_j|}{\sqrt{|\hat{R}_k||R_j|}}$, and further define the

$G$-score measure as $\frac{1}{J}\sum_{j=1}^{J} S(R_j)$. The $G$-score aims to measure the specificity and sensitivity together. The three metrics were also applied to quantify ρ-HESS, coloc, and gwas-pw.

**Implementation of different methods**. We used ldetect[86] to pre-specify 1703 approximately LD-independent blocks (spanning 1.6 Mb on average) as candidate genomic regions, as suggested by ρ-HESS and gwas-pw. We also used these LD-independent blocks as candidate genomic regions for coloc. In simulation studies, we used 133 approximately LD-independent regions in chromosome 1 as the pre-specified genomic regions for ρ-HESS, coloc, and gwas-pw. For ρ-HESS, the 1000 Genomes Project Phase 3 data[30] was used as the reference panel, the number of eigenvectors used in the truncated-SVD for LD matrix inversion is determined as 50 by default, and the minimum eigenvalue cut off in truncated-SVD is determined as 1.0 by default, as suggested by the ρ-HESS software (https://huwenboshi.github.io/hess/). ρ-HESS reported the estimate and significance of local genetic correlation for each candidate genomic region, and we applied Benjamini–Hochberg procedure[87] to control FDR with a cutoff of 0.05, accounting for the multiple testing problem concerning all genomic regions. Coloc (https://CRAN.R-project.org/package=coloc) and gwas-pw (https://github.com/joepickrell/gwas-pw) estimated the posterior probability that two traits shared at least one causal SNP for each genomic region, and those genomic regions with posterior probability above 0.95 are determined as identified regions.

We used LDSC (https://github.com/bulik/ldsc) to estimate heritability in each chromosome. Stratified-LDSC was used to estimate genetic covariance of the identified regions. In detail, we manually created two annotations: the identified regions and the remaining genome, then we ran the standard LDSC software to calculate the genetic covariance and the proportion of genetic covariance of each annotation. For both LDSC and stratified-LDSC, LD scores were computed with the standard LDSC software from 503 individuals with European ancestry from the 1000 Genomes Project Phase 3 data. Both methods were applied with an unconstrained intercept, using all SNPs as observations in the dependent variable and LD scores as regression weights.

**Application of LOGODetect to seven neuropsychiatric traits**. We applied LOGODetect to seven neuropsychiatric traits. The European ancestry genotype data from 1000 Genomes Project was used as the reference panel to estimate the LD matrix. For each GWAS data, indels and SNPs not present in the reference panel were removed. The SNPs of MAF < 0.01 in the reference panel were also removed. Then for each trait pair, we filtered out all the strand-ambiguous SNPs and took the overlaps. For SNPs whose effect alleles were the same in the two GWASs, the original z-scores were used. For SNPs whose effect alleles were reversed in two GWASs, we reversed the sign of z-score in the second GWAS accordingly. Thus, the allele coding schemes between any two studies were consistent. Then we applied LOGODetect to perform the downstream analysis.

**Enrichment analysis**. We aggregated 227 non-overlapping segments identified by LOGODetect in seven neuropsychiatric traits and investigated if these segments are enriched in predicted functional regions for a given tissue or cell type. Tissue or cell type-specific functional regions were defined using GenoSkyline-Plus annotations and dichotomized with a cutoff of 0.5. The annotation is robust to the cutoff due to the bimodal pattern in raw GenoSkyline-Plus annotation scores. To assess the statistical significance of enrichment, we randomly selected 227 non-overlapping segments across the genome while matching their sizes with the detected segments, and calculated the overlaps with GenoSkyline-Plus annotations. We repeated the permutation procedure 100,000 times to evaluate the significance of the observed overlap.

We also assessed whether the detected regions were enriched in non-brain tissue types after adjusting for the overlap of brain and non-brain annotations. Specifically, for the pancreatic islets cell type annotation, we removed the annotations that overlap with any of the eight significant brain cell type annotations to define the conditional annotation of pancreatic islets. The same procedure was taken to define the conditional annotation of mononuclear cells from peripheral blood. Afterwards, permutation tests were performed on these two conditional annotations. We performed conditional analysis on six generic annotations including coding regions, enhancers, introns, promoters, 5′UTRs and 3′UTRs (extended by a 500-bp window around each of the annotations) in Finucane et al. [52] by removing the overlapped regions between each generic annotation and the brain tissue-specific annotations (merged from eight significant brain cell type annotations). We used permutation test to assess the statistical significance of enrichment in conditional analyses.

Using GENCODE V33lift37 on the UCSC genome browser, we extracted 968 genes with recognized Ensembl IDs in the genomic regions found to harbor local genetic correlations among seven neuropsychiatric traits. We used FUMA[53] to run the Gene Ontology enrichment analysis with these 968 genes.

**Reporting summary**. Further information on research design is available in the Nature Research Reporting Summary linked to this article.

## Data availability
Summary statistics data of five psychiatric disorder were downloaded on the PGC website, http://www.med.unc.edu/pgc/downloads; Summary statistics data of neuroticism and intelligence were downloaded at the website of the Department of Complex Trait Genetics, CNCR, https://ctg.cncr.nl/software/summary_statistics; Summary statistics data of body-mass index and height were downloaded on the GIANT consortium website http://portals.broadinstitute.org/collaboration/giant/index.php/GIANT_consortium_data_files; Summary statistics for bipolar disorder, schizophrenia, body-mass index, and height in the replication cohort were downloaded from UK Biobank repository, http://www.nealelab.is/uk-biobank; phase 3 of the 1000 Genomes Project ftp://ftp.1000genomes.ebi.ac.uk/vol1/ftp/release/20130502/; FUMA, https://fuma.ctglab.nl/; LDSC, https://github.com/bulik/ldsc; coloc, https://CRAN.R-project.org/package=coloc;ρ-HESS, https://huwenboshi.github.io/hess/; gwas-pw, https://github.com/joepickrell/gwas-pw; 66 GenoSkyline-Plus cell-type specific functional annotations, http://genocanyon.med.yale.edu/GenoSkyline. Source data are provided with this paper.

## Code availability
LOGODetect software is available at https://github.com/ghm17/LOGODetect (https://doi.org/10.5281/zenodo.4559388).

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

## Acknowledgements

This study makes use of summary statistics from the Psychiatric Genomics Consortium and the UK Biobank. We thank the investigators for providing publicly accessible GWAS summary statistics. This study makes use of data generated by the Wellcome Trust Case Control Consortium. A full list of the investigators who contributed to the generation of the data is available from www.wtccc.org.uk. L.H. acknowledges research support from the National Science Foundation of China (Grant Nos. 11601259, 12071243) and Shanghai Municipal Science and Technology Major Project (Grant No. 2017SHZDZX01). J.J.L. and Q.L. acknowledge research support from the Waisman Center pilot grant program at University of Wisconsin-Madison.

## Author contributions

H.G., Q.L. and L.H. designed the idea. H.G. performed data analysis. H.G. developed the software. H.G., J.J.L., Q.L., and L.H. interpreted the results. H.G., Q.L., and L.H. drafted the manuscript.

## Competing interests

The authors declare no competing interests.
