## [Peer Review File · Nature Communications]

Reviewers' comments:

Reviewer #1 (Remarks to the Author):

Summary

The authors proposed a scan statistic to assess local genetic correlations between two traits. The validity of this method is not entirely clear due to multiple issues in the reported simulations (Major Comment 1) and data analyses (Major Comment 2, Other Comments 3-5).

Major Comments

1. Issues in simulations

1A. To perform a valid statistical power comparison, one must ensure all methods achieve the same level of type I error. (If a method like rho-HESS shows inflated results in the null simulations, the one should adjust the significance cutoff accordingly in the power simulations such that it achieves the same type I error as the well-calibrated method like LOGODetect.) Indeed the authors seemed to be aware of this issue, as they wrote "type I error was inflated for rho-HESS and therefore ..." on page 5. Due to the potentially different type I error rates across methods, it remains unclear whether LOGODetect is truly more statistically powerful than rho-HESS in simulations, purely based on data shown in Fig. 2.

1B. As highlighted in the Abstract, LOGODetect "automatically identifies genetic regions showing consistent associations with multiple phenotypes". Indeed, methods in Ref. 17 and 77 are suitable for the same analysis. To assess this feature, the author should compare LOGODetect with Ref. 17 and 77 via simulations, and discuss the results.

1C. As detailed on page 18, all simulations in this work assumed the infinitesimal model where the effect of each SNP was drawn from some normal distribution. This is a sensible scenario because theoretical derivations of LOGODetect were built on the same assumption. The authors should also check whether LOGODetect yields robust results when the underlying effect sizes are not generated from normal distributions. For example, how will LOGODetect behave if effects of SNPs are drawn from a heavy-tail distribution (t, Laplace), and/or, most SNPs have zero effects?

1D. In addition, all simulations in this work assumed two traits had the same total heritability (h^2 , pages 18-19). This is different from real GWAS data. Can the authors report results based on two traits with different heritability values?

2. Issues in data analyses

2A. The authors assessed LOGODetect in the analysis of 5 psychiatric disorders (binary traits); however, both theoretical derivations and simulations were based on continuous traits (pages 13-18). To bridge this logical gap, the authors should assess the method on real data of continuous traits as in Ref. 12.

2B. I wonder if the replication analysis on page 8 could be complemented by a more direct approach as follows. The authors could apply LOGODetect to the UKBB summary data of SCZ and BIP, and then check how many of 33 identified SCZ-BIP segments (page 6) are replicated here. Compared with the authors' replication analysis (Table 1), this suggested approach seems more straightforward, and more consistent with current GWAS literature.

2C. Enrichments shown in Fig. 5A are not necessarily "tissue-specific", because they could be potentially driven by generic annotations such as intron and promoter regions. This is not accounted for in the current analysis (pages 19-20). At minimum, the authors should compare

enrichments of generic annotations with the enrichments shown in Fig. 5A. A better analysis is to compute tissue-specific enrichments conditioning on multiple generic annotations.

2D. Can the hub regions shown in Fig. 6 be identified by previous methods such as rho-HESS (Ref. 12) or colocalization methods like Ref. 17 and 77? If yes, then what is the additional gain of using LOGODetect here? If no, then the authors should highlight this. Following the same logic, can genes highlighted on pages 12-13 be identified by previous methods? (This comment is related to Major Comment 1B).

2E. All analyses here were based on setting $\theta=0.5$ (pages 5 & 15). I understand this value was chosen based on simulation performance. However, the simulated data and real GWAS data are different in many aspects. If the authors cannot provide a more principled way to estimate θ from real data, then at least they should analyze the real GWAS data with different values of θ , and check whether the results are robust to θ .

Other Comments

1. On page 4, for the same SNP i , are z_{1i} and z_{2i} based on the same allele coding scheme (i.e. do they have the same effect allele)? Please clarify.
2. On page 4 or Fig. 1, please elaborate on how LOGODetect "scans the genome while allowing the segment size to vary". This seems to be a computationally expensive step, but I cannot find enough information on how the authors deal with it in the Methods section (page 15).
3. On page 7 the authors claimed that "the exact sample overlap across studies is unknown". To support this claim, the authors should review the supplementary information of Ref.s 33-37 and summarize what they find here. (Usually a GWAS publication will have a supplementary table listing cohort information, which can be used to assess sample overlap.) If the authors happen to be able to approximate the sample overlap from the supplementary information, they should report a version of results by accounting for the (approximate) sample overlap.
4. When comparing rho-HESS to LOGODetect on real GWAS data, the authors used "pre-specify regions" in rho-HESS. How are these regions defined? What is the size of "pre-specify regions"? Can the authors try different sizes of "pre-specify regions" in rho-HESS analysis, using the inferred segment size from LOGODetect (24 Kb - 1.6 Mb, page 6). Is it possible that the rho-HESS results in Fig. 4 and Table 1 could be improved by using "better" defined regions?
5. The number of cases in UKBB BIP and SCZ datasets (page 8) are quite small compared with published BIP and SCZ GWAS (Ref.s 33-34). Is LOGODetect robust to case-control ratios? This is not clear from the mathematical formulas, and it is not assessed by simulations.
6. To assess the "functional relevance" of identified local genetic correlations, I wonder if the authors could also run a standard Gene Ontology analysis, in addition to the tissue-specific annotation analysis (page 9).
7. On page 12, "hi-C" should be "Hi-C" instead.
8. The discussion section (pages 12-13) does not summarize the limitations of LOGODetect. Any manuscript describing a new method must have such discussions. An obvious limitation of LOGODetect compared to rho-HESS (Ref. 12) and LDSC (Ref. 10) is that LOGODetect only tests the significance of genetic correlation, but cannot estimate it (please see Fig. 3 of Ref. 12).
9. On page 14, I do not understand why the authors say that their method "does not assume the global genetic covariance to be the same across all the SNPs in the whole genome". To me the "global genetic covariance" is just a single number measuring how genetically correlated two traits

are. Why should this single number be varying across the genome?

10. On page 15, the claim that "the expected value of [sum] is larger in regions with strong LD" is non-trivial. Either mathematical derivations or numerical simulations should be shown here.

11. The citation for WTCCC data seems missing.

12. The authors did not add page or line numbers in the submitted main text file, which makes the review process less convenient.

Reviewer #2 (Remarks to the Author):

The paper proposes a statistic to identify regions of the genome that contribute to two traits. Not surprisingly, the statistic is based on the cross product of z scores over SNPs within a region R. The denominator is an LD Score for R to correct over-weighting of high LD regions, the LD score being "tempered" by raising to power θ , and the authors decide to set $\theta = 0.5$. Calculation of the statistic for an R is cheap, and so can be done for many R in order to search for the best value: the authors consider all R with up to C SNPs for some C.

While the numerator of the statistic seems reasonable, there is little justification given for the precise form of the denominator: that it should be some increasing function of LD is clear, but the particular functional form seems arbitrary and the justification for the choice $\theta = 0.5$ is weak, there seems to be no formal criteria and in any case it is based on unrealistic simulations.

The authors report good performance of their method in a simulation study, relative to only one alternative, rho-Hess from Shi et al 2017. The simulation study is simplistic with only two effect sizes for SNPs. In particular the simulated effects were independent of LD and so failed to check the main feature of the statistic which is how it models the LD-effect size relationship.

In the analysis of real data (5 psychiatric traits), the new method and rho-Hess were compared based on "enrichment of genetic covariance", but this was not defined and there has been controversy over different approaches to enrichment. Moreover the enrichment was based on LD Score and the authors method is also based on LD Score, so it hardly seems a fair metric for comparison. In particular the finding of lower enrichment in regions detected by rho-HESS alone seems of little value.

Figure 2 seems to show a high detection rate when the heritability is close to zero, which needs discussion; it seems implausible, and may be a consequence of the unrealistic simulation model.

Overall I think the new statistic is poorly justified and the evidence presented for its properties is weak.

Reviewer #3 (Remarks to the Author):

Guo et al provide a well-written manuscript, with a flexible method to estimate the local correlation of SNP effects (or local genetic correlations) that does not depend on pre-defined annotations/regions and will likely be very useful for the human genetics community.

Suggestions:

1. This work compares LOGODetect performance with rho-HESS performance (using both simulations and real data analysis). While I think this is a useful comparison to make, rho-HESS defines "genetic correlation" differently from LOGODetect or cross-trait LDSC. Rho-HESS (along

with quantitative genetic theory) defines genetic correlation to be the correlation between additive genetic values while LOGODetect and LDSC appear to define genetic correlation as the correlation of SNP effects.

How might differences in genetic correlation definitions impact the comparison of LOGODetect and rho-HESS? Can you discuss how differences in type-I error or resolution may arise from differences in genetic correlation definitions?

2. If I'm understanding correctly, one simulation was used to assess type-I error while another simulation was used to assess power. While the results from these simulations are informative, is it possible to assess error and power under the same set of simulated data? Particularly, how is type-I error affected when there are true signal segments with genetic correlations?

3. In order to perform inference, the test statistic Q_{\max} is compared against the null distribution of Q_{\max} estimated from MC methods. It's not clear to me how family-wise error is controlled. If $\max(|Q(R)|)$ is approximately $Q_{0.95}$, wouldn't that result in a p-value that is approximately 0.05? Could you elaborate on how multiple testing is accounted for?

4. When sampling z-scores from the null, it's unclear how the vector z is decomposed into z_1, z_2, z_3 . How were these 1MB regions defined?

5. In your model where sample overlap exists, the first equation under the header "Extension for sample overlaps" appears to be missing SNP effects, β and γ .

Response to Reviewer #1:

1A. *To perform a valid statistical power comparison, one must ensure all methods achieve the same level of type I error. (If a method like rho-HESS shows inflated results in the null simulations, the one should adjust the significance cutoff accordingly in the power simulations such that it achieves the same type I error as the well-calibrated method like LOGODetect.) Indeed the authors seemed to be aware of this issue, as they wrote "type I error was inflated for rho-HESS and therefore ..." on page 5. Due to the potentially different type I error rates across methods, it remains unclear whether LOGODetect is truly more statistically powerful than rho-HESS in simulations, purely based on data shown in Fig. 2.*

Response:

Thank you for the constructive comment. Indeed, in the simulations of our original submissions, ρ -HESS showed inflated type I error rate under the heritability enrichment model, where 30% of trait heritability was attributed to 3% randomly selected SNPs and 70% of trait heritability was attributed to the remaining SNPs. When trait heritability was set to 0.5 and 0.4 in simulations, the type I error of ρ -HESS was 0.732 and 0.278, respectively, under a significance cutoff of 0.05 (**Table 1**). Following the reviewer's suggestion, we adjusted the critical value of ρ -HESS in those simulation settings to maintain the type I error rate at 0.05, so that power can be compared to other methods. In addition, in response to Comment-1B, we added two colocalization methods, coloc¹ and gwas-pw², into the simulations. Both coloc and gwas-pw are Bayesian approaches that calculate the posterior probability of colocalization as output. By default, we used 0.95 as the cutoff of posterior probabilities. For simulation settings in which colocalization methods showed inflated type I error rates, we increased the cutoff of the posterior probabilities so that the empirical rate of type I error is controlled at 0.05.

Table 1. Type I error rates under a heritability enrichment model. Type I error rates denote the proportion of simulations that significant segments harboring local genetic correlation are identified in null simulations. In these simulations, 30% of trait heritability was assigned to 500 randomly selected SNPs and 70% trait heritability was assigned to the other SNPs. Two cohorts did not have any overlapping samples. Type I error of colocalization approaches is discussed in detail in the response to comment 1B. Simulation settings with type I error rate larger than 0.1 are highlighted in the red box.

Type I error rate at significance level $\alpha=0.05$		LOGODetect			ρ -HESS
		$\theta=0.4$	$\theta=0.5$	$\theta=0.6$	
Trait heritability	0.9	0.028	0.029	0.036	1
	0.8	0.021	0.029	0.034	1
	0.7	0.027	0.035	0.045	1
	0.6	0.032	0.034	0.04	0.97
	0.5	0.028	0.029	0.04	0.732
	0.4	0.04	0.039	0.047	0.278
	0.3	0.04	0.044	0.053	0.029
	0.2	0.039	0.041	0.051	0
	0.1	0.059	0.066	0.054	0
	0.05	0.048	0.056	0.063	0
	0.03	0.042	0.053	0.056	0
	0.01	0.041	0.051	0.055	0

After type I error adjustment, we compared the performance of our method with that of ρ -HESS, coloc, and gwas-pw. The results are shown below in **Figure 1**.

Figure 1. Assessment of statistical power under a heritability enrichment model with varying trait heritability. The X-axis shows the heritability for simulated traits. The Y-axis shows the statistical power assessed by three different measures: (A) signal points detection rate, (B) signal segments detection rate, and (C) G-score. Due to the observed inflation in type I error rates, we adjusted the significance cutoff for ρ -HESS, coloc, and gwas-pw so that their type I error rates achieve 0.05.

As shown in **Table 1** and **Figure 1**, when heritability increases, LOGODetect showed improvements in all three measures of statistical power without inflating type I error. Regarding to signal points detection rates, when heritability is low to moderate, gwas-pw performed best. LOGODetect achieved greater points detection rates compared to ρ -HESS and coloc. When heritability is high, ρ -HESS achieved the best points detection rates, and LOGODetect was the second best. Notably, LOGODetect

performs reasonably well in all heritability settings, while the performance of gwas-pw and ρ -HESS is strongly affected by different heritability (**Figure 1A**). Regarding to segment detection rate and G-score, LOGODetect outperformed the other three methods at various heritability settings (**Figures 1B-C**).

Importantly, we note that correction for type I error cannot be easily performed in real data analysis. Here, we recalibrated the significance cutoff of ρ -HESS and the colocalization approaches according to the type I error in null simulations. In practice, actual type I error cannot be obtained on real data. Consequently, if a method has inflated type I error rates, findings will not be statistically justified. We also performed statistical power analysis in other scenarios (**Figures 3-7, Tables 1 and 4-7**), per the comments of the reviewers, and the results are discussed in detail in later sections. In particular, we have adjusted the thresholds for ρ -HESS, coloc, and gwas-pw to make sure type I error is controlled at the same levels in all subsequent comparisons. The set of extensive simulation studies have consistently demonstrated that LOGODetect is statistically more powerful than other methods.

The discussions above have been incorporated into the revised manuscript (page 6, line 146-162; page 22, line 643-647) and **Supplementary Table 2. Figure 2** and the relevant information in the main text have been updated accordingly.

1B. As highlighted in the Abstract, LOGODetect "automatically identifies genetic regions showing consistent associations with multiple phenotypes". Indeed, methods in Ref. 17 and 77 are suitable for the same analysis. To assess this feature, the author should compare LOGODetect with Ref. 17 and 77 via simulations, and discuss the results.

Response:

Thank you for the comment. In the revised manuscript, we compared LOGODetect with two colocalization methods, coloc and gwas-pw, in simulations and analyses of real data.

These two colocalization methods partition the whole genome into independent LD blocks, and estimate the posterior probability that each block contains single causal SNP associated with both traits. There are several drawbacks for coloc and gwas-pw. First, there is not a principled way to choose the cutoff of the posterior probability to determine significance of the block. **Table 2** shows the empirical type I error rates for coloc and gwas-pw under a posterior probability cutoff of 0.95, a commonly used cutoff in practice. The coloc approach showed inflated type I error rate. Here, the simulation setting is identical to what was used in our response to Comment 1A (and **Table 1**).

Table 2. Empirical type I error rates for colocalization methods under a heritability enrichment model. Simulation settings with type I error rate larger than 0.1 are highlighted in the red box.

Empirical type I error rate with posterior probability cutoff being 0.95		coloc	gwas-pw
Trait heritability	0.9	0.589	0.034
	0.8	0.514	0.025
	0.7	0.473	0.021
	0.6	0.389	0.029
	0.5	0.277	0.032
	0.4	0.182	0.038
	0.3	0.077	0.037
	0.2	0.01	0.076
	0.1	0.004	0.022
	0.05	0	0.001
	0.03	0	0.001
	0.01	0	0

Second, both methods assume the independence of two GWASs (i.e. no sample overlap), which is often violated in large, biobank/consortium-scale studies. This could lead to technical correlation between the associations in two studies and thus inflated type I error in colocalization (**Table 3**).

Table 3. Empirical type I error rates for colocalization methods with sample overlaps. Here we assume heritability enrichment model that 30% of trait heritability was assigned to 500 randomly selected SNPs and 70% trait heritability was assigned to the other SNPs. Two cohorts have 50% overlapping samples. Simulation settings with type I error rate larger than 0.1 are highlighted in the red box.

Empirical type I error rate with posterior probability cutoff being 0.95	coloc	gwas-pw	
Trait heritability	0.9	0.636	0.071
	0.8	0.588	0.08
	0.7	0.479	0.082
	0.6	0.398	0.087
	0.5	0.279	0.101
	0.4	0.194	0.12
	0.3	0.062	0.143
	0.2	0.023	0.156
	0.1	0.002	0.045
	0.05	0	0.011
	0.03	0	0.007
	0.01	0	0.002

Third, coloc and gwas-pw pre-specify regions as ρ -HESS does. In general, pre-specified candidate regions are much larger than the true signal regions. An appealing property of LOGODetect is that it automatically searches for signal regions with improved precision. An example is shown in **Figure 2**.

Figure 2. LOGODetect is data-adaptive and identifies segments that tightly covers signal regions, while the other methods require signal regions to be pre-specified. The locuszoom plots of a genomic segment are shown for bipolar disorder (upper panel) and schizophrenia (middle panel). The orange band represents the region identified by LOGODetect, and the light purple band represent the pre-specified region (defined by independent LD blocks) used in ρ -HESS, coloc, and gwas-pw.

Here, the orange band denotes the detected region using LOGODetect, which spans about 100KB, and the light purple band denotes the pre-specified region used in ρ -HESS, coloc, and gwas-pw, which spans about 2MB. LOGODetect could precisely detect genomic segments harboring genetic correlation, therefore achieving higher true segments detection rates and G-scores, as demonstrated in our simulations (we have discussed this in detail in our response to comment 1A). Finally, we note that coloc and gwas-pw impose a strong assumption that no more than one causal genetic variant exists in a local region, which can be violated in real case.

The discussions above have been incorporated into the revised manuscript (page 5, line 109-123; page 8, line 231-235), **Supplementary Tables 2 and 5**, and

Supplementary Figure 56. Simulation results have been added to Figure 2 and the main text (page 4, line 104-107).

1C. As detailed on page 18, all simulations in this work assumed the infinitesimal model where the effect of each SNP was drawn from some normal distribution. This is a sensible scenario because theoretical derivations of LOGODetect were built on the same assumption. The authors should also check whether LOGODetect yields robust results when the underlying effect sizes are not generated from normal distributions. For example, how will LOGODetect behave if effects of SNPs are drawn from a heavy-tail distribution (*t*, Laplace), and/or, most SNPs have zero effects?

Response:

This is a great suggestion. We appreciate the comment. As suggested, we first tried t-distribution as the effect size distribution. Technically, this remains an infinitesimal model but there are more SNPs with large effects compared to our original simulations. Here, per-SNP heritability for each variant was set to be the same as before, but the effect size of each SNP follows a t-distribution with 10 degrees of freedom. Under this setting, type I error rate was well-controlled for LOGODetect (**Table 4**), but ρ -HESS still showed type I error inflation when heritability was high (e.g. ≥ 0.4).

Table 4. Type I error rates under an infinitesimal model with t-distributed effects. Simulation settings with type I error rate larger than 0.1 are highlighted in the red box.

Type I error rate at significance level $\alpha = 0.05$		LOGODetect			ρ -HESS
		$\theta=0.4$	$\theta=0.5$	$\theta=0.6$	
Trait heritability	0.9	0.017	0.014	0.015	1
	0.8	0.017	0.018	0.016	1
	0.7	0.019	0.019	0.018	0.999
	0.6	0.022	0.017	0.019	0.963
	0.5	0.019	0.017	0.019	0.73
	0.4	0.014	0.014	0.021	0.266
	0.3	0.034	0.035	0.036	0.035
	0.2	0.028	0.027	0.03	0
	0.1	0.039	0.048	0.054	0
	0.05	0.039	0.054	0.056	0
	0.03	0.043	0.047	0.054	0
	0.01	0.05	0.049	0.055	0

Second, we compared the statistical power of LOGODetect with other methods. We randomly selected $N = 5$ segments, each containing $L = 100$ SNPs, as the signal regions shared between two traits. We attributed $p = 0.3$ trait heritability to the signal regions. The effect size correlation of shared causal SNPs, ρ , is set to be 0.9. The genetic effect size for the SNPs in the signal regions are generated as follows:

$$\beta_i = (\sqrt{\rho}\xi_i + \sqrt{1-\rho}\zeta_i) * \sqrt{\frac{4ph^2}{5NL}};$$

$$\gamma_i = (\sqrt{\rho}\xi_i + \sqrt{1-\rho}\eta_i) * \sqrt{\frac{4ph^2}{5NL}},$$

and genetic effect size for the SNPs outside the signal regions are generated as follows:

$$\beta_i = \sqrt{\frac{4(1-p)h^2}{5(N_{total} - NL)}} \zeta_i;$$

$$\gamma_i = \sqrt{\frac{4(1-p)h^2}{5(N_{total} - NL)}} \eta_i,$$

where ξ_i, ζ_i, η_i independently follow t_{10} distribution and N_{total} is the number of SNPs. The above genetic model guaranteed that the effect sizes are generated proportional to the t-distribution, and the variance explained by all SNPs is exactly equal to the trait heritability, h^2 . We varied the trait heritability to examine the change of power. Under this setting, LOGODetect showed almost universally higher signal segments detection rates and G-scores compared to the other three methods (**Figure 3**). These results are highly consistent with the original simulation results under an infinitesimal model with normal effects (**Figure 1** in the response to Comment 1A).

Figure 3. Assessment of statistical power under an infinitesimal model with t-distributed effects. X-axis represents the trait heritability. Each simulation setting was repeated 100 times.

Next, we evaluated our method under non-infinitesimal models with sparse effects. We split up chromosome 1 into two parts (i.e., chr1:1-116,000,000 and chr1:116,000,001-249,143,646), each containing half of the SNPs. For the first trait, we randomly sampled 1000 variants from the first half of chr1 as causal variants. Similarly, we randomly sampled 1000 variants from the second half of chr1 as causal variants for the second trait. Per-SNP heritability for all the causal variants were the same as previous simulations. Results were comparable to simulations under infinitesimal models. Under this setting, type I error rate was well-controlled for LOGODetect (**Table 5**), but ρ -HESS showed type I error inflation when heritability is high (e.g. ≥ 0.5).

Table 5. Type I error rates under a non-infinitesimal model with sparse effects. Simulation settings with type I error rate larger than 0.1 are highlighted in the red box.

Type I error rate at significance level $\alpha = 0.05$		LOGODetect			ρ -HESS
		$\theta=0.4$	$\theta=0.5$	$\theta=0.6$	
Trait heritability	0.9	0.001	0.001	0.002	1
	0.8	0	0	0	0.986
	0.7	0.001	0.001	0.003	0.859
	0.6	0	0	0.003	0.567
	0.5	0.001	0.001	0.004	0.217
	0.4	0.004	0.002	0.002	0.071
	0.3	0.01	0.012	0.017	0.004
	0.2	0.01	0.014	0.028	0
	0.1	0.031	0.039	0.063	0
	0.05	0.044	0.051	0.056	0
	0.03	0.042	0.036	0.054	0
	0.01	0.046	0.044	0.05	0

To perform power analysis, we considered two scenarios. In the first scenario, we assumed a sparse genetic model with few causal SNPs. We randomly sampled 10 causal regions for each trait, among which 5 causal regions were shared by both traits. We assumed the per-SNP heritability for all the causal variants to be the same, and varied the length of causal regions. Consistent with other simulations, LOGODetect achieved greater signal points detection rates compared to ρ -HESS when signal region length is small to moderate and compared to colocalization approaches in all settings. In addition, LOGODetect obtained universally higher signal segments detection rates and G-scores (**Figure 4**).

Figure 4. Assessment of statistical power under a non-infinitesimal model with sparse effects. X-axis is in log scale. Signal region length, represents the length of one true signal region. The trait heritability is set to be 0.2 for both traits. Each simulation setting is repeated for 100 times.

Further, in another scenario, we assumed a more sophisticated model with sparse large effects and a polygenic background. For each trait, we randomly selected 10 SNPs to have a large heritability enrichment, among which 5 SNPs were shared by two traits. We refer to other SNPs as low-enrichment SNPs. The effect size correlation of shared, high-enrichment SNPs was set to be 0.9. We varied the heritability

proportion explained by the high-enrichment SNPs to see how the power changes. Note that when proportion of heritability explained by the high-enrichment SNPs is 1, this model becomes a simpler sparse model with 10 causal SNPs. LOGODetect and gwas-pw achieved similar performance in three metrics, and they performed universally better than ρ -HESS and coloc (**Figure 5**).

Figure 5. Assessment of statistical power under a non-infinitesimal model with sparse large effects. X-axis represents the heritability explained by the high enrichment SNPs divided by the total trait heritability. The trait heritability is set to be 0.2 for both traits. Each simulation setting is repeated for 100 times.

Taken together, we have demonstrated through additional simulations that LOGODetect can obtain robust results when the underlying effect sizes are drawn from a heavy-tail distribution (i.e. t-distribution), or when most SNPs have zero effects. The discussions above have been incorporated into the revised manuscript (page 5, line 117-118; page 6, line 156-159), **Supplementary Notes**, **Supplementary Tables 7-8**, and **Supplementary Figures 15-17**.

1D. In addition, all simulations in this work assumed two traits had the same total heritability (h^2 , pages 18-19). This is different from real GWAS data. Can the authors report results based on two traits with different heritability values?

Response:

Thank you for the suggestion. In the revision, we conducted simulations where two traits have different heritability values. In the simulations in our original submission, we varied the heritability values from 0.01 to 0.9 for two traits simultaneously. In the new analysis, we fixed the heritability of the second trait to be 0.2, and let the heritability of the first trait vary between 0.01 and 0.9. We observed highly consistent results under these settings. Type I error rate is well-controlled for LOGODetect (**Table 6**), but ρ -HESS showed type I error inflation (type I error rate > 0.1 when heritability ≥ 0.6).

Table 6. Type I error rates under different heritability setting. Simulation settings with type I error rate larger than 0.1 are highlighted in the red box.

Type I error rate at significance level $\alpha = 0.05$		LOGODetect			ρ -HESS
		$\theta=0.4$	$\theta=0.5$	$\theta=0.6$	
Trait heritability	0.9	0.024	0.021	0.023	0.574
	0.8	0.029	0.026	0.026	0.485
	0.7	0.025	0.025	0.029	0.345
	0.6	0.007	0.006	0.007	0.194
	0.5	0.02	0.02	0.022	0.079
	0.4	0.017	0.018	0.019	0.031
	0.3	0.025	0.028	0.031	0.003
	0.2	0.043	0.039	0.045	0.002
	0.1	0.067	0.063	0.064	0
	0.05	0.046	0.044	0.043	0
	0.03	0.033	0.038	0.052	0
	0.01	0.043	0.041	0.046	0

We also compared statistical power of different approaches under this setting. Similar to other simulations in the revised manuscript, we adjusted the posterior probability cutoff of coloc and gwas-pw so that the type I error rate of all methods were controlled at 0.05. The results of power comparison are shown in **Figure 6**. Note that when heritability was less than 0.5, type I error rate for ρ -HESS was well-controlled, so adjustment for ρ -HESS was not necessary. LOGODetect consistently achieved higher signal segments detection rates and G-scores. As to signal points detection rate, the relative performance of the four methods varied as heritability changed. While LOGODetect was the best method when heritability of the first trait is low, and gwas-pw and ρ -HESS achieved higher points detection rates when the heritability was higher. Overall, these results are highly consistent with simulations based on equal heritability, suggesting that our method is robust to unequal heritability values of two GWAS traits.

Figure 6. Assessment of statistical power under a heritability enrichment model with varying heritability for the first trait and fixed heritability for the second trait. X-axis represents the heritability of the first trait, while the heritability of the second trait is fixed at 0.2. For each trait, we randomly choose $N=5$ segments, with each containing $L=100$ SNPs, as the signal regions. The heritability for the signal regions is set to 30% of the total heritability. The correlation of genetic effect size of two traits ρ is set to 0.9. Posterior probability cutoff of coloc and gwas-pw was adjusted so that the empirical type I error rate was 0.05, the same level as the other methods. Each simulation setting is repeated for 100 times.

The discussions above have been incorporated into the revised manuscript (page 5, line 112-113), **Supplementary Notes, Supplementary Table 4, and Supplementary Figure 10.**

2A. The authors assessed LOGODetect in the analysis of 5 psychiatric disorders (binary traits); however, both theoretical derivations and simulations were based on continuous traits (pages 13-18). To bridge this logical gap, the authors should assess the method on real data of continuous traits as in Ref. 12.

Response:

Thank you for the comment. In our original submission, we derived the LOGODetect framework assuming that the both traits are continuous. This is a common approach in genetic correlation methodologies^{3,4}. However, most genetic correlation methods, including LOGODetect, can be directly applied to GWAS summary statistics of binary outcomes^{3,4}. In the revised manuscript, we have added some discussions about applying our method to binary traits. It has been shown that under the liability threshold model, the following formulas hold³,

$$h_{\beta,obs}^2 = \frac{h_{\beta}^2 \phi(\tau_1)^2 S_1(1 - S_1)}{P_1^2(1 - P_1)^2},$$

$$\rho_{g,obs} = \rho_g \frac{\sqrt{\phi^2(\tau_1)\phi^2(\tau_2)S_1(1 - S_1)S_2(1 - S_2)}}{P_1(1 - P_1)P_2(1 - P_2)},$$

where $h_{\beta,obs}^2$ and $\rho_{g,obs}$ denote heritability and genetic covariance on the observed scale, respectively; P_1 and P_2 denote population prevalence for two traits; S_1 and S_2 denote sample prevalence for two traits; $\tau_1 = \Phi^{-1}(1 - S_1)$, $\tau_2 = \Phi^{-1}(1 - S_2)$, ϕ and Φ denote the standard normal distribution density and its cumulative distribution function, respectively. When applying LOGODetect to binary traits, we replace $h_{\beta}^2, h_{\gamma}^2$ (i.e., heritability on the liability scale) with $h_{\beta,obs}^2, h_{\gamma,obs}^2$ (i.e., heritability on the observed scale).

We conducted simulations to demonstrate that LOGODetect works well on binary traits. We simulated two binary traits under the liability threshold model, where the continuous liabilities followed the infinitesimal model. Both population prevalence and sample prevalence were fixed at 0.5. Overall, simulation results are highly consistent with results based on continuous traits. Type I error rate was well-controlled for LOGODetect, while ρ -HESS (type I error rate > 0.1 when heritability \geq 0.6) still had type I error inflation in this setting (**Table 7**).

Table 7. Type I error rates under a liability threshold model. Simulation settings with type I error rate larger than 0.1 are highlighted in the red box.

Type I error rate at significance level $\alpha=0.05$		LOGODetect			ρ -HESS
		$\theta=0.4$	$\theta=0.5$	$\theta=0.6$	
Trait heritability	0.9	0.018	0.02	0.019	0.941
	0.8	0.022	0.019	0.021	0.755
	0.7	0.02	0.024	0.028	0.475
	0.6	0.027	0.024	0.022	0.206
	0.5	0.028	0.031	0.032	0.038
	0.4	0.028	0.029	0.03	0.005
	0.3	0.037	0.04	0.05	0
	0.2	0.044	0.047	0.05	0
	0.1	0.05	0.047	0.048	0
	0.05	0.05	0.048	0.044	0
	0.03	0.04	0.048	0.044	0
	0.01	0.064	0.07	0.063	0

To perform power analysis, we simulated two binary traits under the liability threshold model, where the continuous liabilities followed the same heritability enrichment model as previous simulations. Population prevalence and sample prevalence were set to be the same, varying from 0.01 to 0.5. We compared the statistical power of LOGODetect with other methods. LOGODetect achieved greatest signal points detection rates when sample prevalence is low and performed second best when sample prevalence is high (**Figure 7A**). In addition, LOGODetect obtained universally higher signal segments detection rates and G-scores (**Figure 7B-C**). Overall, LOGODetect showed robust performance under varying sample prevalence.

Figure 7. Assessment of statistical power under a liability threshold model with varying sample prevalence and fixed total sample size. X-axis represents the sample prevalence (the proportion of cases in the cohort). Here we assume the liability threshold model for the binary trait, which means the disease status is determined by an unobserved continuous liability.

Further, we have added two continuous neurological traits, neuroticism and intelligence (abbreviated as NEU and IQ, respectively), into our analysis. In the revised manuscript, we report pair-wise correlation findings for 5 binary traits (BIP, SCZ, MDD, ADHD, and ASD) and 2 continuous traits (NEU and IQ). In total, we

identified 134 regions (78 non-overlapping genomic segments) that showed concordant associations with multiple traits ($FDR < 0.05$; **Figure 8**). The number of significant segments identified in our analysis is proportional to the absolute value of genetic correlation between each trait pair (correlation $r_g=0.71$). In addition to shared genetic architecture among five psychiatric disorders we studied before, we identified 2 shared regions for BIP and NEU ($r_g=0.14$, $p=9.99e-7$), 7 regions for SCZ and NEU ($r_g=0.22$, $p=6.55e-18$), 23 regions for MDD and NEU ($r_g=0.78$, $p=6.38e-41$), 1 region for BIP and IQ ($r_g=-0.04$, $p=1.23e-1$), 14 regions for SCZ and IQ ($r_g=-0.23$, $p=4.36e-28$), 1 region for MDD and IQ ($r_g=-0.18$, $p=7.19e-8$), 6 regions for NEU and IQ ($r_g=-0.19$, $p=7.68e-20$), 9 regions for ADHD and IQ ($r_g=-0.39$, $p=3.24e-23$), and 5 regions for ASD and IQ ($r_g=0.23$, $p=1.47e-10$), indicating strong, pervasive genetic sharing across these neuropsychiatric traits. Here, r_g is the global genetic correlation estimated by LD score regression. These results showcased the effectiveness of LOGODetect on both binary and continuous traits.

Figure 8. LOGODetect identifies genome regions contributing to multiple neuropsychiatric traits. Heatmap shows the genetic correlation estimates (upper triangle) and the number of segments with local genetic correlation identified by LOGODetect (lower triangle) between seven neuropsychiatric traits; Barplot shows the observed scale heritability estimates and standard errors of seven traits.

The discussions above have been incorporated into the revised manuscript (page 5, line 118; page 6, line 155-156, line 164-186; page 19, line 558-571), **Supplementary Notes**, **Supplementary Table 6**, and **Supplementary Figure 12**. **Figure 3** and relevant information in the main text have been updated accordingly.

2B. I wonder if the replication analysis on page 8 could be complemented by a more direct approach as follows. The authors could apply LOGODetect to the UKBB summary data of SCZ and BIP, and then check how many of 33 identified SCZ-BIP segments (page 6) are replicated here. Compared with the authors' replication analysis (Table 1), this suggested approach seems more straightforward, and more consistent with current GWAS literature.

Response:

We acknowledge that the replication approach suggested by the reviewer would be direct and convincing. However, In the UKBB, GWAS of BIP and SCZ had very limited case counts ($n_{\text{case}}=1,064$ for BIP; $n_{\text{case}}=571$ for SCZ) despite the large total sample size ($n_{\text{total}}=366,540$ for BIP; $n_{\text{total}}=366,047$ for SCZ), which led to insufficient statistical power for direct replication. We calculated the effective sample size, n_{eff} , for each study using the formula $n_{\text{eff}} = \frac{4}{\frac{1}{n_{\text{case}}} + \frac{1}{n_{\text{ctrl}}}} = 4n_{\text{total}}S(1-S)$, where S is the within-sample prevalence. n_{eff} for BIP and SCZ were 49,367 and 99,863 respectively in discovery GWASs, and only 4,244 and 2,280 in the UKBB.

In addition, we assessed the impact on statistical power given reduced n_{eff} . For simplicity, we treated all traits as continuous and assumed the total sample size in GWAS to be $n = n_{\text{eff}}$. As illustrated in our response to comment 2A, this would not change the results of LOGODetect. Using GWAS summary statistics as input, we mimicked the down-sampled summary statistics analogous to the approach PUMAS⁵. Consider a linear model $Y = X\beta + \epsilon$, where Y is the normalized continuous trait, X is the normalized genotypes, β is the fixed effect vector and ϵ is the random error.

The z-score is given as $z = \frac{1}{\sqrt{n}}X^TY \approx \sqrt{n}V\beta + \frac{1}{\sqrt{n}}X^T\epsilon$, where V is the LD (correlation)

matrix and is approximated using empirical LD in a reference panel. Suppose a random subsample of the original cohort (denoted with subscript 0) with sample size n_0 follows the linear model $Y_0 = X_0\beta + \epsilon_0$, and the remaining samples (denoted with subscript 1) with sample size $n_1 = n - n_0$ follows the linear model $Y_1 = X_1\beta + \epsilon_1$.

Z-score of the subsample is given as $z_0 = \sqrt{n_0}V\beta + \frac{1}{\sqrt{n_0}}X_0^T\epsilon_0$ which can be

reformulated as $z_0 = \sqrt{\frac{n_0}{n}}z + \frac{n_1X_0^T\epsilon_0 - n_0X_1^T\epsilon_1}{n\sqrt{n_0}}$. The term $\frac{n_1X_0^T\epsilon_0 - n_0X_1^T\epsilon_1}{n\sqrt{n_0}}$ follows normal distribution, the mean is 0 and the variance is given as:

$$\begin{aligned} & \frac{n_1^2}{n^2n_0} \text{var}[X_0^T\epsilon_0] + \frac{n_0}{n^2} \text{var}[X_1^T\epsilon_1] \\ &= \frac{n_1^2}{n^2n_0} * \frac{n_0}{n} \text{var}[X^T\epsilon] + \frac{n_0}{n^2} * \frac{n_1}{n} \text{var}[X^T\epsilon] \\ &= \frac{n_1}{n^2} \text{var}[X^T\epsilon] \\ &\approx \frac{n_1}{n^2} * nV \end{aligned}$$

$$= \frac{n - n_0}{n} V$$

Therefore, for a given $n_0 \leq n$, we can down-sample summary statistics from the original summary statistics as: $z_0 = \sqrt{\frac{n_0}{n}} z + N\left(0, \frac{n-n_0}{n} V\right)$.

Using this approach, we decreased the n_{eff} of SCZ and BIP GWASs to that in the UKBB replication cohort. As expected, the number of detected regions declined as n_{eff} decreased (**Figure 9**). We do not expect to see any significant replication given the strength of local genetic correlation in input GWASs and sample size in the UKBB. Therefore, directly applying LOGODetect to the UKBB for replication would be statistically underpowered.

Figure 9. Statistical power decreases as effective sample size decreases. Here the X-axis denotes the squared root of the effective sample size of the simulated BIP cohort times that of the simulated SCZ cohort. The Y-axis denotes the number of regions detected by LOGODetect. The box on the right corresponds to effective sample size in the BIP-SCZ discovery cohorts ($\sqrt{ESS_1 * ESS_2} = 70,214$), the box on the left corresponds to effective sample size in the BIP-SCZ UKBB replication cohorts ($\sqrt{ESS_1 * ESS_2} = 3,111$). We repeated 100 simulations for each effective sample size.

Therefore, we chose to assess the enrichment of aggregated genetic covariance across all identified local genomic segments to replicate our findings. Here, we provide more empirical justifications to this approach. We repeated the down-sampling analysis described above but assessed the enrichment of genetic covariance in all identified regions in the replication dataset instead of statistical significance of each segment. We found that the detected regions are still expected to be substantially enriched for genetic covariance even when n_{eff} decreased to the level of UKBB replication cohort (**Figure 10A**), whereas randomly selected regions

show no enrichment for genetic covariance when n_{eff} varies (**Figure 10B**).

Figure 10. Enrichment of aggregated genetic covariance with varying effective sample size. (A) Regions identified by LOGODetect in discovery cohort show substantial genetic covariance enrichment in UKBB replication cohort. (B) Randomly selected regions with same sizes as (A) show no genetic covariance enrichment in UKBB replication cohort. Genetic covariance of randomly selected regions may have opposite sign compared to global genetic covariance, therefore the corresponding genetic covariance enrichment may be negative. Here the X-axis denotes the squared root of the effective sample size of the simulated BIP cohort times that of the simulated SCZ cohort. For each panel, the box on the right corresponds to effective sample size in the BIP-SCZ discovery cohorts ($\sqrt{ESS_1 * ESS_2} = 70,214$), the box on the left corresponds to effective sample size in the BIP-SCZ UKBB replication cohorts ($\sqrt{ESS_1 * ESS_2} = 3,111$). We repeated the simulations for 100 times for each effective sample size.

We applied GNOVA to the replication GWASs to stratify genetic covariance in the regions identified by four methods (LOGODetect, ρ -HESS, coloc, and gwas-pw) (**Table 8**). The regions identified by LOGODetect and ρ -HESS both showed significant genetic covariance. Moreover, the regions identified by LOGODetect had a 5-fold higher enrichment of genetic covariance compared to that of ρ -HESS, which demonstrated again that LOGODetect can detect the true signal regions with improved precision. Regions identified by gwas-pw showed no significant genetic covariance, while regions identified by coloc showed genetic covariance with the opposite sign.

Table 8. Stratified genetic covariance analysis on UKBB replication cohorts

	Genetic Cov*	s.e.	p-value	Proportion of genetic cov	Proportion of SNPs	Fold enrichment
LOGODetect	7.58e-3	3.23e-5	1.89e-2	3.50%	0.56%	6.29
ρ -HESS	6.62e-4	3.16e-4	3.61e-2	30.00%	20.12%	1.49
coloc	-5.70e-5	2.30e-5	1.33e-2	-2.85%**	0.34%	-8.36**
gwas-pw	3.84e-5	6.54e-5	5.57e-1	1.92%	1.61%	1.20

* Genetic Cov represents estimated genetic covariance of the identified regions using GNOVA.

** Genetic covariance of regions identified by coloc has opposite sign compared to global genetic covariance, therefore the corresponding proportion of genetic covariance and fold enrichment are negative.

Taken together, these additional simulation results demonstrate the feasibility to replicate our findings by evaluating the aggregated genetic covariance over all identified local regions, as what we did in the manuscript of original submission. The discussions above have been incorporated into the revised manuscript (page 9, line 251-253), **Supplementary Notes**, and **Supplementary Figures 62-63. Table 1** and relevant information in the main text have been updated accordingly.

2C. Enrichments shown in Fig. 5A are not necessarily "tissue-specific", because they could be potentially driven by generic annotations such as intron and promoter regions. This is not accounted for in the current analysis (pages 19-20). At minimum, the authors should compare enrichments of generic annotations with the enrichments shown in Fig. 5A. A better analysis is to compute tissue-specific enrichments conditioning on multiple generic annotations.

Response:

Thank you for the comment. First, as suggested by the reviewer, we repeated the analysis after conditioning on six generic annotations including coding regions, enhancers, introns, promoters, 5'UTRs, and 3'UTRs (extended by a 500-bp window around each of the annotations) in Finucane et al⁶. More specifically, we removed the overlapped regions between each generic annotation and two brain tissue-specific annotations (cingulate gyrus and angular gyrus). We chose these two brain tissues since they showed the strongest enrichment for LOGODetect findings in our unconditioned analysis. We used permutation test to assess the statistical significance of enrichment in conditional analyses. The enrichment in cingulate gyrus (minimum enrichment=1.56, $p=3.80e-3$; **Figure 11**) and angular gyrus remained significant (minimum enrichment=1.56, $p=3.50e-3$; **Figure 11**) after conditioning on generic functional annotations, suggesting that the observed enrichment in functional genome in these brain tissues were not driven by generic annotations alone.

Figure 11. Tissue-specific enrichment of genome regions conditioning on six generic annotations. Enrichment in the predicted functional regions in cingulate gyrus and angular gyrus remains significant when conditioned on the generic annotations. Fold enrichment is labeled next to each bar.

In addition, we calculated the overlap between each of the 66 cell type annotations and six generic functional annotations. The two significantly enriched brain tissue types, i.e. cingulate gyrus and angular gyrus, did not show a higher proportion of overlapped regions with generic annotations compared to other 64 tissue and cell types (**Table 9**). Therefore, the fact that these two cell type annotations, but not other cell types, showed significant enrichment for regions identified by LOGODetect (illustrated in Fig 5A in the main text) was not driven by generic annotations. Taken

together, these results strengthened the claim that significantly correlated regions among neuropsychiatric disorders were enriched functional genome regions in brain tissues.

Table 9. Overlaps between two brain cell type annotations and six generic annotations. Each row represents the proportion of cell type specific annotation overlapped with six generic annotations. Rank of proportion in 66 cell types is provided in the parenthesis.

Cell_type	Coding	Enhancer	Intron	Promoter	UTR_3	UTR_5
Brain cingulate gyrus	0.29 (20/66)	0.46 (52/66)	0.72 (6/66)	0.35 (37/66)	0.09 (24/66)	0.2 (34/66)
Brain angular gyrus	0.29 (20/66)	0.47 (48/66)	0.73 (3/66)	0.37 (26/66)	0.09 (24/66)	0.2 (34/66)

The discussions above have been incorporated into the revised manuscript (page 10, line 281-287; page 23, line 689-694) and **Supplementary Table 16**.

2D. Can the hub regions shown in Fig. 6 be identified by previous methods such as rho-HESS (Ref. 12) or colocalization methods like Ref. 17 and 77? If yes, then what is the additional gain of using LOGODetect here? If no, then the authors should highlight this. Following the same logic, can genes highlighted on pages 12-13 be identified by previous methods? (This comment is related to Major Comment 1B).

Response:

Thank you for the comment. Here, we walk through the four hub regions we identified in our updated analysis of 7 complex traits.

1. The first hub region (**Figure 12A**; chr11:112,742,000 – 113,459,000) was identified for 7 trait pairs (BIP-SCZ, BIP-NEU, SCZ-MDD, SCZ-NEU, SCZ-IQ, MDD-NEU, and NEU-IQ) by LOGODetect, 3 trait pairs (BIP-SCZ, SCZ-NEU, and SCZ-IQ) by ρ -HESS, 0 trait pair by coloc, and 1 trait pair (SCZ-NEU) by gwas-pw. Here regions identified by LOGODetect are defined as those with significant scan statistic absolute value in our scanning procedure, regions identified by ρ -HESS are defined as those pre-specified regions with significant local genetic covariance, and regions identified by colocalization methods are defined as the pre-specified regions whose posterior probabilities containing shared association signal are larger than 0.95.

2. The second hub region (**Figure 12B**; chr14:103,783,000 – 104,364,000) was identified for 5 trait pairs (BIP-SCZ, SCZ-MDD, SCZ-IQ, MDD-IQ, and ASD-IQ) by LOGODetect, 1 trait pair (BIP-SCZ) by ρ -HESS, and 1 trait pair (MDD-IQ) by both coloc and gwas-pw.

3. The third hub region (**Figure 12C**; chr5:103,577,000 – 104,095,000) was identified for 4 trait pairs (BIP-MDD, MDD-ADHD, MDD-ASD, and ADHD-ASD) by LOGODetect, 2 trait pairs (BIP-SCZ and SCZ-MDD) by ρ -HESS, 0 trait pair by coloc, and 5 trait pairs (BIP-MDD, MDD-ADHD, MDD-ASD, MDD-NEU, and NEU-ADHD) by gwas-pw.

4. The fourth hub region (**Figure 12D**; chr10:106,385,000 – 106,835,000) was identified for 4 trait pairs (SCZ-ADHD, MDD-NEU, MDD-ADHD, and ADHD-ASD) by LOGODetect, 3 trait pairs (BIP-SCZ, SCZ-MDD, and SCZ-IQ) by ρ -HESS, 0 trait pair by coloc, and 1 trait pair (SCZ-ADHD) by gwas-pw.

Figure 12. Four hub regions shared by four or more neuropsychiatric traits. (A) shows the significant region on chr11 shared by seven trait pairs, (B) shows the significant region on chr14 shared by five trait pairs, (C-D) show two significant regions, each is shared by four trait pairs. Locuszoom plot, recombination rate, and the gene names are provided. The color band denote the location of the significant region and the corresponding trait pair. TAD in dorsolateral prefrontal cortex (DLPFC) in adult brain and TAD in the germinal zone (GZ) and postmitoticzone cortical plate (CP) in the developing fetal brain are shown as long solid line in panel (C).

Taken together, LOGODetect consistently identified these hub regions in more trait pairs compared to other methods. The only one exception was the third hub region, where gwas-pw identified the region to be shared by 5 trait pairs while our results implicated 4 pairs. However, gwas-pw was not effective in the other three hub regions. We also note that there is substantial biological evidence to support the critical role of these hub regions in neuropsychiatric disorders. For example, the first hub region contains multiple variants associated with BIP, SCZ, MDD and NEU⁷⁻⁹.

In addition, several genes associated with the neuropsychiatric traits are identified by LOGODetect but failed to be found using other methods. One example is *NCAM1*, which is associated with MDD⁹. Expression of *PSA-NCAM* is increased in antidepressant treatment, while in animal models of depression or in depressed patients *PSA-NCAM* is reduced⁹. Specially, *NCAM1* falls into the first hub-region for MDD-NEU identified by LOGODetect, but it cannot be identified by other three methods. Another example is *SORCS3*, which is associated with ASD. It is known that *SORCS3* plays an important role in protein networks associated with *PICK1*, *NGF*, and *PDGF-BB*^{10,11}, which have been implicated in ASD¹². Specially, *SORCS3* falls into

the fourth hub-region for ADHD-ASD identified by LOGODetect, but it cannot be identified by other three methods. These results suggest that LOGODetect produce novel biologically-relevant findings that could be missed by other approaches.

The discussions above have been incorporated into the revised manuscript (page 12, line 320-356; page 14, line 395-397; page 15, line 415-417). **Figure 6** and the relevant information in the main text have been updated accordingly.

2E. All analyses here were based on setting $\theta=0.5$ (pages 5 & 15). I understand this value was chosen based on simulation performance. However, the simulated data and real GWAS data are different in many aspects. If the authors cannot provide a more principled way to estimate θ from real data, then at least they should analyze the real GWAS data with different values of θ , and check whether the results are robust to θ .

Response:

Thank you for the suggestions. Scan statistics approaches have been developed and applied to single-trait analysis to identify genetic regions enriched for GWAS signals^{13,14}. In those applications, it is known that parameters conceptually similar to θ in our method control the size of detected signal regions. Here, we demonstrate that users of LOGODetect may choose θ in real applications based on the size of desired signal regions shared between traits.

For this purpose, we conducted additional simulations that are more comparable to real GWAS data. We used imputed genotype data on chromosome 22 from the Wellcome Trust Case Control Consortium (WTCCC) cohort to perform simulations. Genotypes were imputed to the 1000 Genomes Project European reference panel. We then simulated phenotypes under a heritability enrichment model so that 30% trait heritability was assigned to signal regions and 70% trait heritability was assigned to other SNPs for each trait. We fixed the effect size correlation in signal regions to be 0.9. We varied the trait heritability from 0.9 to 0.01, and varied the size of the signal regions from 1000 to 50 (SNPs). We applied LOGODetect with different θ values (i.e., 0.4, 0.5, and 0.6) to analyze simulated data. We observed a clear correlation between θ and the size of the detected regions – LOGODetect tends to identify larger regions with smaller θ (**Figure 13**). Applied to seven neuropsychiatric traits, we observed a similar negative correlation between detected region size and θ (**Figure 14**). This is consistent with what is known in single-trait applications¹⁴. In our analyses, under $\theta=0.5$, LOGODetect identified regions with a median size of about 300KB (**Figure 14**). Under θ values of 0.4 and 0.6, the median region sizes were about 600KB and 150KB (**Figure 14**). Our implemented software allows customized choice of θ based on their preference on signal region size.

Figure 13. Regions detected by LOGODetect with different tuning parameter θ in simulations. (A) Boxplot of number of SNPs in detected regions with different θ in simulations. (B) Boxplot of sizes of detected regions with different θ in simulations. Here data are simulated under alternative model with trait heritability varying from 0.9 to 0.01 and size of signal regions varying from 1000 to 50 SNPs. Correlation of genetic effect size of two traits is fixed to be 0.9. The heritability for the signal regions is set to be 30% trait heritability. Each model setting is repeated for 50 times.

Figure 14. Regions detected by LOGODetect with different tuning parameter θ in application to seven neuropsychiatric traits. (A) Boxplot of number of SNPs in detected regions in 7 neuropsychiatric traits. (B) Boxplot of sizes of detected regions in 7 neuropsychiatric traits.

In addition, following the reviewer's suggestion, we repeated our analysis of BIP and SCZ GWAS with different values of θ . LOGODetect identified 23 shared genomic regions under $\theta=0.4$, 33 shared regions under $\theta=0.5$, and 63 shared regions under $\theta=0.6$. All the identified regions under $\theta=0.4$ overlapped with regions identified under $\theta=0.5$ or 0.6. Similarly, all the identified regions under $\theta=0.5$ overlapped with findings

under $\theta=0.6$. Notably, five regions identified with $\theta=0.5$ were broken down to 13 smaller regions identified when using a larger $\theta=0.6$. However, regardless the choice of θ , the users would reach similar conclusions since LOGODetect based on different θ values still identified a consistent set of genomic regions. To further illustrate this, we ranked the regions identified by LOGODetect with different θ , ρ -HESS, coloc, and gwas-pw, by the corresponding p-values and posterior probabilities. We then evaluated the proportion of genetic covariance by SNPs in the top regions at various thresholds. Although there is some minor variability among LOGODetect analyses based on different θ , LOGODetect always outperformed other methods, explaining more genetic covariance with the same proportion of SNPs (**Figure 15**).

Figure 15. Genetic covariance explained by the same number of SNPs in regions identified by LOGODetect, ρ -HESS, coloc, and gwas-pw.

Taken together, the users of LOGODetect can choose θ based on the expected (or wanted) size of signal regions. And the performance of LOGODetect is robust to different choices of θ . The discussions above have been incorporated into the revised manuscript (page 5, line 137-138; page 8, line 227-228; page 13, line 358-375), and **Supplementary Figures 6** and **60**. **Figure 4** and the relevant information in the main text have been updated accordingly.

Other Comments

1. On page 4, for the same SNP i , are z_{1i} and z_{2i} based on the same allele coding scheme (i.e. do they have the same effect allele)? Please clarify.

Response:

Thank you for pointing this out. For any trait pair, we only included SNPs present in both GWASs in our analysis. For SNPs whose effect alleles were the same in the two GWASs, the original z-scores were used. For SNPs whose effect alleles were reversed in two GWASs, we reversed the sign of z-score in the second GWAS accordingly. Thus, the allele coding schemes between any two studies were consistent. The discussions above have been incorporated into the revised manuscript (page 23, line 668-671).

2. On page 4 or Fig. 1, please elaborate on how LOGODetect "scans the genome while allowing the segment size to vary". This seems to be a computationally expensive step, but I cannot find enough information on how the authors deal with it in the Methods section (page 15).

Response:

Thanks for the comments.

In our analysis, we use the maximal scan statistic $Q_{max} = \max_{|R| \leq C} |Q(R)|$ to control the family-wise type I error rate, where C is a pre-specified parameter that defines the upper boundary of the number of SNPs in a region R . In real GWAS analysis, we set C to be 2000, which covers genome regions of approximate sizes of 1.5MB on average. Calculating $Q(R)$ over all possible candidate regions is indeed computationally expensive, so we constrain $|R|$ to be a multiple of 10 in practice, which reduced the computation burden by approximately 10 folds, with minimal reduction in accuracy.

The discussions above have been incorporated into the revised manuscript (page 17, line 517-520). The computation details are provided in the revised manuscript (page 21, lines 607-611).

3. On page 7 the authors claimed that "the exact sample overlap across studies is unknown". To support this claim, the authors should review the supplementary information of Ref.s 33-37 and summarize what they find here. (Usually a GWAS publication will have a supplementary table listing cohort information, which can be used to assess sample overlap.) If the authors happen to be able to approximate the sample overlap from the supplementary information, they should report a version of results by accounting for the (approximate) sample overlap.

Response:

Thank you for the comments.

As suggested, we looked into the supplementary information of original publications of the GWAS datasets to identify the corresponding cohorts. Since all the GWASs included in our study were large meta-analyses of numerous cohorts, it was challenging to accurately identify the shared sample size. In **Table 10**, we summarized the approximated sample overlap based on the cohort information detailed in the supplementary material of the GWAS references. We used these as inputs for additional ρ -HESS analysis in the revision. Results for ρ -HESS did not substantially change after using these approximated values as shared sample sizes (**Table 11**).

Table 10. Approximated sample overlaps for different GWAS datasets.

	BP	SCZ	MDD	NEU	ADHD	ASD	Q
BP		40809	27933	0	1777	0	0
SCZ			29221	0	1413	0	0
MDD				29740	38247	36470	29740
NEU					0	0	195653
ADHD						35740	0
ASD							0
Q							

Table 11. Number of detected regions using ρ -HESS.

Trait 1	Trait 2	ρ -HESS (full sample overlap)	ρ -HESS (approximate sample overlap)
BP	SCZ	778	967
BP	MDD	0	0
BP	NEU	0	0
BP	ADHD	0	0
BP	ASD	0	0
BP	IQ	1	1
SCZ	MDD	93	36
SCZ	NEU	131	118
SCZ	ADHD	8	10
SCZ	ASD	0	20
SCZ	IQ	304	237
MDD	NEU	3	3
MDD	ADHD	0	0
MDD	ASD	0	0
MDD	IQ	0	0
NEU	ADHD	0	0
NEU	ASD	0	0
NEU	IQ	2	2
ADHD	ASD	0	0
ADHD	IQ	1	1
ASD	IQ	0	0

The discussions above have been incorporated into the revised manuscript (page 7, line 205-208; page 8, line 228-229) and **Supplementary Tables 13-14**.

4. When comparing rho-HESS to LOGODetect on real GWAS data, the authors used "pre-specify regions" in rho-HESS. How are these regions defined? What is the size of "pre-specify regions"? Can the authors try different sizes of "pre-specify regions" in rho-HESS analysis, using the inferred segment size from LOGODetect (24 Kb - 1.6 Mb, page 6). Is it possible that the rho-HESS results in Fig. 4 and Table 1 could be improved by using "better" defined regions?

Response:

Thank you for the comment. We used 1,703 approximately LD-independent regions in the genome (1.6 Mb in width on average) as the pre-specified genomic regions in our ρ -HESS and gwas-pw applications. This set of regions was inferred using ldetect¹⁵ and was recommended by the original papers of both ρ -HESS and gwas-pw². The paper of coloc did not give any suggestion on pre-specifying regions, therefore we used the same set of regions as ρ -HESS and gwas-pw.

The statistical framework underlying ρ -HESS was developed under a strong assumption that genotypes in different pre-specified genomic regions are independent. In fact, the authors of ρ -HESS suggested in their paper that, "In order to obtain unbiased estimates, the partitions should be as LD-independent as possible. Using a small window may result in LD leakage and biased estimates". The colocalization methods also require the same assumption, so that the likelihood function can be partitioned into the products of likelihood within each independent LD block. Therefore, using regions with much smaller sizes will violate this assumption and may lead to unreliable results.

In addition to the theoretical concerns, there are also practical difficulties in applying ρ -HESS to pre-specified regions with smaller sizes. Variance of local genetic covariance estimate frequently falls negative (may be due to insufficient number of SNPs in one region), which prohibits the subsequent inference. This also suggests that ρ -HESS cannot be flexibly applied to small regions.

The discussions above have been incorporated into the revised manuscript (page 7, line 198-200).

5. The number of cases in UKBB BIP and SCZ datasets (page 8) are quite small compared with published BIP and SCZ GWAS (Ref.s 33-34). Is LOGODetect robust to case-control ratios? This is not clear from the mathematical formulas, and it is not assessed by simulations.

Response:

In our response to comment 2B, we have detailed additional analyses and discussed the replication performance. Here, in response to this comment, we further investigate if LOGODetect is robust to unbalanced case-control ratios, or to say, extreme sample prevalence.

We simulated a binary trait under the liability threshold model, where the disease prevalence in population is fixed at 1/6. In specific, to simulate the disease status in population, we first simulated the continuous liability following the infinitesimal model, then assigned the samples whose liability is larger than 16.7% upper-quantile as cases and the rest as controls. We randomly collected cases and controls within the population into the cohort, with effective sample size fixed at 2,500 and sample prevalence varying from 0.1 to 0.5. Similar to previous simulations, we randomly selected $N = 5$ segments, each containing $L = 100$ SNPs, as the signal regions shared between two traits. We attributed $p = 0.3$ trait heritability to the signal regions. The effect size correlation of shared causal SNPs, ρ , is set to be 0.9. The trait heritability is set to be 0.2. We compared the statistical power of LOGODetect with other methods. LOGODetect achieved greatest signal points detection rates when sample prevalence is low and performed second best when sample prevalence is high (**Figure 16A**). In addition, LOGODetect obtained universally higher signal segments detection rates and G-scores (**Figure 16B-C**). Overall, LOGODetect showed robust performance under varying sample prevalence.

Figure 16. Assessment of statistical power under a liability threshold model with varying sample prevalence and fixed effective sample size. X-axis represents the sample prevalence (the proportion of cases in the cohort). Here we assume the liability threshold model for the binary trait, which means the disease status is determined by an unobserved continuous liability.

The discussions above have been incorporated into the revised manuscript (page 6, line 155-156) and **Supplementary Figure 13**.

6. To assess the "functional relevance" of identified local genetic correlations, I wonder if the authors could also run a standard Gene Ontology analysis, in addition to the tissue-specific annotation analysis (page 9).

Response:

Thank you for the advice. In the revision, we used FUMA¹⁶ to run the Gene Ontology enrichment analysis. Using GENCODE V33lift37 on the UCSC genome browser, we extracted 468 genes in the genomic regions found to harbor local genetic correlations among seven neuropsychiatric traits. Among these genes, 438 genes have recognized Ensembl ID and were considered as prioritized genes. These prioritized genes were significantly enriched in 28 GO terms (**Figure 17**) after multiple testing correction, including neuron differentiation (enrichment=3.70, $p=1.22e-7$), sialic acid transport (enrichment=65.34, $p=9.69e-7$), and neurogenesis (enrichment=3.28, $p=1.20e-6$).

Figure 17. Prioritized genes show significant enrichment in three gene ontology domains: (A) GO biological process, (B) GO cellular component, (C) GO molecular function.

The discussions above have been incorporated into the revised manuscript (page 10, line 289-293; page 23, line 696-699) and **Supplementary Figure 59**.

7. *On page 12, "hi-C" should be "Hi-C" instead.*

Response:

Thank you. We have corrected it in the revised manuscript.

8. *The discussion section (pages 12-13) does not summarize the limitations of LOGODetect. Any manuscript describing a new method must have such discussions. An obvious limitation of LOGODetect compared to rho-HESS (Ref. 12) and LDSC (Ref. 10) is that LOGODetect only tests the significance of genetic correlation, but cannot estimate it (please see Fig. 3 of Ref. 12).*

Response:

We appreciate this comment. In our revised manuscript, we have added discussions about the limitations and future directions of LOGODetect. Here, we summarize what has been discussed.

First, as noted by the reviewer, the goal of LOGODetect is to identify genomic regions harboring local genetic correlations. We do not give explicit estimation of local genetic correlation, but the sign of the correlation can be inferred. Although local genetic correlation in identified regions can be estimated by other methods (e.g., ρ -HESS) in principle, this remains a statistically challenging problem. As shown in our simulations, the estimation is inaccurate (**Tables 1, 4-7**). Under the null that local genetic correlation is zero, ρ -HESS underestimates the standard error of local genetic covariance when heritability is high and leads to inflated type I error rates, while overestimates the standard error of local genetic covariance when heritability is low and leads to deflated type I error rates. These problems are further exacerbated when ρ -HESS is applied to very small local genomic regions identified by LOGODetect.

Second, LOGODetect scans a large number of genomic regions to search for local regions where the scan statistic significantly deviates from the null distribution. We currently do not have an analytical solution to derive or approximate the theoretical null distribution. Instead, a Monte Carlo approach is employed to quantify the null distribution of the maximal scan statistic, which is computationally expensive.

Finally, LOGODetect is an effective approach to identifying shared genetic components between a pair of traits. Several recent methods have been proposed to jointly model more than two GWAS traits to infer the structure of shared genetics across multiple phenotypes¹⁷⁻¹⁹. A future direction is to generalize our method to search for genomic regions shared by more than two traits.

The discussions above have been incorporated into the revised manuscript (page 15, line 419-435).

9. *On page 14, I do not understand why the authors say that their method "does not assume the global genetic covariance to be the same across all the SNPs in the whole genome". To me the "global genetic covariance" is just a single number measuring how genetically correlated two traits are. Why should this single number be varying across the genome?*

Response:

We apologize for the confusion. We meant to state that our method does not assume "per-SNP heritability" to be equal across the genome. We have corrected the inaccurate expression to "does not assume the global genetic covariance to be equally attributed to all SNPs in the genome" in the revised manuscript (page 16, line 468).

10. On page 15, the claim that "the expected value of [sum] is larger in regions with strong LD" is non-trivial. Either mathematical derivations or numerical simulations should be shown here.

Response:

Thank you for the suggestion. We use a toy example to illustrate this. Consider two independent GWAS with K causal SNPs: $y_1 = \sum_{i=1}^K X_i^T \beta_i + \epsilon$, $y_2 = \sum_{i=1}^K Z_i^T \gamma_i + \delta$, where y_1, y_2 are standardized trait vectors with n_1, n_2 samples respectively, X_i, Z_i are standardized genotypes of i -th SNP, ϵ and δ are independent environment effect.

Suppose $\begin{pmatrix} \beta \\ \gamma \end{pmatrix} \sim N\left(\begin{bmatrix} 0 \\ 0 \end{bmatrix}, \begin{bmatrix} \frac{0.1}{K} I_K & \frac{0.05}{K} I_K \\ \frac{0.05}{K} I_K & \frac{0.1}{K} I_K \end{bmatrix}\right)$, and suppose the first $K-1$ SNPs are

in perfect LD, and are in perfect LE (linkage equilibrium) with the K -th SNP, which means $X_1 = X_2 = \dots = X_{K-1}$, $Z_1 = Z_2 = \dots = Z_{K-1}$, and $X_1^T X_K = Z_1^T Z_K = 0$. Suppose the environment effect is independent with the genotype. We ignore the distinction between independence in population level and in sample level, which means $X_i^T \epsilon = 0$, $Z_i^T \delta = 0$, $\epsilon^T \delta = 0$. Note that $z_{1i} = \frac{1}{\sqrt{n_1}} X_i^T y_1$, $z_{2i} = \frac{1}{\sqrt{n_2}} Z_i^T y_2$, it can be shown that,

$$\begin{aligned} & \left| \sum_{i=1}^{K-1} z_{1i} z_{2i} \right| = (K-1) |z_{11} z_{21}| \\ &= \frac{K-1}{\sqrt{n_1 n_2}} \left| X_1^T \left(X_1 \sum_{i=1}^{K-1} \beta_i + X_K \beta_K + \epsilon \right) * Z_1^T \left(Z_1 \sum_{i=1}^{K-1} \gamma_i + Z_K \gamma_K + \delta \right) \right| \\ &= \frac{K-1}{\sqrt{n_1 n_2}} \left| n_1 \sum_{i=1}^{K-1} \beta_i * n_2 \sum_{i=1}^{K-1} \gamma_i \right| \\ &= (K-1) \sqrt{n_1 n_2} \left| \sum_{i=1}^{K-1} \beta_i \gamma_i \right|. \end{aligned}$$

Similarly, we have $\left| \sum_{i=K}^K z_{1i} z_{2i} \right| = \sqrt{n_1 n_2} |\beta_K \gamma_K|$. Therefore, inequality $E \left| \sum_{i=1}^{K-1} z_{1i} z_{2i} \right| \geq E \left| \sum_{i=K}^K z_{1i} z_{2i} \right|$ holds if $E \left| \sum_{i=1}^{K-1} \beta_i \gamma_i \right| \geq E |\beta_K \gamma_K|$. To prove the latter inequality, one only needs to show that if $\xi_1, \xi_2 \dots \xi_K$ are independently and identically distributed random variables, then $E \left| \sum_{i=1}^{K-1} \xi_i \right| \geq E |\xi_K|$. Suppose the cumulative distribution function of ξ_i

is denoted by F , then $E |\xi_1 + \xi_2| - E |\xi_1 - \xi_2| = \int_0^\infty [1 - F(u) - F(-u)]^2 du \geq 0$. Thus

$E |\xi_1 + \xi_2| \geq \frac{1}{2} [E |\xi_1 + \xi_2| + E |\xi_1 - \xi_2|] \geq E |\xi_1| = E |\xi_K|$. Similarly, one can show that $E \left| \sum_{i=1}^{K-1} \xi_i \right| \geq E |\xi_K|$, and it follows that $E \left| \sum_{i=1}^{K-1} z_{1i} z_{2i} \right| \geq E \left| \sum_{i=K}^K z_{1i} z_{2i} \right|$. This indicates that the expected absolute value of $\sum_{i \in R} z_{1i} z_{2i}$ is larger in regions with strong LD.

The discussions above have been incorporated into **Supplementary Notes**.

11. The citation for WTCCC data seems missing.

Response:

Thanks for pointing out the mistake. We have updated the citation for WTCCC data and added acknowledgements to the WTCCC resource in the revised manuscript (page 25, line 719-721).

12. The authors did not add page or line numbers in the submitted main text file, which makes the review process less convenient.

Response:

Thanks for the suggestion. We have added page and line numbers in the revised manuscript.

Response to Reviewer #2:

1. The paper proposes a statistic to identify regions of the genome that contribute to two traits. Not surprisingly, the statistic is based on the cross product of z scores over SNPs within a region R . The denominator is an LD Score for R to correct over-weighting of high LD regions, the LD score being "tempered" by raising to power θ , and the authors decide to set $\theta = 0.5$. Calculation of the statistic for an R is cheap, and so can be done for many R in order to search for the best value: the authors consider all R with up to C SNPs for some C .

While the numerator of the statistic seems reasonable, there is little justification given for the precise form of the denominator: that it should be some increasing function of LD is clear, but the particular functional form seems arbitrary and the justification for the choice $\theta = 0.5$ is weak, there seems to be no formal criteria and in any case it is based on unrealistic simulations.

Response:

Thanks for your constructive comment.

We have proposed the scan statistic as follows, to search for regions harboring local genetic correlation:

$$Q(R) = \frac{\sum_{i \in R} z_{1i} z_{2i}}{(\sum_{i \in R} l_i)^\theta}.$$

As mentioned in the manuscript, this is a generalization of the scan statistic proposed in Jeng et al¹³ and Li et al¹⁴, which detects signal regions associated with a single trait. The scan procedure in Jeng et al was based on the mean of the marginal test statistics in a candidate region. We refer to this as the mean scan procedure. The scan statistic for a given region R is defined as $\frac{\sum_{i \in R} z_i}{\sqrt{|R|}}$, where $|R|$ denotes number of

SNPs in region R . The authors showed that the mean scan procedure is asymptotically optimal as long as SNPs are independent and the SNPs in the signal region have the same effect size. In our paper, we extend the mean scan framework to the context of detecting local genetic correlation. We replace $|R| = \sum_{i \in R} 1$ in the mean scan procedure, with $\sum_{i \in R} l_i$, to account for LD. Notably, under the case that all the SNPs have the same LD effect, e.g. l_i are the same for all SNPs, $\sum_{i \in R} l_i$ just reduces to $|R|$ up to a multiple constant. Following Li et al¹⁴, we relax the choice of power θ from 0.5 to a numeric in $[0,1]$ to ensure that the scan statistic is comparable for different region sizes.

Second, the scan statistics we proposed has good genetics interpretation. It has been shown that under the polygenic model that per-SNP genetic covariance is the same for all SNPs, and if two GWASs are independent, then the following equation holds³:

$$E[z_{1i} z_{2i}] = \frac{\sqrt{N_1 N_2} \rho_g}{M} l_i,$$

where the N_1, N_2 represents the sample size for two GWASs, M is the SNP count and ρ_g is the (global) genetic covariance. If we let θ be 1 and R be the whole genome, then the expectation of the scan statistic $Q(R)$ reduces to $\frac{\sqrt{N_1 N_2} \rho_g}{M}$, which is

exactly the (global) genetic covariance ρ_g multiplied by a constant. This implicates that when the candidate region is the whole genome, the scan statistic actually represents the strength of (global) genetic covariance. In our framework, we do not assume the polygenic model that per-SNP genetic covariance is the same for all SNPs across genome, but allow heterogeneity of per-SNP genetic covariance. Therefore, we use the scan statistic in a local region, as a metric to detect significant local genetic sharing.

In applications of scan statistics approaches to single-trait analysis¹¹, it is known that parameters conceptually similar to θ in our method control the size of detected signal regions. Here, we demonstrate that users of LOGODetect may choose θ in real applications based on the size of desired signal regions shared between traits. For this purpose, we conducted additional simulations that are more comparable to real GWAS data. We used imputed genotype data on chromosome 22 from the Wellcome Trust Case Control Consortium (WTCCC) cohort to perform simulations. Genotypes were imputed to the 1000 Genomes Project European reference panel. We then simulated phenotypes under a heritability enrichment model so that 30% trait heritability was assigned to signal regions and 70% trait heritability was assigned to other SNPs for each trait. We fixed the effect size correlation in signal regions to be 0.9. We varied the trait heritability from 0.9 to 0.01, and varied the size of the signal regions from 1000 to 50 (SNPs). We applied LOGODetect with different θ values (i.e., 0.4, 0.5, and 0.6) to analyze simulated data. We observed a clear correlation between θ and the size of the detected regions – LOGODetect tends to identify larger regions with smaller θ (**Figure 13**, response to Comment-2E of Reviewer #1). Applied to seven neuropsychiatric traits (In response to Comment-2A of Reviewer #1, we have added two continuous neurological traits, neuroticism and intelligence, abbreviated as NEU and IQ respectively, into our analysis), we observed a similar negative correlation between detected region size and θ (**Figure 14**, response to Comment-2E of Reviewer #1). This is consistent with what is known in single-trait applications¹⁴. In our analyses, under $\theta=0.5$, LOGODetect identified regions with a median size of about 300KB (**Figure 14**). Under θ values of 0.4 and 0.6, the median region sizes were about 600KB and 150KB (**Figure 14**). Our software allows customized choice of θ based on their preference on signal region size.

Figure 13 (Response to Comment-2E of Reviewer #1). Regions detected by LOGODetect with different tuning parameter θ in simulations. (A) Boxplot of number of SNPs in detected regions with different θ in simulations. (B) Boxplot of sizes of detected regions with different θ in simulations. Here data are simulated under alternative model with trait heritability varying from 0.9 to 0.01 and size of signal regions varying from 1000 to 50 SNPs. Correlation of genetic effect size of two traits is fixed to be 0.9. The heritability for the signal regions is set to be 30% trait heritability. Each model setting is repeated for 50 times.

Figure 14 (Response to Comment-2E of Reviewer #1). Regions detected by LOGODetect with different tuning parameter θ in application to seven neuropsychiatric traits. (A) Boxplot of number of SNPs in detected regions in 7 neuropsychiatric traits. (B) Boxplot of sizes of detected regions in 7 neuropsychiatric traits.

In addition, we repeated our analysis of BIP and SCZ GWAS with different values of θ . LOGODetect identified 23 shared genomic regions under $\theta=0.4$, 33 shared regions under $\theta=0.5$, and 63 shared regions under $\theta=0.6$. All the identified regions under

$\theta=0.4$ overlapped with regions identified under $\theta=0.5$ or 0.6 . Similarly, all the identified regions under $\theta=0.5$ overlapped with findings under $\theta=0.6$. Notably, five regions identified with $\theta=0.5$ were broken down to 13 smaller regions identified when using a larger $\theta=0.6$. However, regardless the choice of θ , the users would reach similar conclusions since LOGODetect based on different θ values still identified a consistent set of genomic regions. To further illustrate this, we ranked the regions identified by LOGODetect with different θ , ρ -HESS, by the corresponding p-values and posterior probabilities. In addition, in response to Comment 2, we added two colocalization methods, coloc¹ and gwas-pw², into the analysis. We then evaluated the proportion of genetic covariance by SNPs in the top regions at various thresholds. Although there is some minor variability among LOGODetect analyses based on different θ , LOGODetect always outperformed other methods, explaining more genetic covariance with the same proportion of SNPs (Figure 15, response to Comment-2E of Reviewer #1).

Figure 15 (Response to Comment-2E of Reviewer #1). Genetic covariance explained by the same number of SNPs in regions identified by LOGODetect, ρ -HESS, coloc, and gwas-pw.

Taken together, the users of LOGODetect can choose θ based on the expected (or wanted) size of signal regions. And the performance of LOGODetect is robust to different choices of θ . The discussions above have been incorporated into the revised manuscript (page 4, line 86-87, line 90-93; page 5, line 137-138; page 8, line 227-228; page 13, line 358-375), **Supplementary Notes**, and **Supplementary Figures 6 and 60**. **Figure 4** and the relevant information in the main text have been updated accordingly.

2. The authors report good performance of their method in a simulation study, relative to only one alternative, rho-Hess from Shi et al 2017. The simulation study is simplistic with only two effect sizes for SNPs. In particular the simulated effects were independent of LD and so failed to check the main feature of the statistic which is how it models the LD-effect size relationship.

Response:

Thank you for the suggestions. As suggested, we added two colocalization methods, coloc¹ and gwas-pw², into the simulations. Both coloc and gwas-pw are Bayesian approaches that calculate the posterior probability of colocalization as output. By default, we used 0.95 as the cutoff of posterior probabilities.

First, we evaluated performance of the colocalization approaches under the heritability enrichment model as in the simulations of our original submissions. The heritability enrichment model assumed that 30% trait heritability was attributed to 3% randomly selected SNPs and 70% trait heritability was attributed to the remaining SNPs. **Table 2** (Response to Comment-1B of Reviewer #1) showed the empirical type I error rates for coloc and gwas-pw under the heritability model. The coloc approach (type I error rate > 0.1 when heritability \geq 0.4) showed inflated type I error rate. In response to Comment-1A of Reviewer #1, we adjusted the significance cutoff of different methods to achieve the same type I error accordingly in the power simulations, such that a valid statistical power comparison can be performed. In this heritability enrichment model, we adjusted the critical value of ρ -HESS and coloc to maintain the empirical type I error rate at 0.05 for simulation settings in which these methods showed inflated type I error rates.

Table 2 (Response to Comment-1B of Reviewer #1). Empirical type I error rates for colocalization methods under a heritability enrichment model. Simulation settings with type I error rate larger than 0.1 are highlighted in the red box.

Empirical type I error rate with posterior probability cutoff being 0.95		coloc	gwas-pw
Trait heritability	0.9	0.589	0.034
	0.8	0.514	0.025
	0.7	0.473	0.021
	0.6	0.389	0.029
	0.5	0.277	0.032
	0.4	0.182	0.038
	0.3	0.077	0.037
	0.2	0.01	0.076
	0.1	0.004	0.022
	0.05	0	0.001
	0.03	0	0.001
	0.01	0	0

To perform power analysis, we simulated two phenotypes under the alternative model that two traits shared genetic correlation. We randomly selected $N = 5$ segments,

each containing $L = 100$ SNPs, as the signal regions shared between two traits. We attributed $p = 0.3$ trait heritability to the signal regions. The effect size correlation of shared signal SNPs, ρ , is set to be 0.9. Under this setting, we compared the performance of our method with that of ρ -HESS, coloc, and gwas-pw after type I error adjustment. As shown in **Figure 1** (Response to Comment-1A of Reviewer #1), when heritability increases, LOGODetect showed improvements in all three measures of statistical power without inflating type I error. Regarding to signal points detection rates, when heritability is low to moderate, gwas-pw performed best. LOGODetect achieved greater points detection rates compared to ρ -HESS and coloc. When heritability is high, ρ -HESS achieved the best points detection rates, and LOGODetect was the second best. Notably, LOGODetect performs reasonably well in all heritability settings, while the performance of gwas-pw and ρ -HESS is strongly affected by different heritability (**Figure 1A**). Regarding to segment detection rate and G-score, LOGODetect outperformed the other three methods at various heritability settings (**Figures 1B-C**). Therefore, we conclude that LOGODetect robustly controls for type I error and retains statistical power with heritability enrichment model.

Figure 1 (Response to Comment-1A of Reviewer #1). Assessment of statistical power under a heritability enrichment model with varying trait heritability. The X-axis shows the heritability for simulated traits. The Y-axis shows the statistical power assessed by three different measures: (A) signal points detection rate, (B) signal segments detection rate, and (C) G-score. Due to the observed inflation in type I error rates, we adjusted the significance cutoff for ρ -HESS, coloc, and gwas-pw so that their type I error rates achieve 0.05.

Further, although it is not uncommon for genetic correlation methods to assume SNP effects to be independent from LD^{3,4}, recent work on heritability estimation has explored LD-dependent genetic architecture and has shed important light on the genetic basis of complex traits. In our revised manuscript, we assessed the robustness of LOGODetect to LD-dependent genetic architecture. One well-known approach that considers LD-dependent SNP effect is the LDK²⁰ model, which assumes $h_j^2 \sim (f_j(1 - f_j))^\alpha * w_j$, where h_j^2 is the heritability attributed to the j -th SNP, f_j is the minor allele frequency (MAF) of the j -th SNP, and w_j denotes the LD-dependent weight value. Here, we set $\alpha = 0$ which assumes heritability to be independent with MAF which is consistent with the GCTA model²¹, and set $w_j \sim \frac{1}{l_j}$

where l_j is the LD-score of the j -th SNP, which assumes a lower per-SNP heritability for SNPs in strong LD with other variants.

To evaluate the type I error rates, we simulated two phenotypes and the corresponding summary stats using the LDAK framework under the null (i.e., genetic effects on two traits are independent). LOGODetect showed well-controlled type I error rates (**Table 12**), but ρ -HESS (type I error rate > 0.1 when heritability ≥ 0.5) and coloc (type I error rate > 0.1 when heritability ≥ 0.7) showed an clear inflation in its type I error (**Tables 12-13**).

Table 12. Type I error rates under a model with LD-dependent genetic effects. Type I error rates denote the proportion of simulations that significant segments harboring local genetic correlation are identified under the null. Simulation settings with type I error rate larger than 0.1 are highlighted in the red box.

Type I error rate at significance level $\alpha = 0.05$		LOGODetect			ρ -HESS
		$\theta=0.4$	$\theta=0.5$	$\theta=0.6$	
Trait heritability	0.9	0.001	0.001	0.006	0.999
	0.8	0	0.001	0.002	0.985
	0.7	0	0	0.005	0.909
	0.6	0	0.001	0.008	0.605
	0.5	0	0	0.007	0.255
	0.4	0	0	0.004	0.037
	0.3	0.001	0.002	0.016	0.004
	0.2	0.003	0.005	0.034	0.001
	0.1	0.014	0.021	0.057	0
	0.05	0.025	0.036	0.066	0
	0.03	0.027	0.048	0.064	0
	0.01	0.028	0.049	0.057	0

Table 13. Empirical type I error rates for colocalization methods with LD-dependent genetic effects. Simulation settings with type I error rate larger than 0.1 are highlighted in the red box.

Empirical type I error rate with posterior probability cutoff being 0.95		coloc	gwas-pw
Trait heritability	0.9	0.242	0.028
	0.8	0.192	0.016
	0.7	0.129	0.024
	0.6	0.077	0.026
	0.5	0.038	0.031
	0.4	0.011	0.039
	0.3	0.005	0.053
	0.2	0	0.018
	0.1	0.001	0.001
	0.05	0	0
	0.03	0	0
	0.01	0	0

To perform power analysis, we simulated two phenotypes using the LDAK framework under the alternative model (i.e. two traits shared local genetic correlation in $N = 5$ genomic segments). Regarding to signal points detection rates, when heritability is

low to moderate, gwas-pw performed best and LOGODetect achieved the second best. When heritability is high, ρ -HESS achieved the best points detection rates (**Figure 18A**). Regarding to segment detection rate and G-score, LOGODetect outperformed the other three methods at various heritability settings (**Figures 18B-C**). Therefore, we concluded that LOGODetect robustly controls for type I error and retains statistical power with LD-dependent model.

Figure 18. Assessment of statistical power under a model with LD-dependent genetic effects. (A-C) show statistical power assessed by three measures: Signal points detection rate, Signal segments detection rate and G-score. Heritability represents the trait heritability for both traits. For each trait, we randomly choose 5 segments, each contains 100 SNPs, as the signal regions. Here we assume the LD dependent genetic architecture, where heritability attributed to each SNP is inversely proportional to its LD score. The correlation of genetic effect size of two traits ρ is set to be 0.9. Each simulation setting is repeated for 100 times.

The discussions above have been incorporated into the revised manuscript (page 4, line 104-123; page 6, line 146-162), **Supplementary Notes, Supplementary Tables 2-3**, and **Supplementary Figure 14**. **Figure 2** and the relevant information in the main text have been updated accordingly.

3. In the analysis of real data (5 psychiatric traits), the new method and rho-Hess were compared based on "enrichment of genetic covariance", but this was not defined and there has been controversy over different approaches to enrichment. Moreover the enrichment was based on LD Score and the authors method is also based on LD Score, so it hardly seems a fair metric for comparison. In particular the finding of lower enrichment in regions detected by rho-HESS alone seems of little value.

Response:

We appreciate the comment. First, we provide a clear definition for genetic covariance enrichment. Suppose the genetic covariance between two traits explained by additive genetic effects is ρ_g , then the per-SNP genetic covariance is $\frac{\rho_g}{M}$ where M is the total number of SNPs in the analysis. For a given genomic region (or multiple regions which we consider as a SNP set) containing m SNPs, we denote the local genetic covariance explained by SNPs within this genomic region as $\rho_{g,local}$, and the per-SNP genetic covariance within this genomic region is $\frac{\rho_{g,local}}{m}$. We define the genetic covariance enrichment of this given region to be $\frac{\rho_{g,local}}{m} / \frac{\rho_g}{M}$. This term quantifies the ratio of per-SNP heritability within the given genomic region and that in the genome. Such a definition is conceptually similar to the heritability enrichment widely used in the field⁶ and can be quantified using tools such as GNOVA⁴. If a region shows strong enrichment for genetic covariance, it means that SNPs in this region have more contributions to the global genetic covariance compared to randomly selected regions in the genome.

In addition, to demonstrate that genetic covariance enrichment is a fair and effective metric for comparing different approaches, we performed two simulations (i.e. under the alternative and null) and evaluated the genetic covariance enrichment respectively. First, we simulated two genetically correlated traits and applied four methods (i.e., LOGODetect, ρ -HESS, gwas-pw, and coloc) to these two traits. We used 22 autosomes genotype data from the Wellcome Trust Case Control Consortium (WTCCC) cohort to perform simulations. We randomly selected $N = 100$ segments, each containing $L = 100$ SNPs, as the signal regions shared between two traits. We attributed $p = 0.3$ trait heritability to the signal regions. The effect size correlation of shared signal SNPs, ρ , is set to be 0.9. Heritability h^2 is set to be 0.5 for both traits. Under this setting, the genetic covariance enrichment is higher in the regions identified by LOGODetect than regions identified by the other three methods (**Figure 19 solid line**), which is concordant with the real data results between BIP and SCZ. To investigate whether the higher genetic covariance enrichment for the regions identified by LOGODetect are potential bias introduced by using LD scores in calculating the scan statistic, we performed null simulations. For each trait, we attributed 30% of the trait heritability to 10000 randomly chosen SNPs, while the remaining SNPs explain 70% of the trait heritability. We applied the four methods,

LOGODetect, ρ -HESS, coloc, and gwas-pw to these simulated independent traits, and the corresponding top regions are identified. Then we evaluated the proportion of genetic covariance explained by these top regions with respect to the correlated traits under the alternative, and we found no significant enrichment (**Figure 19 dashed line**). This demonstrates that LOGODetect does not favor regions with high LD scores and will not induce systematic bias in calculating genetic covariance enrichment. Thus, the observed enrichment of genetic covariance in real data analysis of LOGODetect regions is attributed to the shared genetic architecture instead of the systematic bias. We concluded that genetic covariance enrichment is a fair metric comparing performance of different methods, and LOGODetect actually identify more precise regions than other methods.

Figure 19. Genetic covariance enrichment curve in simulations. Two genetic correlated traits are simulated and the corresponding top regions are identified using four methods. Regions identified in the two correlated traits have significant genetic covariance enrichment with respect to the two correlated traits, as depicted in solid line. We simulate another two independent traits and obtain the corresponding top regions using four methods. Regions identified in the two independent traits have no genetic covariance enrichment with respect to the two correlated traits, as depicted in dashed line.

The discussions above have been incorporated into the revised manuscript (page 8, line 213-214), **Supplementary Notes**, and **Supplementary Figure 61**.

4. *Figure 2 seems to show a high detection rate when the heritability is close to zero, which needs discussion; it seems implausible, and may be a consequence of the unrealistic simulation model.*

Response:

Thank you for the comment. First, in our originally submitted manuscript, the minimum heritability in our various simulation settings was 0.05. Indeed, we observed a non-zero detection rate even when heritability was 0.05. However, we note that since we performed simulations using genotypes on chromosome 1, a heritability of 0.05 actually represents moderate genetic signals. In the revised manuscript, we have conducted additional simulations under a more extreme setting with heritability being 0.01. As expected, we observed a sharp decline of signal detection rate as the heritability was low (**Figure 1**, related to Comment 2). **Figure 2** and the relevant information in the main text have been updated accordingly.

5. Overall I think the new statistic is poorly justified and the evidence presented for its properties is weak.

Response:

Thank you and the other two reviewers for the constructive comments and suggestions. We agree that statistical rigor is absolutely critical in both simulations and real data analysis. In the revised manuscript, we have added more details on the theoretical justification of the LOGODetect framework, regarding the functional forms of the scan statistics, the choice of the parameter, and the application to binary traits. In addition, we have added extensive simulations to evaluate our method's performance under mis-specified models, including a non-infinitesimal model with fixed causal effects, LDAK model with LD-dependent effect sizes, and infinitesimal models with non-normal, heavy-tailed effect distributions. We have also added two colocalization approaches in the analysis to compare their performance with LOGODetect. In all simulations, LOGODetect has well-controlled type I error and superior statistical power compared to other methods. Simulation results under various settings are consistent with the conclusions we made in the original submission. Following other reviewers' suggestions, we have added two additional neuropsychiatric traits into our real data analysis and have identified more genomic regions showing significant local genetic correlations across traits, including multiple hub regions shared by multiple phenotypes. Moreover, we have provided more justifications on the enrichment of genetic covariance, the metric we used to showcase the performance of our method. Through simulations, we showed it is an appropriate and effective metric to evaluate and compare different methods. We hope these additional analyses and discussions could improve our manuscript and address your concerns.

Response to Reviewer #3:

Guo et al provide a well-written manuscript, with a flexible method to estimate the local correlation of SNP effects (or local genetic correlations) that does not depend on pre-defined annotations/regions and will likely be very useful for the human genetics community.

Suggestions:

1. This work compares LOGODetect performance with rho-HESS performance (using both simulations and real data analysis). While I think this is a useful comparison to make, rho-HESS defines “genetic correlation” differently from LOGODetect or cross-trait LDSC. Rho-HESS (along with quantitative genetic theory) defines genetic correlation to be the correlation between additive genetic values while LOGODetect and LDSC appear to define genetic correlation as the correlation of SNP effects.

How might differences in genetic correlation definitions impact the comparison of LOGODetect and rho-HESS? Can you discuss how differences in type I error or resolution may arise from differences in genetic correlation definitions?

Response:

Thank you for the comment. An essential factor underlying different definitions of genetic correlation is the genetic model. LOGODetect assumes a random-effect model, and genetic correlation is defined as the correlation of effect sizes of a SNP in two traits, while ρ -HESS is based on a fixed-effect model, thus defining genetic correlation as the correlation between the effect size vectors for a set of SNPs in two traits. Despite of the conceptual difference, the estimators have similar forms. In the revised manuscript, we have performed extensive simulations under both fixed-effect and random-effect models, and investigated if LOGODetect and ρ -HESS are robust to model mis-specification. Results for simulations under a random-effect model have been discussed in our original submission (Supplementary Tables 1-3). Here, we report simulation results under a fixed-effect model. For statistical analysis, in addition to ρ -HESS, we added two colocalization methods including coloc¹ and gwas-pw² into comparison. In addition, in response to Comment-1B of Reviewer #1, we added two colocalization methods, coloc¹ and gwas-pw², into the simulations. Both coloc and gwas-pw are Bayesian approaches that calculate the posterior probability of colocalization as output. By default, we used 0.95 as the cutoff of posterior probabilities.

First, we evaluated the four methods under non-infinitesimal models with sparse effects. We split up chromosome 1 into two parts (i.e., chr1:1-116,000,000 and chr1:116,000,001-249,143,646), each containing half of the SNPs. For the first trait, we randomly sampled 1000 variants from the first half of chr1 as causal variants. Similarly, we randomly sampled 1000 variants from the second half of chr1 as causal variants for the second trait. Per-SNP heritability for all the causal variants were the same. Notably, under this model, local genetic correlation is exactly equal to 0 according to genetic correlation definition in ρ -HESS. Results were comparable to simulations under infinitesimal models. Under this setting, type I error rate was

well-controlled for LOGODetect, coloc, and gwas-pw (**Table 5**, response to Comment-1C of Reviewer #1; **Table 14**), but ρ -HESS showed type I error inflation when heritability is high (e.g. ≥ 0.5).

Table 5 (Response to Comment-1C of Reviewer #1). Type I error rates under a non-infinitesimal model with sparse effects. Simulation settings with type I error rate larger than 0.1 are highlighted in the red box.

Type I error rate at significance level $\alpha = 0.05$		LOGODetect			ρ -HESS
		$\theta=0.4$	$\theta=0.5$	$\theta=0.6$	
Trait heritability	0.9	0.001	0.001	0.002	1
	0.8	0	0	0	0.986
	0.7	0.001	0.001	0.003	0.859
	0.6	0	0	0.003	0.567
	0.5	0.001	0.001	0.004	0.217
	0.4	0.004	0.002	0.002	0.071
	0.3	0.01	0.012	0.017	0.004
	0.2	0.01	0.014	0.028	0
	0.1	0.031	0.039	0.063	0
	0.05	0.044	0.051	0.056	0
	0.03	0.042	0.036	0.054	0
	0.01	0.046	0.044	0.05	0

Table 14. Empirical type I error rates for colocalization methods under a non-infinitesimal model with sparse effects.

Empirical type I error rate at posterior probability cutoff 0.95		coloc	gwas-pw
Trait heritability	0.9	0.005	0.002
	0.8	0.001	0.001
	0.7	0.005	0.001
	0.6	0.005	0.002
	0.5	0.001	0.001
	0.4	0.003	0.003
	0.3	0.002	0.002
	0.2	0	0.001
	0.1	0	0.004
	0.05	0	0.001
	0.03	0	0.001
	0.01	0	0

To perform power analysis, we considered two scenarios. In the first scenario, we assumed a sparse genetic model with few causal SNPs. We randomly sampled 10 causal regions for each trait, among which 5 causal regions were shared by both traits. We assumed the per-SNP heritability for all the causal variants to be the same, and varied the size of causal regions. Consistent with other simulations, LOGODetect achieved greater signal points detection rates compared to ρ -HESS when signal region length is small to moderate and compared to colocalization approaches in all settings. In addition, LOGODetect obtained universally higher signal segments detection rates and G-scores (**Figure 4**, response to Comment-1C of Reviewer #1).

Figure 4 (Response to Comment-1C of Reviewer #1). Assessment of statistical power under a non-infinitesimal model with sparse effects. X-axis is in log scale. Signal region length, represents the length of one true signal region. The trait heritability is set to be 0.2 for both traits. Each simulation setting is repeated for 100 times.

Further, in another scenario, we assumed a more sophisticated model with sparse large effects and a polygenic background. For each trait, we randomly selected 10 SNPs to have a large heritability enrichment, among which 5 SNPs were shared by two traits. We refer to other SNPs as low-enrichment SNPs. The effect size correlation of shared, high-enrichment SNPs was set to be 0.9. We varied the heritability proportion explained by the high-enrichment SNPs to see how the power changes. Note that when proportion of heritability explained by the high-enrichment SNPs is 1, this model becomes a simpler sparse model with 10 causal SNPs. LOGODetect and gwas-pw achieved similar performance in three metrics, and they performed universally better than ρ -HESS and coloc (Figure 5, response to Comment-1C of Reviewer #1).

Figure 5 (Response to Comment-1C of Reviewer #1). Assessment of statistical power under a non-infinitesimal model with sparse large effects. X-axis represents the heritability explained by the high enrichment SNPs divided by the total trait heritability. The trait heritability is set to be 0.2 for both traits. Each simulation setting is repeated for 100 times.

Taken together, these results suggest that LOGODetect is robust to fixed effect model and is statistically more powerful than other methods. The discussions above have been incorporated into the revised manuscript (page 5, line 116-118; page 6, line

156-159), **Supplementary Notes**, **Supplementary Table 8**, and **Supplementary Figures 16-17**.

2. If I'm understanding correctly, one simulation was used to assess type I error while another simulation was used to assess power. While the results from these simulations are informative, is it possible to assess error and power under the same set of simulated data? Particularly, how is type I error affected when there are true signal segments with genetic correlations?

Response:

This is a great suggestion. We performed new analyses to assess error under the same set of simulated data in our original submission. The simulation procedure was described again as follows. We randomly selected $N = 5$ segments, each containing $L = 100$ SNPs, as the signal regions shared between two traits. We attributed $p = 0.3$ trait heritability to the signal regions. The genetic effect size for the SNPs in the signal regions follows a multivariate normal distribution:

$$\begin{pmatrix} \beta_i \\ \gamma_i \end{pmatrix} \sim N \left(\begin{bmatrix} 0 \\ 0 \end{bmatrix}, \begin{bmatrix} \frac{p * h^2}{NL} & \frac{p * h^2 * \rho}{NL} \\ \frac{p * h^2 * \rho}{NL} & \frac{p * h^2}{NL} \end{bmatrix} \right).$$

The genetic effect size for the SNPs outside the signal regions follows a different multivariate normal distribution without a genetic correlation:

$$\begin{pmatrix} \beta_i \\ \gamma_i \end{pmatrix} \sim N \left(\begin{bmatrix} 0 \\ 0 \end{bmatrix}, \begin{bmatrix} \frac{(1-p) * h^2}{N_{total} - NL} & 0 \\ 0 & \frac{(1-p) * h^2}{N_{total} - NL} \end{bmatrix} \right).$$

We considered two scenarios. In the first scenario, heritability h^2 for both traits was set to vary from 0.9 to 0.01, and the correlation of genetic effect size of two traits, ρ , was set at 0.9. In the second scenario, the trait heritability h^2 was set at 0.2 for both traits, and the correlation of genetic effect size of two traits ρ was set to vary from 0.9 to 0.1. Each simulation setting was repeated 100 times.

Table 15 shows type I error under the first simulation scenario with varying trait heritability. Here we define an identified region as false positive if its genome distance to the nearest true signal region is larger than 500 KB (to account for LD effect), and type I error rate denotes the proportion of simulations that false positives are identified. We can see that type I error rate of LOGODetect was well-controlled for all settings, which is consistent with the result when we assessed type I error and power separately.

Table 15. Type I error rates under an alternative model with varying heritability.

Type I error rate at significance level $\alpha=0.05$		LOGODetect		
		$\theta=0.4$	$\theta=0.5$	$\theta=0.6$
Heritability	0.5	0.05	0.05	0.06
	0.4	0.06	0.05	0.05
	0.3	0.01	0.01	0.02
	0.2	0.01	0.02	0.02
	0.1	0.07	0.07	0.04
	0.05	0.03	0.07	0.08
	0.01	0.06	0.07	0.08

Table 16 shows type I error under the second simulation scenario with varying correlation. We can see that LOGODetect is still well-calibrated. We conclude that having true signal segments with genetic correlation will not affect the type I error in other genome regions.

Table 16. Type I error rates under an alternative model with varying correlation.

Type I error rate at significance level $\alpha=0.05$		LOGODetect		
		$\theta=0.4$	$\theta=0.5$	$\theta=0.6$
Correlation	0.9	0.01	0.02	0.02
	0.8	0.08	0.08	0.09
	0.7	0.02	0.02	0.02
	0.6	0.05	0.07	0.08
	0.5	0.03	0.04	0.04
	0.4	0	0	0.02
	0.3	0.01	0.03	0.05
	0.2	0.06	0.06	0.07
0.1	0.05	0.06	0.06	

The discussions above have been incorporated into the revised manuscript (page 6, line 161-162) and **Supplementary Tables 9-10**.

3. In order to perform inference, the test statistic Q_{max} is compared against the null distribution of Q_{max} estimated from MC methods. It's not clear to me how family-wise error is controlled. If $\max(|Q(R)|)$ is approximately $Q_{0.95}$, wouldn't that result in a p-value that is approximately 0.05? Could you elaborate on how multiple testing is accounted for?

Response:

Thank you for the question. We first illustrate how family-wise error rate is controlled. We define $Q_{max} = \max_{|R| \leq C} |Q(R)|$, and obtain the null distribution of Q_{max} using Monte Carlo method. The corresponding 95% upper quantile for Q_{max} is denoted as $Q_{0.95}$. For any multiple regions R_1, \dots, R_l , the following inequality holds under the null.

$$\begin{aligned} & \Pr[|Q(R_1)| \geq Q_{0.95} \text{ or } |Q(R_2)| \geq Q_{0.95} \text{ or } \dots |Q(R_l)| \geq Q_{0.95}] \\ & \leq \Pr\left[\max_{1 \leq i \leq l} |Q(R_i)| \geq Q_{0.95}\right] \\ & \leq \Pr[Q_{max} \geq Q_{0.95}] \\ & = 0.05 \end{aligned}$$

In another word, the probability that there is a genomic region R with $Q(R)$ greater than $Q_{0.95}$ is less than 0.05 under the null. Therefore, family-wise error is controlled. Second, we will illustrate how multiple testing is accounted for. Because family-wise error control is conservative, we split the genome into 204 LD blocks (Each LD block spans 15 MB on average) and controlled for family wise error within each LD block. Next, we used FDR to account for the multiple testing problem concerning all LD blocks.

4. When sampling z-scores from the null, it's unclear how the vector z is decomposed into z_1, z_2, z_3 . How were these 1MB regions defined?

Response:

Thank you for the question. As the dimension of the vector z is high, we partition z into a set of sub-vectors: z_1, z_2, z_3 , etc. These sub-vectors were defined by their genome positions, each spanning 1MB bases (i.e. chr1:0-1MB, chr1:1-2MB, and so on). We implicitly assumed that SNPs that are 2 blocks away are independent with each other while allowing SNPs in neighboring blocks to be in LD. We used an iterative approach to generate each block by conditioning on the previous block: $z_i | z_{i-1} \sim N(\Sigma_{i,i-1} \Sigma_{i-1,i-1}^{-1} z_{i-1}, \Sigma_{i,i} - \Sigma_{i,i-1} \Sigma_{i-1,i-1}^{-1} \Sigma_{i-1,i} z_{i-1})$, where $\Sigma_{i,j} = \text{cov}[z_i, z_j]$. The distribution of z can be approximated by the product of successive multivariate normal distributions.

The discussions above have been incorporated into the revised manuscript (page 18, line 541-543).

5. *In your model where sample overlap exists, the first equation under the header “Extension for sample overlaps” appears to be missing SNP effects, β and γ .*

Answer:

Thank you so much for pointing out the mistake. We have corrected it in the revised main text (page 19, line 576-577).

Reference.

1. Giambartolomei, C. *et al.* Bayesian test for colocalisation between pairs of genetic association studies using summary statistics. *PLoS genetics* **10**, e1004383 (2014).
2. Pickrell, J.K. *et al.* Detection and interpretation of shared genetic influences on 42 human traits. *Nature genetics* **48**, 709 (2016).
3. Bulik-Sullivan, B. *et al.* An atlas of genetic correlations across human diseases and traits. *Nature genetics* **47**, 1236 (2015).
4. Lu, Q. *et al.* A Powerful Approach to Estimating Annotation-Stratified Genetic Covariance via GWAS Summary Statistics. *Am J Hum Genet* **101**, 939-964 (2017).
5. Zhao, Z. *et al.* Fine-tuning Polygenic Risk Scores with GWAS Summary Statistics. *bioRxiv*, 810713 (2019).
6. Finucane, H.K. *et al.* Partitioning heritability by functional annotation using genome-wide association summary statistics. *Nature genetics* **47**, 1228 (2015).
7. Noble, E.P. The DRD2 gene in psychiatric and neurological disorders and its phenotypes. *Pharmacogenomics* **1**, 309-333 (2000).
8. Atz, M.E., Rollins, B. & Vawter, M.P. NCAM1 association study of bipolar disorder and schizophrenia: polymorphisms and alternatively spliced isoforms lead to similarities and differences. *Psychiatric genetics* **17**, 55 (2007).
9. Wainwright, S.R. & Galea, L.A. The neural plasticity theory of depression: assessing the roles of adult neurogenesis and PSA-NCAM within the hippocampus. *Neural plasticity* **2013**(2013).
10. Christiansen, G.B. *et al.* The sorting receptor SorCS3 is a stronger regulator of

- glutamate receptor functions compared to GABAergic mechanisms in the hippocampus. *Hippocampus* **27**, 235-248 (2017).
11. Oetjen, S., Mahlke, C., Hermans-Borgmeyer, I. & Hermeijer, G. Spatiotemporal expression analysis of the growth factor receptor SorCS3. *Journal of Comparative Neurology* **522**, 3386-3402 (2014).
 12. Kajizuka, M. *et al.* Serum levels of platelet-derived growth factor BB homodimers are increased in male children with autism. *Progress in Neuro-Psychopharmacology and Biological Psychiatry* **34**, 154-158 (2010).
 13. Jeng, X.J., Cai, T.T. & Li, H. Optimal sparse segment identification with application in copy number variation analysis. *Journal of the American Statistical Association* **105**, 1156-1166 (2010).
 14. Li, Z., Liu, Y. & Lin, X. Simultaneous Detection of Signal Regions Using Quadratic Scan Statistics With Applications in Whole Genome Association Studies. *arXiv preprint arXiv:1710.05021* (2017).
 15. Berisa, T. & Pickrell, J.K. Approximately independent linkage disequilibrium blocks in human populations. *Bioinformatics* **32**, 283 (2016).
 16. Watanabe, K., Taskesen, E., Van Bochoven, A. & Posthuma, D. Functional mapping and annotation of genetic associations with FUMA. *Nature communications* **8**, 1-11 (2017).
 17. Grotzinger, A.D. *et al.* Genomic structural equation modelling provides insights into the multivariate genetic architecture of complex traits. *Nature human behaviour* **3**, 513 (2019).

18. Turley, P. *et al.* Multi-trait analysis of genome-wide association summary statistics using MTAG. *Nature genetics* **50**, 229 (2018).
19. Bhattacharjee, S. *et al.* A subset-based approach improves power and interpretation for the combined analysis of genetic association studies of heterogeneous traits. *The American Journal of Human Genetics* **90**, 821-835 (2012).
20. Speed, D., Hemani, G., Johnson, M.R. & Balding, D.J. Improved heritability estimation from genome-wide SNPs. *The American Journal of Human Genetics* **91**, 1011-1021 (2012).
21. Yang, J. *et al.* Common SNPs explain a large proportion of the heritability for human height. *Nat Genet* **42**, 565-9 (2010).

REVIEWER COMMENTS

Reviewer #1 (Remarks to the Author):

I appreciate the additional results that the authors added to address referees' comments. I also have concerns on different aspects of this revision.

First, I have three main comments on how methods are evaluated in simulations and data analyses.

Across all null simulations (SUPP TAB 1-8), the published method rho-HESS (REF 12) consistently shows a worrying pattern: its empirical type I error is deflated (e.g. exact 0) when trait heritability is small (e.g. <0.2) and it is inflated (e.g. exact 1) when trait heritability is large (e.g. >0.6). It seems that the authors do not provide sufficient information to help readers understand this observation. Perhaps the null datasets are simulated in some way such that key assumptions of rho-HESS do not hold? If so, then the authors should consider null simulations where rho-HESS can achieve the nominal type I error rate.

The present simulation settings seem quite different from what we have learned from real GWAS data. In particular, some simulated datasets assume that "20,211 SNPs" (LINE 597) explain moderate to high trait heritability (e.g. 0.4-0.9), whereas in current literature several millions of genome-wide SNPs (LINE 591) can only explain trait heritability of similar magnitude (e.g. 0.1-0.6). This further suggests that the simulated per-SNP heritability is much higher than the actual per-SNP heritability in real data. If I am correct here, then there are at least two important implications. First, the "substantial inflation of type I error for rho-HESS" observed in high heritability ("greater than 0.4", LINE 120-121) could be due to the (somehow unrealistically) large per-SNP heritability. (I make such speculations mainly by looking at SUPP TAB 4: if the authors fix the heritability of the 2nd trait as 0.2, the rho-HESS's empirical type I error is no longer as high as 1 when heritability of the 1st trait is high.) Second (and more importantly), the rho-HESS performance in high heritability simulations (>0.4) does not seem adequate to explain its performance in BIP and SCZ ($h^2=0.35, 0.43$, LINE 235-237), since these BIP and SCZ heritability estimates are derived from millions of SNPs rather than 20K SNPs.

I still find it hard to understand the replication analysis in UKBB data -- the authors present one method LOGODetect to identify "local" genetic correlations, and then they use a different method GNOVA (REF 11) for replication based on the "aggregated" genetic covariance of all regions identified by each method. For BIP and SCZ I understand the authors cannot apply LOGODetect in both discovery and replication datasets, largely because of low case counts in UKBB (LINE 250-251). But is it possible to apply the same local method on both discovery and replication datasets for the two continuous traits (NEU and IQ)?

I have two comments related to theta, a parameter that seems to be a key component of LOGODetect. It is widely accepted that LD plays an important role in the analysis of GWAS association statistics (REF 10), and in this work, the parameter theta "controls the impact of LD" (LINE 84).

The authors repeated LOGODetect analysis on BIP and SCZ with different theta values as I previously recommended. The results do not seem to support the claim that "LOGODetect is robust to theta" (LINE 373). Notably, if theta changes from 0.5 (recommended value) to 0.6, the number of shared regions increases from 33 to 63 (LINE 369). The difference is still quite obvious even if accounting for the fact that "five regions identified with theta=0.5 were broken down to 13 smaller regions identified when theta=0.6" (LINE 372).

Since LOGODetect results seem to be sensitive to the choice of theta, now I feel it might be necessary to estimate theta from real data, either by optimizing some likelihood/loss functions to

select the "best" theta, or, by placing some priors on theta and then averaging results across "all" thetas. That said, I am more concerned now whether users should follow the authors' recommendation to use "0.5 as the choice of theta in practice" (LINE 142-143), since the simulations are based on 20K SNPs (much smaller than real GWAS).

The remaining three comments focus on presentation (from a reader's perspective).

The current captions of main display items are not informative enough for readers to understand the displays easily. For example, FIG 2 reports three key metrics to assess statistical power (LINE 126-127), but their definitions are not easily accessible. It is important that FIG 2 caption and/or associated main text provide a simple explanation in words of what these metrics mean. The reader should not have to sort through the Method section (LINE 630-641) to understand the basic definition of these metrics. In addition, which metrics should we trust more? For example, when trait heritability is between 0.1 and 0.2 (which is more realistic in terms of per-SNP heritability; see above), *gwas-pw* (REF 17) seems to outperform LOGODetect in terms of metric A alone.

There also seems to be room for improvement in data and code reporting. For example, the 1000 Genomes project is a key resource that enables the application of LOGODetect on real GWAS (LINE 523, 646), but the authors did not provide a database link and did not cite it properly. The 1000 Genome project also has multiple releases, and the authors did not report which version they used. Another example is rho-HESS. The software link is missing. The authors also did not describe how they used rho-HESS in this work, which makes it difficult for readers to understand the poor performance of rho-HESS in simulations (see above).

I wonder if the authors should be more careful with non-trivial technical statements. For example, when explaining "the expected value of [sum] is larger in regions with strong LD" (SUPP NOTE PAGE 8), the authors only used a toy example with some hand-waving argument (SUPP NOTE LINE 220). Does this claim only hold in this toy example? If yes, then the authors should let readers know. If no, then I hope the authors could provide a more general derivation (while keeping the toy example for introductory purposes).

Reviewer #2 (Remarks to the Author):

I appreciate the efforts that the authors have made to respond to my previous comments. The responses and additions were so substantial that it took me almost as much time to re-review as my original review. My major concerns have been addressed. I have a few remaining concerns below.

L33 this is the first of many mentions of "hub" which is not properly defined.

L52-54 "Thus, ... different traits" this is a non-sequitur: genetic correlation is not an alternative to single-SNP association, it answers a different question. Genome-wide single-trait SNP analysis is the alternative. It's not a major error but I find it worrying that this mistake should appear so early in the paper.

L126-7 The panel descriptions just repeat the Y-axis legends which is useless - you miss an opportunity to try to give insight to the casual reader. In particular you discuss G-score here and in the text nearby without saying what it is in either method. You should also state the theta value used for your method as this issue has not been discussed yet.

L139-142 "LOGODetect obtained a larger G-score with smaller theta values ... and in particular, the G-score was almost the same for theta ..." This doesn't make sense, was the G-score larger or the same? "LOGODetect worked reasonably well in all three measures when theta was set to 0.5." This is a disappointingly weak justification given that I raised this issue in the first review. You

should quantify e.g. the loss from using 0.4 or 0.6 instead of 0.5. There is now discussion of theta later in the manuscript but that is not referenced here - it seems that the authors have not understood their own manuscript organisation - why make the recommendation before discussing the evidence and without even mentioning that there is later a discussion of this issue?

Starting from P6, but it gets particularly bad on P 12 and 13, there are too many numerical results in the text making it very tedious and hard to read. Numerical results should be in tables with general commentary in the text and just a few highlighted values that are of particular interest.

L374 "LOGODetect is robust to theta and identifies more short segments as theta increased". Again this is inconsistent: is it robust or is the segment length sensitive to theta?

L375 "Our implemented software allows customized choice of theta based on users' preference on signal region size" This seems inconsistent with your previous recommendation for theta = 0.5, so are you letting users decide based on the preference for region size or are you recommending 0.5 ? Even in this expanded discussion you produce no real support for this recommendation.

L379 "identified genetic regions that may be shared " inadequate wording - Firstly "may" is too weak to mean anything, and what does "shared" mean?

Reviewer #4 (Remarks to the Author):

The editor requested an evaluation of the authors response to reviewer 3. Below I summarize an opinion in point by point of that response.

R3.1.-

Reviewer 1 is absolutely on target by commenting that the genetic correlation is well defined (Falconer and Mackay, 'introduction to quantitative genetics'). The authors' response is inaccurate, while they are arguing that several definitions exist about the genetic correlation.

In short, if for two traits, we could observe the exact amount of mean deviations due to additive genetic effects, we could calculate the additive genetic correlation between these traits.

However, additive effects are always unknown and need to be estimated. Researchers can use estimates of SNP markers, partial effects captured by a model. Inevitably, the study will have missing any not mapped genomic variation in an array. The genomic heritability has demonstrated that as much as 50% of the additive effects can be estimated with today's array density (even less for some traits and assays. Thus, the genetic correlation can be PARTIALLY estimated. Further, the 'missing' aspect of the genetic variation in genotyping assays has been the center of the 'missing heritability' debate. Similarly, the 'misleading' genetic correlation debate of what exactly is being estimated as 'genomic correlation based on a specific genotyping array' (e.g., Genomic heritability: what is it? De los Campos et al., PLoS Genet. 2015).

In sum, the authors are calculating correlations between SNPs (not additive genetic correlations). That should be clear in the paper and well acknowledge, it is untrue that genetic correlation has different but similar definitions. However, there indeed exists an array of methods to estimate approximations to the genomic correlations.

The authors attempt a series of simulations which are not entirely meaningful; all simulations generate 'correlations between SNPs', and then retrieve their simulations. The only solution for them is acknowledging in the discussion or limitations the differences between the correlation of SNPs and the concept of additive genetic correlation.

R3.2.-The reviewer brought a critical point; there is no need to create multiple simulations to

measure type I error and power. The standard is to generate one simulation and measure everything in that simulation.

The authors fixed the issue by modifying the simulation as requested by reviewer 3.

Besides, the simulation is unrealistic and oversimplified. Regions are generated with total independence of one and another, which does not exist in reality. LD blocks are also not all of the same sizes, at all. Finally, common ancestry load / differentiation of origin generates genome-wide similarities (reflected in population structure), unaccounted on simulation as the one proposed.

However, the manuscript also includes a real data analysis. Thus, the simulation has a minor component that could demonstrate an idealistic 'best' possible performance in an uncomplicated scenario.

R3.3. The authors explain their multiple testing control.

R3.4. The sampling of Z-scores in the null is clear in the present version of the manuscript.

R3.5. The authors have resolved the typo pointed out by R3.

Response to Reviewer #1:

I appreciate the additional results that the authors added to address referees' comments. I also have concerns on different aspects of this revision.

1. First, I have three main comments on how methods are evaluated in simulations and data analyses.

Across all null simulations (SUPP TAB 1-8), the published method rho-HESS (REF 12) consistently shows a worrying pattern: its empirical type I error is deflated (e.g. exact 0) when trait heritability is small (e.g. <0.2) and it is inflated (e.g. exact 1) when trait heritability is large (e.g. >0.6). It seems that the authors do not provide sufficient information to help readers understand this observation. Perhaps the null datasets are simulated in some way such that key assumptions of rho-HESS do not hold? If so, then the authors should consider null simulations where rho-HESS can achieve the nominal type I error rate.

The present simulation settings seem quite different from what we have learned from real GWAS data. In particular, some simulated datasets assume that "20,211 SNPs" (LINE 597) explain moderate to high trait heritability (e.g. 0.4-0.9), whereas in current literature several millions of genome-wide SNPs (LINE 591) can only explain trait heritability of similar magnitude (e.g. 0.1-0.6). This further suggests that the simulated per-SNP heritability is much higher than the actual per-SNP heritability in real data. If I am correct here, then there are at least two important implications. First, the "substantial inflation of type I error for rho-HESS" observed in high heritability ("greater than 0.4", LINE 120-121) could be due to the (somehow unrealistically) large per-SNP heritability. (I make such speculations mainly by looking at SUPP TAB 4: if the authors fix the heritability of the 2nd trait as 0.2, the rho-HESS's empirical type I error is no longer as high as 1 when heritability of the 1st trait is high.) Second (and more importantly), the rho-HESS performance in high heritability simulations (>0.4) does not seem adequate to explain its performance in BIP and SCZ ($h^2=0.35, 0.43$, LINE 235-237), since these BIP and SCZ heritability estimates are derived from millions of SNPs rather than 20K SNPs.

Response:

Thank you for the comments. As suggested, we have performed additional simulations using genotype data with denser markers and lower heritability settings. Based on 503 individuals with European ancestry from the 1000 Genomes Project Phase 3 data, we simulated genotype data for 50,000 individuals with minor allele frequency (MAF) greater than 5% on chromosome 1 using HAPGEN2¹. 336,532 variants remained in the dataset after removing strand ambiguous SNPs. We note that this genotype simulation approach is consistent with the simulation setting in the ρ -HESS paper².

We evaluated our method under a heritability enrichment model, where 30% of trait heritability was attributed to 5,000 randomly selected SNPs and 70% of trait heritability was attributed to the remaining SNPs, similar to the simulation settings in our previous submission. We varied the trait heritability from 0.01 to 0.03. Note that if

we assume heritability is distributed proportionally in the genome, then the heritability values of 0.01 and 0.03 for chromosome 1 will correspond to the approximate heritability values of 0.12 and 0.36 for the whole genome, respectively, which are on the same scale of SNP heritability for various traits. Under this setting, the type I error of both LOGODetect and ρ -HESS were under control (**Table 1**). We note that the deflation of type I error observed for ρ -HESS is not contradictory to results published in ρ -HESS paper². ρ -HESS was formulated as an estimation problem, which aims to quantify the genetic correlation of local genomic regions, instead of the hypothesis testing problem in our manuscript. In their paper, they have shown simulation results to demonstrate that the local genetic correlation can be accurately estimated (Figure 3B in ρ -HESS paper) when the true parameter is 0. However, the evidence shown in the ρ -HESS paper could not rule out deflation when the method is used for inference.

Table 1 (Supplementary Table 9 in the revised manuscript). Type I error rates under a heritability enrichment model. Type I error rates denote the proportion of simulations that significant segments harboring local genetic correlation are identified in null simulations. In these simulations, 30% of trait heritability was assigned to 5,000 randomly selected SNPs and 70% trait heritability was assigned to the other SNPs. Two cohorts did not have any overlapping samples.

	Type I error rate at significance level $\alpha=0.05$				
Trait heritability	LOGODetect			LOGODetect with adaptive θ	ρ -HESS
	$\theta=0.4$	$\theta=0.5$	$\theta=0.6$		
0.03	0.02	0.01	0	0.02	0
0.02	0.05	0.01	0	0.05	0
0.01	0	0	0.01	0.01	0

To perform power analysis, we assumed a heritability enrichment model, where five segments, each containing 1000 SNPs, were randomly selected as the signal regions shared between two traits, and 30% trait heritability was assigned to the signal regions. As shown in **Figure 1**, LOGODetect achieved greater signal points detection rates compared to ρ -HESS across all heritability values, gwas-pw showed greater signal points detection when heritability was 0.01 and 0.02, but LOGODetect showed universally higher signal segments detection rates and G-scores compared to all other methods, which is consistent with simulation results in our previous submission.

Taken together, these new simulations have complemented the analyses we have performed in the original submission. We have demonstrated that LOGODetect can obtain robust results under realistic simulation settings, and outperform ρ -HESS in both type I error control and power measures. The discussions above have been incorporated into the revised manuscript (page 5, line 126-128), **Supplementary Notes, Supplementary Table 9, and Supplementary Figure 17.**

Figure 1 (Supplementary Figure 17 in the revised manuscript). Assessment of statistical power under a heritability enrichment model with varying trait heritability. The X-axis shows the heritability for simulated traits. The Y-axis shows the statistical power assessed by three different measures: (A) signal points detection rate, (B) signal segments detection rate, and (C) G-score. Due to the observed inflation in type I error rates, we adjusted the significance cutoff for gwas-pw so that its type I error rates achieve 0.05.

2. I still find it hard to understand the replication analysis in UKBB data -- the authors present one method LOGODetect to identify "local" genetic correlations, and then they use a different method GNOVA (REF 11) for replication based on the "aggregated" genetic covariance of all regions identified by each method. For BIP and SCZ I understand the authors cannot apply LOGODetect in both discovery and replication datasets, largely because of low case counts in UKBB (LINE 250-251). But is it possible to apply the same local method on both discovery and replication datasets for the two continuous traits (NEU and IQ)?

Response:

We acknowledged that the replication approach suggested by the reviewer would be direct and convincing. However, the two continuous traits in the discovery study comprise data from UKBB. In fact, 372,903 of the 390,278 total samples in the NEU GWAS were from the UKBB. Similarly, 195,653 of the 269,867 total samples in the GWAS on IQ were from the UKBB. Therefore, it would be inappropriate to carry out replication studies with GWAS conducted on UKBB.

In our previous submission, we have demonstrated that we expect the direct replication approach to have insufficient statistical power due to limited sample sizes (especially number of cases). We have also developed an alternative strategy to demonstrate the validity of our findings. We chose to assess the enrichment of aggregated genetic covariance across all identified local genomic segments in the replication dataset instead of statistical significance of each segment. Our previous simulation results showed that the detected regions are expected to be substantially

enriched for genetic covariance even when n_{eff} (effective sample size) decreased to the level of UKBB replication cohort (**Figure 2A**), whereas randomly selected regions showed no enrichment for genetic covariance when n_{eff} varied (**Figure 2B**). These results demonstrated the effectiveness of the proposed replication approach.

Figure 2 (Supplementary Figure 70 in the revised manuscript). Enrichment of aggregated genetic covariance with varying effective sample size. (A) Regions identified by LOGODetect in discovery cohort show substantial genetic covariance enrichment in UKBB replication cohort. (B) Randomly selected regions with same sizes as (A) show no genetic covariance enrichment in UKBB replication cohort. Genetic covariance of randomly selected regions may have opposite sign compared to global genetic covariance, therefore the corresponding genetic covariance enrichment may be negative. Here the X-axis denotes the squared root of the effective sample size of the simulated BIP cohort times that of the simulated SCZ cohort. For each panel, the box on the right corresponds to effective sample size in the BIP-SCZ discovery cohorts ($\sqrt{ESS_1 * ESS_2} = 70,214$), the box on the left corresponds to effective sample size in the BIP-SCZ UKBB replication cohorts ($\sqrt{ESS_1 * ESS_2} = 3,111$). We repeated the simulations for 100 times for each effective sample size.

We applied this replication approach to the UKBB replication datasets of BIP and SCZ (**Table 2**). The regions identified by LOGODetect and ρ -HESS both showed significant genetic covariance in the replication data. Moreover, the regions identified by LOGODetect had a 6.7-fold higher enrichment of genetic covariance compared to that of ρ -HESS, which demonstrated again that LOGODetect can detect the true signal regions with improved precision. Regions identified by gwas-pw did not show significant genetic covariance, while regions identified by coloc showed genetic covariance with the opposite sign.

Finally, although a direct replication analysis of NEU and IQ cannot be conducted due to sample overlap, we applied our method to two anthropometric traits: height and body-mass index (BMI) in this revision. We conducted analysis on summary statistics of height³ (n=253,288) and BMI⁴ (n=236,231) from the GIANT consortium and replicated our findings in the UKBB (n=455,332 and 454,841). We identified 24

regions with significant local genetic correlation in the discovery analysis. 17 of 24 regions identified in the discovery stage were successfully replicated, suggesting the effectiveness of LOGODetect to identify replicable genomic regions with local genetic correlations.

Table 2 (Table 1 in the revised manuscript). Stratified genetic covariance analysis on UKBB replication cohorts

	Genetic Cov*	s.e.	p-value	Proportion of genetic cov	Proportion of SNPs	Fold enrichment
LOGODetect	2.18e-4	6.65e-5	1.04e-3	11.50%	1.15%	10.02
ρ -HESS	6.62e-4	3.16e-4	3.61e-2	30.00%	20.12%	1.49
coloc	-5.70e-5	2.30e-5	1.33e-2	-2.85%**	0.34%	-8.36**
gwas-pw	3.84e-5	6.54e-5	5.57e-1	1.92%	1.61%	1.20

* Genetic Cov represents estimated genetic covariance of the identified regions using GNOVA.

** Genetic covariance of regions identified by coloc has opposite sign compared to global genetic covariance, therefore the corresponding proportion of genetic covariance and fold enrichment are negative.

Taken together, we highlighted the lack of feasibility to directly replicate our findings on neuropsychiatric traits. We proposed an alternative strategy to evaluate the aggregated genetic covariance over all identified local regions, and demonstrated its effectiveness through simulations. In addition, we applied the direct replication approach to two additional traits (height and BMI) and replicated 17 of 24 regions identified in the discovery stage. The results and discussions above have been incorporated into the revised manuscript (page 9, line 274-279), **Supplementary Notes, Supplementary Table 18, and Supplementary Figure 70.**

3. I have two comments related to θ , a parameter that seems to be a key component of LOGODetect. It is widely accepted that LD plays an important role in the analysis of GWAS association statistics (REF 10), and in this work, the parameter θ "controls the impact of LD" (LINE 84).

The authors repeated LOGODetect analysis on BIP and SCZ with different θ values as I previously recommended. The results do not seem to support the claim that "LOGODetect is robust to θ " (LINE 373). Notably, if θ changes from 0.5 (recommended value) to 0.6, the number of shared regions increases from 33 to 63 (LINE 369). The difference is still quite obvious even if accounting for the fact that "five regions identified with $\theta=0.5$ were broken down to 13 smaller regions identified when $\theta=0.6$ " (LINE 372).

Since LOGODetect results seem to be sensitive to the choice of θ , now I feel it might be necessary to estimate θ from real data, either by optimizing some likelihood/loss functions to select the "best" θ , or, by placing some priors on θ

and then averaging results across "all" thetas. That said, I am more concerned now whether users should follow the authors' recommendation to use "0.5 as the choice of theta in practice" (LINE 142-143), since the simulations are based on 20K SNPs (much smaller than real GWAS).

Response:

We appreciate the constructive comments. We acknowledge that the genetic architecture of the real GWAS datasets may be different from that of our simulated datasets. Therefore, it can be better to infer the "best" θ adaptively from the data. Here, we propose to select the optimal θ by maximizing the aggregated genetic covariance in the identified regions. We varied the value of θ from 0.4 to 0.6, and chose the best θ such that the identified regions have the largest genetic variance. We investigated the performance of LOGODetect with adaptive θ under the same simulation settings in response to Comment 1. Type I error rate was well-controlled for LOGODetect with adaptive θ (Table 1).

We have also updated the results on seven neuropsychiatric traits using adaptively selected θ (Table 3 and Figure 3). For four of the trait pairs, 0.5 was selected as the optimal θ , and for the other 15 pairs, 0.6 was selected.

The discussions above have been incorporated into the revised manuscript (page 7, line 182-186; page 19, line 516-520) and Supplementary Table 13, and all the related figures and analyses in the main text have been updated accordingly.

Table 3 (Supplementary Table 13 in the revised manuscript). Proportion of genetic covariance identified by LOGODetect with varying θ . Proportion of genetic covariance identified by LOGODetect with varying θ is calculated via stratified LDSC, and the largest absolute value of proportion of genetic covariance is highlighted in bold and red font. Notably, proportion of genetic covariance identified by LOGODetect with $\theta=0.6$ for BIP-IQ and MDD-IQ are negative because the local genetic covariance has opposite sign compared to global genetic covariance.

Trait 1	Trait 2	$\theta=0.4$	$\theta=0.5$	$\theta=0.6$
BP	SCZ	7.80%	9.38%	18.76%
BP	M DD	0%	1.74%	2.47%
BP	NEU	0%	1.52%	0.77%
BP	ADHD	0%	0%	0%
BP	ASD	0%	0%	0%
BP	IQ	0%	0%	-12%
SCZ	M DD	4.25%	6.12%	6.93%
SCZ	NEU	2.74%	5.19%	9.06%
SCZ	ADHD	3.73%	3.36%	9.27%
SCZ	ASD	0%	0%	3.56%
SCZ	IQ	3.53%	4.72%	8.70%
M DD	NEU	3.35%	6.77%	8.75%
M DD	ADHD	1.66%	2.59%	2.50%
M DD	ASD	0.85%	1.79%	2.07%
M DD	IQ	0%	0%	-2.83%
NEU	ADHD	0%	0%	3.78%
NEU	ASD	0%	0%	1.52%
NEU	IQ	0%	4.68%	8.57%
ADHD	ASD	4.98%	6.16%	3.23%
ADHD	IQ	0.78%	4.23%	8.67%
ASD	IQ	0%	3.55%	1.96%

Figure 3 (Figure 3A in the revised manuscript). LOGODetect identifies genome regions contributing to multiple neuropsychiatric traits. Heatmap shows the genetic correlation estimates (upper triangle) and the number of segments with local genetic correlation identified by LOGODetect (lower triangle) between seven neuropsychiatric traits; Barplot shows the observed scale heritability estimates and standard errors of seven traits.

4. The remaining three comments focus on presentation (from a reader's perspective).

The current captions of main display items are not informative enough for readers to understand the displays easily. For example, FIG 2 reports three key metrics to assess statistical power (LINE 126-127), but their definitions are not easily accessible. It is important that FIG 2 caption and/or associated main text provide a simple explanation in words of what these metrics mean. The reader should not have to sort through the Method section (LINE 630-641) to understand the basic definition of these metrics. In addition, which metrics should we trust more? For example, when trait heritability is between 0.1 and 0.2 (which is more realistic in terms of per-SNP heritability; see above), gwas-pw (REF 17) seems to outperform LOGODetect in terms of metric A alone.

Response:

Thank you for the constructive suggestions. We have included more illustrative definition for three statistical power metrics in Figure 2 in the main text. As for the three metrics of statistical power, a limitation for using signal points detection rate and

signal segments detection rate to measure the sensitivity is that both metrics are 1 if the identified region is a long segment, e.g. the whole genome, whereas G-score is a more informative alternative which can measure the specificity and sensitivity jointly. Therefore, G-score should be the priority. The discussions above have been incorporated into the revised manuscript (page 6, line 141-145).

5. There also seems to be room for improvement in data and code reporting. For example, the 1000 Genomes project is a key resource that enables the application of LOGODetect on real GWAS (LINE 523, 646), but the authors did not provide a database link and did not cite it properly. The 1000 Genome project also has multiple releases, and the authors did not report which version they used. Another example is rho-HESS. The software link is missing. The authors also did not describe how they used rho-HESS in this work, which makes it difficult for readers to understand the poor performance of rho-HESS in simulations (see above).

Response:

Thank you for pointing this out. We used individuals with European ancestry from the 1000 Genomes Project phase 3 data⁵ as the reference panel (<ftp://ftp.1000genomes.ebi.ac.uk/vol1/ftp/release/20130502/>). We used ρ -HESS following the standard procedure described in https://huwenboshi.github.io/hess/local_rhog/. In particular, we used 1,703 approximately LD-independent regions in the genome (1.6 Mb in width on average) as the pre-specified genomic regions in real data applications of ρ -HESS. This set of regions was inferred using ldetect⁶ and was recommended by the original papers of ρ -HESS. In simulation studies with WTCCC chromosome 1 genotypes, variance estimation of local genetic correlation by ρ -HESS was negative (due to small number of SNPs in one LD-independent region, suggested by the ρ -HESS software). Therefore, we merged three adjacent LD blocks into one block as the pre-specified genomic regions. The discussions above have been incorporated into the **Supplementary Notes**.

6. I wonder if the authors should be more careful with non-trivial technical statements. For example, when explaining "the expected value of [sum] is larger in regions with strong LD" (SUPP NOTE PAGE 8), the authors only used a toy example with some hand-waving argument (SUPP NOTE LINE 220). Does this claim only hold in this toy example? If yes, then the authors should let readers know. If no, then I hope the

authors could provide a more general derivation (while keeping the toy example for introductory purposes).

Response:

We appreciate this comment. We added new analyses to demonstrate this point. We used BIP and SCZ as an example. We partitioned the genome into 30,957 blocks, each spanning 200 SNPs. Then we grouped these blocks into three equally-sized categories (i.e., high LD, medium LD, and low LD) according to their LD strength (sum of LD scores of SNPs in each block). Each category contains 10,319 blocks. Two sample t-test for each category pair suggest significantly larger $|\sum_i z_{1i}z_{2i}|$ in blocks with strong LD (maximum $p=2.16e-68$, **Figure 4**). The discussions above have been incorporated into the **Supplementary Notes** and **Supplementary Figure 67**.

Figure 4 (Supplementary Figure 67 in the revised manuscript). $|\sum_i z_{1i}z_{2i}|$ are larger in blocks with strong LD. Genome is partitioned into 30,957 blocks, each containing 200 SNPs. Blocks are classified into three categories according to the LD strength (sum of LD scores). For each category pair, one-sided two-sample t-test is performed and the corresponding p value is provided. Here, blocks beyond the extreme of the lower whisker and the extreme of the upper whisker are omitted for visualization purposes.

Response to Reviewer #2:

I appreciate the efforts that the authors have made to respond to my previous comments. The responses and additions were so substantial that it took me almost as much time to re-review as my original review. My major concerns have been addressed. I have a few remaining concerns below.

1. L33 this is the first of many mentions of "hub" which is not properly defined.

Response:

Thank you for the comment. We define a hub region as a small genome segment harboring local genetic correlations for multiple trait pairs. We have clarified this concept in the revised manuscript (page 3, line 76-77).

2. L52-54 "Thus, ... different traits" this is a non-sequitur: genetic correlation is not an alternative to single-SNP association, it answers a different question. Genome-wide single-trait SNP analysis is the alternative. It's not a major error but I find it worrying that this mistake should appear so early in the paper.

Response:

We apologize for the confusion. We meant to state that single SNP-based methods to model pleiotropy effects of genome-wide significant SNPs may not be sufficient to characterize the full landscape of genetic similarity of complex traits. We have updated this claim in the revised manuscript (page 3, line 52-53).

3. L126-7 The panel descriptions just repeat the Y-axis legends which is useless - you miss an opportunity to try to give insight to the casual reader. In particular, you discuss G-score here and in the text nearby without saying what it is in either method. You should also state the theta value used for your method as this issue has not been discussed yet.

Response:

Thank you for pointing this out. We have updated the panel legends to provide more complete information on the three metrics. In addition, the theta value for LOGODetect in simulations was set to be 0.5. The panel description of **Figure 2** in the main text has been updated in the revised manuscript (page 5, line 132-134; page 6, line 141-145).

4. L139-142 "LOGODetect obtained a larger G-score with smaller theta values ... and in particular, the G-score was almost the same for theta ..." This doesn't make sense, was the G-score larger or the same? "LOGODetect worked reasonably well in all three measures when theta was set to 0.5." This is a disappointingly weak justification given that I raised this issue in the first review. You should quantify e.g. the loss from using 0.4 or 0.6 instead of 0.5. There is now discussion of theta later in the manuscript but that is not referenced here - it seems that the authors have not understood their own manuscript organization - why make the recommendation before discussing the evidence and without even mentioning that there is later a discussion of this issue?

Response:

Thank you for pointing this out. We have updated the sentence into "LOGODetect obtained a smaller G-score with larger θ values, but G-score did not show substantial changes when θ was small (e.g. 0.3, 0.4, or 0.5)" in the revised manuscript (page 6, line 150-152). In addition, following comments from reviewers 1 and 2, we proposed an approach to adaptively select θ by maximizing the explained genetic covariance. We discuss the details of this approach in our response to Comment 6.

5. Starting from P6, but it gets particularly bad on P 12 and 13, there are too many numerical results in the text making it very tedious and hard to read. Numerical results should be in tables with general commentary in the text and just a few highlighted values that are of particular interest.

Response:

Thank you for pointing this out. We have summarized the main numerical results into **Table 4**, corresponding to Supplementary Table 21 in the revised manuscript.

Table 4 (Supplementary Table 21 in the revised manuscript). Putative target genes for five hub regions identified in at least six pair-wise analyses

Locus	LOGODetect	ρ -HESS	coloc	gwas-pw	Putative target genes
Chr11: 112.742-113.458 MB	BIP-SCZ, BIP-NEU, BIP-IQ, SCZ-MDD, SCZ-NEU, SCZ-IQ, MDD-NEU, NEU-ADHD, NEU-IQ	BIP-SCZ, SCZ-NEU, SCZ-IQ		SCZ-NEU	DRD2 , NCAM1
Chr11: 133.793-134.28 1MB	BIP-SCZ, SCZ-MDD, SCZ_NEU, SCZ_ADHD, SCZ-IQ, NEU-IQ, ADHD-IQ	BIP-SCZ, SCZ-NEU, SCZ-IQ		SCZ-NEU, NEU-IQ	IGSF9B
Chr14: 29.499-30.194M B	BIP-SCZ, BIP-IQ, SCZ-NEU, SCZ-ADHD, SCZ-IQ, NEU-IQ, ADHD-IQ	BIP-SCZ			PRKD1 , FOXP1
Chr3: 71.462-71.687M B	SCZ-ADHD, SCZ-ASD, SCZ-IQ, NEU-IQ, ADHD-IQ, ASD-IQ	BIP-SCZ, SCZ-NEU, SCZ-IQ	SCZ-IQ	SCZ-IQ	FOXP1
Chr10: 106.385-106.83 5MB	SCZ-ADHD, MDD-NEU, MDD-ADHD, NEU-ADHD, ADHD-ASD, ADHD-IQ	BIP-SCZ, SCZ-MDD, SCZ-IQ		SCZ-ADHD	SORCS3

Here we listed out the five hub regions with significant local genetic correlation in at least six trait pairs identified by LOGODetect. The second to fifth columns show the trait pairs with significant genetic correlation at the corresponding locus identified by LOGODetect, ρ -HESS, coloc, and gwas-pw, respectively.

6. L374 "LOGODetect is robust to theta and identifies more short segments as theta increased". Again this is inconsistent: is it robust or is the segment length sensitive to theta?

L375 "Our implemented software allows customized choice of theta based on users' preference on signal region size" This seems inconsistent with your previous recommendation for theta = 0.5, so are you letting users decide based on the preference for region size or are you recommending 0.5? Even in this expanded discussion you produce no real support for this recommendation.

Response:

We appreciate this comment. In our previous submission, we demonstrated that although the number and size of segments identified by LOGODetect may be sensitive to θ (i.e., with a smaller θ , LOGODetect tends to identify fewer genomic regions but the identified regions have larger sizes), the same genetic loci can be robustly identified regardless of θ . This is because as θ increases, the large signal regions tend to be identified as a collection of smaller genomic regions from the same

loci. This is why we initially recommended a theta value of 0.5 (based on our simulations) but still claimed that the user can choose theta based on their preference of region size.

However, we acknowledged that the genetic architecture of the real GWAS datasets may be different from that of our simulated datasets. Therefore, it is more appropriate to consider the “best” θ adaptive to the data in real applications. Here we proposed to use the proportion of genetic covariance of the identified regions as the metric. We varied the value of θ from 0.4 to 0.6, and chose the best θ such that the corresponding identified regions have the largest genetic variance.

We performed simulations to investigate the performance of LOGODetect with adaptive θ . We used dense, simulated genotype data to mimic the genetic architecture of real GWAS in a more authentic way. Based on 503 individuals with European ancestry from the 1000 Genomes Project Phase 3 data, we simulated genotype data for 50,000 individuals with minor allele frequency (MAF) greater than 5% on chromosome 1 using HAPGEN2¹. 336,532 variants remained in the dataset after removing strand ambiguous SNPs. We note that this genotype simulation approach is consistent with the simulation setting in the ρ -HESS paper². We evaluated our method under a heritability enrichment model, where 30% of trait heritability was attributed to 5,000 randomly selected SNPs and 70% of trait heritability was attributed to the remaining SNPs, similar to the simulation settings in our previous submission. We varied the trait heritability from 0.01 to 0.03. Note that if we assume heritability is distributed proportionally in the genome, then the heritability values of 0.01 and 0.03 for chromosome 1 will correspond to the approximate heritability values of 0.12 and 0.36 for the whole genome, respectively, which are on the same scale of SNP heritability for various traits. Under this setting, type I error rate was well-controlled for LOGODetect with adaptive θ (**Table 1**).

Table 1 (This table is also used in response to **Comment 1 of Reviewer #1**). **Type I error rates under a heritability enrichment model.** Type I error rates denote the proportion of simulations that significant segments harboring local genetic correlation are identified in null simulations. In these simulations, 30% of trait heritability was assigned to 5,000 randomly selected SNPs and 70% trait heritability was assigned to the other SNPs. Two cohorts did not have any overlapping samples.

Trait heritability	Type I error rate at significance level $\alpha=0.05$				
	LOGODetect			LOGODetect with adaptive θ	ρ -HESS
	$\theta=0.4$	$\theta=0.5$	$\theta=0.6$		
0.03	0.02	0.01	0	0.02	0
0.02	0.05	0.01	0	0.05	0
0.01	0	0	0.01	0.01	0

We have also updated the results on seven neuropsychiatric traits using adaptively selected θ (**Table 3** and **Figure 3**). For four of the trait pairs, 0.5 was selected as the optimal θ , and for the other 15 pairs, 0.6 was selected.

Table 3 (This table is also used in response to **Comment 3 of Reviewer #1**). **Proportion of genetic covariance identified by LOGODetect with varying θ .** Proportion of genetic covariance identified by LOGODetect with varying θ is calculated via stratified LDSC, and the largest absolute value of proportion of genetic covariance is highlighted in bold and red font. Notably, proportion of genetic covariance identified by LOGODetect with $\theta=0.6$ for BIP-IQ and MDD-IQ are negative because the local genetic covariance has opposite sign compared to global genetic covariance.

Trait 1	Trait 2	$\theta=0.4$	$\theta=0.5$	$\theta=0.6$
BIP	SCZ	7.80%	9.38%	18.76%
BIP	MDD	0%	1.74%	2.47%
BIP	NEU	0%	1.52%	0.77%
BIP	ADHD	0%	0%	0%
BIP	ASD	0%	0%	0%
BIP	IQ	0%	0%	-12%
SCZ	MDD	4.25%	6.12%	6.93%
SCZ	NEU	2.74%	5.19%	9.06%
SCZ	ADHD	3.73%	3.36%	9.27%
SCZ	ASD	0%	0%	3.56%
SCZ	IQ	3.53%	4.72%	8.70%
MDD	NEU	3.35%	6.77%	8.75%
MDD	ADHD	1.66%	2.59%	2.50%
MDD	ASD	0.85%	1.79%	2.07%
MDD	IQ	0%	0%	-2.83%
NEU	ADHD	0%	0%	3.78%
NEU	ASD	0%	0%	1.52%
NEU	IQ	0%	4.68%	8.57%
ADHD	ASD	4.98%	6.16%	3.23%
ADHD	IQ	0.78%	4.23%	8.67%
ASD	IQ	0%	3.55%	1.96%

Figure 3 (This figure is also used in response to **Comment 3 of Reviewer #1**). **LOGODetect identifies genome regions contributing to multiple neuropsychiatric traits.** Heatmap shows the genetic correlation estimates (upper triangle) and the number of segments with local genetic correlation identified by LOGODetect (lower triangle) between seven neuropsychiatric traits; Barplot shows the observed scale heritability estimates and standard errors of seven traits.

Taken together, we have provided a data adaptive approach which bases on the proportion of genetic covariance to select the best θ . We demonstrated that this approach would not induce type I error inflation through additional simulations and updated the results using adaptively selected θ . The discussions above have been incorporated into the revised manuscript (page 7, line 182-186; page 19, line 516-520) and **Supplementary Tables 9** and **13**, and all the related figures and analyses in the main text have been updated accordingly.

7. L379 *"identified genetic regions that may be shared "* inadequate wording - Firstly *"may" is too weak to mean anything, and what does "shared" mean?*

Response:

Thank you for pointing this out. Here, we were describing genetic regions showing concordant associations with multiple traits. We have updated this sentence as "identified genetic regions that show concordant associations across multiple complex traits" in the revised manuscript (page 14, line 360).

Response to Reviewer #4:

The editor requested an evaluation of the authors response to reviewer 3. Below I summarize an opinion in point by point of that response.

R3.1.-Reviewer 3 is absolutely on target by commenting that the genetic correlation is well defined (Falconer and Mackay, 'introduction to quantitative genetics'). The authors' response is inaccurate, while they are arguing that several definitions exist about the genetic correlation.

In short, if for two traits, we could observe the exact amount of mean deviations due to additive genetic effects, we could calculate the additive genetic correlation between these traits. However, additive effects are always unknown and need to be estimated. Researchers can use estimates of SNP markers, partial effects captured by a model. Indefectibly, the study will have missing any not mapped genomic variation in an array. The genomic heritability has demonstrated that as much as 50% of the additive effects can be estimated with today's array density (even less for some traits and assays. Thus, the genetic correlation can be PARTIALLY estimated. Further, the 'missing' aspect of the genetic variation in genotyping assays has been the center of the 'missing heritability' debate. Similarly, the 'misleading' genetic correlation debate of what exactly is being estimated as 'genomic correlation based on a specific genotyping array' (e.g., Genomic heritability: what is it? De los Campos et al., PLoS Genet. 2015).

In sum, the authors are calculating correlations between SNPs (not additive genetic correlations). That should be clear in the paper and well acknowledge, it is untrue that genetic correlation has different but similar definitions. However, there indeed exists an array of methods to estimate approximations to the genomic correlations.

The authors attempt a series of simulations which are not entirely meaningful; all simulations generate 'correlations between SNPs', and then retrieve their simulations. The only solution for them is acknowledging in the discussion or limitations the differences between the correlation of SNPs and the concept of additive genetic correlation.

Response:

We acknowledge that the definition of genetic correlation is clear, which is the correlation of the additive genetic effects on two traits. Traditionally, genetic parameters (i.e. genetic correlation and heritability) are estimated from family and pedigree studies. However, for diseases with low prevalence rates, it is hard to aggregate samples carrying both diseases, which leads to inefficiency in estimation of genetic correlations. An alternative is to estimate genetic parameters from GWAS⁷⁻¹⁰, which has shown a great success. Yet the estimation could be biased to the partial effects of tag SNPs, and the causal effects of untagged SNPs would be absorbed to effect of random error term.

The literature on how to use GWAS data to estimate genetic correlation is rich. We followed ⁷⁻¹⁰ to use a bivariate random effect model and defined the genetic

covariance (correlation) as the covariance (correlation) between SNPs. Under this model, the definition of genetic covariance is consistent with the traditional definition of covariance of additive genetic effects (see Lemma 1 in Supplementary Notes of the LD score regression paper⁹). Therefore, the definition of genetic covariance (correlation) in our study is consistent with that by ρ -HESS. However, the model assumption for two methods are different. We assumed that SNP effects are random variables whereas ρ -HESS assumed SNP effects are fixed.

The discussions above have been incorporated into the revised manuscript (page 5, line 110-112; page 15, line 422-427).

R3.2.- Besides, the simulation is unrealistic and oversimplified. Regions are generated with total independence of one and another, which does not exist in reality. LD blocks are also not all of the same sizes, at all. Finally, common ancestry load / differentiation of origin generates genome-wide similarities (reflected in population structure), unaccounted on simulation as the one proposed.

However, the manuscript also includes a real data analysis. Thus, the simulation has a minor component that could demonstrate an idealistic 'best' possible performance in an uncomplicated scenario.

Response:

We used 15,918 samples from the Wellcome Trust Case Control Consortium (WTCCC) to conduct simulations. We simulated phenotypes using real genetic data, thus there was no assumptions imposed for dependence of genomic regions or size of LD blocks. We did assume the signal regions to be independent from each other. However, this was referring to the causal effect sizes of SNPs, not genotypes. In this way, we simulated genomic regions with correlated contributions to two traits. For SNPs outside these signal regions, effect sizes for two traits were independent. It's true that population structure was not accounted for in simulations. As the reviewer has noted, this could be compensated by the good performance of LOGODetect in real data analysis. We added sentences to discuss the limitation in Discussions (page 15, line 415-418).

R3.3. The authors explain their multiple testing control.

Response: Thank you for the confirmation.

R3.4. The sampling of Z-scores in the null is clear in the present version of the manuscript.

Response: Thank you for the confirmation.

R3.5. The authors have resolved the typo pointed out by R3.

Response: Thank you for the confirmation.

Reference.

1. Su, Z., Marchini, J. & Donnelly, P. HAPGEN2: simulation of multiple disease SNPs. *Bioinformatics* **27**, 2304-2305 (2011).
2. Shi, H., Mancuso, N., Spendlove, S. & Pasaniuc, B. Local genetic correlation gives insights into the shared genetic architecture of complex traits. *The American Journal of Human Genetics* **101**, 737-751 (2017).
3. Wood, A.R. *et al.* Defining the role of common variation in the genomic and biological architecture of adult human height. *Nature genetics* **46**, 1173-1186 (2014).
4. Locke, A.E. *et al.* Genetic studies of body mass index yield new insights for obesity biology. *Nature* **518**, 197-206 (2015).
5. Consortium, G.P. A global reference for human genetic variation. *Nature* **526**, 68-74 (2015).
6. Berisa, T. & Pickrell, J.K. Approximately independent linkage disequilibrium blocks in human populations. *Bioinformatics* **32**, 283 (2016).
7. Yang, J. *et al.* Common SNPs explain a large proportion of the heritability for human height. *Nat Genet* **42**, 565-9 (2010).
8. Lee, S.H., Yang, J., Goddard, M.E., Visscher, P.M. & Wray, N.R. Estimation of pleiotropy between complex diseases using single-nucleotide polymorphism-derived genomic relationships and restricted maximum likelihood. *Bioinformatics* **28**, 2540-2542 (2012).
9. Bulik-Sullivan, B. *et al.* An atlas of genetic correlations across human diseases and traits. *Nature genetics* **47**, 1236 (2015).
10. Bulik-Sullivan, B.K. *et al.* LD Score regression distinguishes confounding from

polygenicity in genome-wide association studies. *Nature genetics* **47**, 291 (2015).

REVIEWER COMMENTS

Reviewer #1 (Remarks to the Author):

I appreciate the additional revisions that the authors made to address the referees' comments.

My main concerns about this revision center around the LD effect parameter θ .

1. Why "maximizing the genetic covariance in all identified regions" could lead to a reasonable estimate of θ ? This is unclear in the main text (Line 184, 516). Related to this, the "variance" in Line 518 seems to be a typo; otherwise it is inconsistent with "covariance" in Line 516.

2. I feel the underlying logic of this criterion is very similar to the authors' replication strategy (Line 262-275): using LOGODetect to identify individual, local genetic correlations, and then using a different method GNOVA for replication based on the "aggregated" genetic covariance of all regions identified by LOGODetect. Since LOGODetect aims to detect local correlations, why is θ chosen to maximize aggregated genetic covariance (Supp Tab 13)?

3. Can the authors explicitly provide the formula of "the genetic covariance in all identified regions" in Line 516 so that readers can understand that the objective is a function of θ and therefore can be potentially used to optimize θ ?

4. According to the previous submissions, LOGODetect results seem quite sensitive to the choice of θ . Here the authors only tested three θ values (0.4,0.5,0.6) to determine the optimal adaptive θ (Supp Tab 9, 13).

5. Since "type I error rate was well-controlled for LOGODetect with adaptive θ ", why did the authors still fix θ as 0.5 throughout simulations (Line 133, 152)?

6. Type I error rates for adaptive θ were not available for simulations reported in Supp Tab 1-8. It is hard to see whether they are "well-controlled" as the authors claimed (Line 186).

7. How does LOGODetect perform in terms of statistical power on simulated data (e.g. Fig. 2) if one uses the adaptive θ approach?

8. Recent studies have suggested various MAF- and LD-dependent genetic architectures of complex traits, as summarized in Table 1 of this paper (<https://doi.org/10.1038/s41588-019-0465-0>) and Supp Note Section 1 of this paper (<https://doi.org/10.1038/s41588-019-0464-1>). The authors' LD modeling based on θ is different from all existing models. How will LOGODetect perform on datasets simulated from other published MAF- and LD-dependent models? (Supp Tab 3 seems to be along this line but the "LD-dependent model" defined there seems oversimplified.)

Collectively (1-8), the treatment of LD and its related parameter θ remains to be a major technical weakness of the present study. However, the authors did not elaborate this key caveat when discussing study limitations (Line 406-431).

Below are comments on other parts.

LOGODetect is built on the scan statistic $Q(R)$, a special topic that many people are not familiar with. The authors did provide some high-level description in Line 89-94, but I don't think it adequately gives readers the motivation of using this statistic and the key intuition. In addition, it is unclear how $Q(R)$ "extends the scan statistic proposed for single trait analysis" (REF 26, 27).

The distinction between LOGODetect and rho-HESS (both weakness and strength of each method)

should be made clear in the main text. This is important for readers to decide which method to use in their own research. Line 408-415 are along this line, but I hope the authors could expand this discussion in a fair way, as they did in the rebuttal letter (page 2).

The term "dense, simulated genotype data" is vague (Line 126, 169), and corresponding details are missing in the main text (Line 597). Because this newly added simulation study corresponds to a set of more realistic scenarios (in terms of per-SNP heritability; rebuttal letter page 2), the authors should provide a concise description in the main text.

Comparison with existing methods (rho-HESS, coloc, gwas-pw) is a key part of this study, and thus the authors should briefly describe how each of these methods was applied to simulated and real datasets in the main text.

Reviewer #2 (Remarks to the Author):

I am happy with the authors' response to my previous set of comments. I have no further comments.

Reviewer #4 (Remarks to the Author):

R 3.1- Thank you to the authors for incorporating the definition, and discussion of additive genetic correlation in the discussion of the paper and highlighting differences in estimation methods depending on the available information. Note that definition is different than estimation methods.

R3.2- I appreciate the clarification, and addition of a discussion of lack of testing in scenarios with population structure. However, I agree with the authors that in a simulation they do not need to replicate every single scenario on a population (e.g. family vs unrelated data, population structure, etc.).

Response to Reviewer #1:

I appreciate the additional revisions that the authors made to address the referees' comments. My main concerns about this revision center around the LD effect parameter θ .

Response:

We really appreciate these thoughtful and constructive comments. During the last round of revision, following reviewer suggestions, we proposed a data-adaptive strategy to select the LD effect parameter θ . In this revision, we have substantially updated all our simulations and real data analyses to incorporate the adaptive strategy to select the best θ in various settings. Throughout these analyses, our results consistently suggest that this strategy could lead to sufficient statistical power and reasonable robustness without inflating the type-I error of identifying local genetic correlations. We believe these new analyses did not contradict any results from our previous submission and have strengthened the manuscript. We provide details of these analyses in the point-by-point response below.

1. Why "maximizing the genetic covariance in all identified regions" could lead to a reasonable estimate of θ ? This is unclear in the main text (Line 184, 516). Related to this, the "variance" in Line 518 seems to be a typo; otherwise it is inconsistent with "covariance" in Line 516.

Response:

The main goal of this study is to identify genomic regions that have correlated genetic effects between two traits. We aim to identify a set of non-overlapping genomic regions R_1, \dots, R_r , so that the global genetic covariance can be solely attributed to the union set which we note as \mathcal{R} ($\mathcal{R} = \cup_{j=1}^r R_j$, explained in detail in Methods, lines 462 and 473-475). Thus, an effective, powerful analytical strategy should ensure that identified genomic regions collectively recapitulate a large proportion of genetic covariance between two traits. This is the idea behind our data-adaptive strategy to select the best θ . We select the θ that maximizes the aggregated genetic covariance of all the identified regions. In this revision, we have conducted additional, extensive simulations to demonstrate that this empirical strategy to estimate θ works well under broad settings and leads to superior performance of LOGODetect compared to other methods, in terms of error control and statistical power.

To fully investigate how LOGODetect performs under realistic genetic architecture, we have updated all the simulations (in this particular response, we show one of the many settings we have performed) to assess type I error and power using genotype data with denser markers and reasonable heritability values. Based on 503 individuals

with European ancestry from the 1000 Genomes Project Phase 3 data, we simulated genotype data for 100,000 individuals with minor allele frequency (MAF) greater than 5% on chromosome 1 using HAPGEN2¹. 336,532 variants remained in the dataset after removing strand-ambiguous SNPs. We assumed a heritability enrichment model, where five genomic segments, each containing 1000 SNPs, were randomly selected as the signal regions shared between two traits, and 30% of trait heritability was assigned to these signal regions. As shown in **Figure 1**, the adaptive θ strategy achieved universally higher statistical power in three measures (i.e., signal points detection rate, signal segments detection rate, and G-score) compared to LOGODetect results with fixed θ values.

Figure 1 (Supplementary Figure 1 in the revised manuscript). Assessment of statistical power under a heritability enrichment model with varying trait heritability. The X-axis shows the heritability for simulated traits on chromosome 1. The Y-axis shows the statistical power assessed by three different measures: (A) signal points detection rate, (B) signal segments detection rate, and (C) G-score. For each trait, we randomly choose $N=5$ segments, each containing $L=1000$ SNPs, as the signal regions. The heritability for the signal regions is set to be 30% of trait heritability. The correlation of genetic effect size of two traits ρ is set to be 0.9. Each simulation setting was repeated 100 times.

Importantly, the adaptive θ approach does not inflate the type-I error of LOGODetect (see our response to Comments #5-6). The discussions and intuitions about the adaptive θ approach have been incorporated into the revised manuscript (page 4, lines 95-104; page 6, lines 153-158) and **Supplementary Figure 1**. We have also corrected the typo.

2. I feel the underlying logic of this criterion is very similar to the authors' replication strategy (Line 262-275): using LOGODetect to identify individual, local genetic correlations, and then using a different method GNOVA for replication based on the "aggregated" genetic covariance of all regions identified by LOGODetect. Since LOGODetect aims to detect local correlations, why is theta chosen to maximize aggregated genetic covariance (Supp Tab 13)?

Response:

Thank you for the comments. We agree the underlying logic to choose θ and to

replicate the results are deeply connected. The aggregated genetic covariance of a set of genomic segments measures how much the segments collectively explains the genetic covariance between two traits. It is a metric for the union of the identified segments. When the sample size is not sufficient for segment-wise replication, we relegate to evaluate the replication of the union of the identified segments (**Table 1**). Similarly, for choosing θ , the empirical strategy we proposed measures and optimizes the performance of the union of identified segments.

Table 1 (Table 1 in the revised manuscript). **Stratified genetic covariance analysis on UKBB replication cohorts**

	Genetic Cov*	s.e.	p-value	Proportion of genetic cov	Proportion of SNPs	Fold enrichment
LOGODetect	2.18e-4	6.65e-5	1.04e-3	11.50%	1.15%	10.02
ρ -HESS	6.62e-4	3.16e-4	3.61e-2	30.00%	20.12%	1.49
coloc	-5.70e-5	2.30e-5	1.33e-2	-2.85%**	0.34%	-8.36**
gwas-pw	3.84e-5	6.54e-5	5.57e-1	1.92%	1.61%	1.20

* Genetic Cov represents estimated genetic covariance of the identified regions using GNOVA.

** Genetic covariance of regions identified by coloc has opposite sign compared to global genetic covariance, therefore the corresponding proportion of genetic covariance and fold enrichment are negative.

LOGODetect scans the genome to identify local segments harboring genetic correlation. In the scan statistic $Q(R) = \frac{\sum_{i \in R} z_{1i} z_{2i}}{(\sum_{i \in R} l_i)^\theta}$, the denominator is a penalty term for segment length and LD, with θ controlling the strength of penalty. In choosing θ , the method is actually choosing the penalty level. In the proposed strategy, the selection of θ is data adaptive. The identified segments enjoy the focal property at appropriate penalty levels, such that the levels also allow optimal explanation of genetic covariance.

3. Can the authors explicitly provide the formula of "the genetic covariance in all identified regions" in Line 516 so that readers can understand that the objective is a function of theta and therefore can be potentially used to optimize theta?

Response:

Thank you for the suggestion. Denote the regions detected by LOGODetect under parameter θ as $\hat{R}_1^\theta, \dots, \hat{R}_m^\theta$. We denote their union as $\hat{\mathcal{R}}^\theta$ and denote the genetic covariance in $\hat{\mathcal{R}}^\theta$ as $\rho(\hat{\mathcal{R}}^\theta)$. In theory, $\rho(\hat{\mathcal{R}}^\theta) = \sum_{i=1}^m |\hat{R}_i^\theta \cap \mathcal{R}| * \frac{\rho_g}{K}$, where \mathcal{R} is union set of true signal regions, ρ_g is the global genetic covariance, and $K = |\mathcal{R}|$ is the number of SNPs in \mathcal{R} . In practice, the true signal regions \mathcal{R} is unknown. $\rho(\hat{\mathcal{R}}^\theta)$ can

be estimated using the stratified LDSC². Let $\pi(\theta) = \frac{\rho(\hat{\mathcal{R}}^\theta)}{\rho_g}$ be the proportion of genetic covariance explained by $\hat{\mathcal{R}}^\theta$ to the global genetic covariance. We assume that $\rho(\hat{\mathcal{R}}^\theta) = 0$ and $\pi(\theta) = 0$ if $\hat{\mathcal{R}}^\theta = \emptyset$. We calculate $\pi(\theta)$ for a candidate set of θ values, and then we determine θ adaptive to data via the following optimization problem:

$$\hat{\theta} = \arg \max_{\theta} |\pi(\theta)|.$$

These details have been added into the revised manuscript (page 19, lines 531-544).

4. According to the previous submissions, LOGODetect results seem quite sensitive to the choice of theta. Here the authors only tested three theta values (0.4,0.5,0.6) to determine the optimal adaptive theta (Supp Tab 9, 13).

Response:

Thank you for this comment. We have expanded the set of candidate θ values to $\{0.4, 0.45, 0.5, 0.55, 0.6, 0.65, 0.7\}$. We updated our results for simulations and real data applications to seven neuropsychiatric traits (**Tables 2-3**).

Table 2 (Supplementary Table 1 in the revised manuscript, **Supplementary Table 9** in the previous submission). **Type I error rates of LOGODetect with adaptive θ under an infinitesimal model.** Type I error rates denote the proportion of simulations that significant segments harboring local genetic correlation are identified in null simulations. In these simulations, per-SNP heritability for all the SNPs are the same. There are no overlapping samples in the two cohorts. Simulations under different parameters scenarios are repeated for 100 times.

Trait heritability	Type I error rates
0.05	0.04
0.04	0.02
0.03	0.06
0.02	0.04
0.01	0.02

Table 3 (Supplementary Table 13 in the revised manuscript). Proportion of genetic covariance identified by LOGODetect with varying θ .

Trait 1	Trait 2	$\theta=0.4$	$\theta=0.45$	$\theta=0.5$	$\theta=0.55$	$\theta=0.6$	$\theta=0.65$	$\theta=0.7$
BIP	SCZ	7.80%	8.80%	9.38%	11.12%	11.63%	10.59%	10.12%
BIP	MDD	0%	0%	1.74%	1.74%	2.47%	1.43%	1.43%
BIP	NEU	0%	0%	1.52%	0.57%	0.77%	2.47%	2.49%
BIP	ADHD	0%	0%	0%	0%	0%	0%	0%
BIP	ASD	0%	0%	0%	0%	0%	0%	0%
BIP	IQ	0%	0%	0%	-2.17%	-12%	0.82%	-5.44%
SCZ	MDD	4.25%	4.52%	6.12%	6.22%	6.93%	7.45%	6.38%
SCZ	NEU	2.74%	3.23%	5.19%	6.44%	9.06%	14.13%	13.89%
SCZ	ADHD	3.73%	3.41%	3.36%	3.36%	9.27%	10.71%	7.37%
SCZ	ASD	0%	0%	0%	0%	3.56%	3.57%	3.94%
SCZ	IQ	3.53%	4.31%	4.72%	4.38%	8.70%	7.08%	9.38%
MDD	NEU	3.35%	5.56%	6.77%	6.77%	8.75%	7.99%	8.33%
MDD	ADHD	1.66%	1.64%	2.59%	2.55%	2.50%	2.13%	1.29%
MDD	ASD	0.85%	1.08%	1.79%	2.89%	2.07%	1.53%	1.19%
MDD	IQ	0%	0%	0%	-1.39%	-2.83%	-2.81%	-2.81%
NEU	ADHD	0%	0%	0%	0%	3.78%	4.37%	5.35%
NEU	ASD	0%	0%	0%	-0.12%	1.52%	0%	1.06%
NEU	IQ	0%	0%	4.68%	5.62%	8.57%	9.12%	10.35%
ADHD	ASD	4.98%	5.64%	6.16%	7.60%	3.23%	2.60%	0.87%
ADHD	IQ	0.78%	1.29%	4.23%	6.43%	8.67%	8.98%	8.78%
ASD	IQ	0%	3.57%	3.55%	1.15%	1.96%	3.08%	2.68%

We note that although different θ values will lead to changes in the length and the total counts of identified genomic regions, LOGODetect results based on different θ were fairly robust at the locus level. As we change the choice of θ , some larger genomic regions may be re-identified as smaller segments. However, since the same locus is identified in the analysis, these discrepancies in region lengths and counts will not lead to issues when interpreting results. For example, although the set of candidate θ values was expanded in our analysis, we identified the same five loci with significant local genetic correlations in seven or more trait pair analyses as in the previous submission, which again shows the robustness of LOGODetect.

Still, we understand the reviewer's concern and we believe the proposed adaptive strategy to select θ will help the users understand our method and reduce confusion. Since extensively searching for θ will increase the computation time, we believe the set of $\{0.4, 0.45, 0.5, 0.55, 0.6, 0.65, 0.7\}$ would be sufficient in practice. The discussions above have been incorporated into the revised manuscript (page 19, lines 531-535) and **Supplementary Tables 1 and 13**.

5. Since "type I error rate was well-controlled for LOGODetect with adaptive theta", why did the authors still fix theta as 0.5 throughout simulations (Line 133, 152)?

Response:

Thank you for the comment. In this revision, we have thoroughly evaluated the type I error rate of LOGODetect when θ is adaptively selected. The type I error rates are

well controlled, and the data can be found in **Supplementary Tables 1-8**. We also provided more detailed information in response to Comments #6-7.

6. Type I error rates for adaptive theta were not available for simulations reported in Supp Tab 1-8. It is hard to see whether they are "well-controlled" as the authors claimed (Line 186).

Response:

Following Comment #5, we have updated all the simulation results using the adaptive θ approach. We first assessed the type I errors of LOGODetect with adaptive θ under an infinitesimal model in which single-trait genetic effects were assumed to be the same for all SNPs. Our method was evaluated across a range of heritability combinations for two traits. Note that a heritability value of 0.01 or 0.05 on chromosome 1 will approximately correspond to heritability values of 0.12 or 0.60 in the whole genome, which are realistic values for typical GWAS traits. The family-wise type I error rate of our method was well-calibrated in all simulation settings with varying heritability values (**Table 2**). We obtained consistent results in additional simulation settings, for detailed information please see **Supplementary Tables 2-8** in the revised manuscript.

Table 2 (This table is also used in response to Comment #4). **Type I error rates of LOGODetect with adaptive θ under an infinitesimal model.** Type I error rates denote the proportion of simulations that significant segments harboring local genetic correlation are identified in null simulations. In these simulations, per-SNP heritability for all the SNPs are the same. There are no overlapping samples in the two cohorts. Simulations under different parameters scenarios are repeated for 100 times.

Trait heritability	Type I error rates
0.05	0.04
0.04	0.02
0.03	0.06
0.02	0.04
0.01	0.02

7. How does LOGODetect perform in terms of statistical power on simulated data (e.g. Fig. 2) if one uses the adaptive theta approach?

Response:

As we stated above, we have updated all simulations based on the adaptive θ approach in the revised manuscript. Specifically, simulations in Fig 2 were performed under a heritability enrichment model. Five segments, each containing 1000 SNPs, were randomly selected as the signal regions shared between two traits, and 30% of

trait heritability was assigned to the signal regions. LOGODetect was the best or second best in signal points detection rates compared to the other methods across various heritability settings (**Figure 2A**). Moreover, LOGODetect showed universally higher signal segments detection rates and G-scores compared to the other methods (**Figure 2B-C**). We reemphasize that G-score is a balanced measure of sensitivity and specificity, while signal point detection rate and signal segment detection rate measures sensitivity alone. We obtained consistent results as the heritability enrichment varies (**Figure 2D-F**) and in additional simulation settings. The complete results are reported in **Supplementary Figures 2-10** in the revised manuscript.

Figure 2 (Figure 2 in the revised manuscript). Assessment of statistical power under a heritability enrichment model with varying trait heritability and heritability enrichment. The Y-axis shows the statistical power assessed by three different metrics: (A, D) signal points detection rate measures sensitivity at the SNP level, (B, E) signal segments detection rate measures sensitivity at the segment level, and (C, F) G-score jointly measures specificity and sensitivity. The heritability represents the trait heritability of chromosome 1 and the proportion of heritability represents the proportion of the trait heritability explained by the signal regions. Significance cutoffs for gwas-pw are adjusted so that the empirical type I error rate is controlled at 0.05.

8. Recent studies have suggested various MAF- and LD-dependent genetic architectures of complex traits, as summarized in Table 1 of this paper (<https://doi.org/10.1038/s41588-019-0465-0>) and Supp Note Section 1 of this paper (<https://doi.org/10.1038/s41588-019-0464-1>). The authors' LD modeling based on theta is different from all existing models. How will LOGODetect

perform on datasets simulated from other published MAF- and LD-dependent models? (Supp Tab 3 seems to be along this line but the "LD-dependent model" defined there seems oversimplified.)

Response:

As suggested, we tested the LDAK model³ with MAF-dependent and LD-dependent effect sizes. We drew the genetic effect of the j -th SNP from a normal distribution $N(0, h_j^2)$, where $h_j^2 \propto [f_j * (1 - f_j)]^{1+\alpha} * w_j$, f_j is the MAF, α is set to be -0.25 as suggested by the LDAK paper³, and w_j is the weight computed by the LDAK software³. Note that the LD-dependent model used in our previous submission was a special case of this LDAK model with $\alpha = -1$ and $w_j = \frac{1}{l_j}$, where l_j is the LD score.

We first assessed the type I errors of our method under the LDAK model. The family-wise type I error rates of our method were well-controlled in all simulation settings with varying heritability values (**Table 4**). We also compared the statistical power of different methods under this model. Colocalization methods (coloc and gwas-pw) achieved better signal point detection rates, however, LOGODetect was comparable or superior to the other methods in terms of signal segments detection rates and G-scores (**Figure 3**).

Table 4 (Supplementary Table 3 in the revised manuscript). Type I error rates of LOGODetect with adaptive θ under the LDAK model. Type I error rates denote the proportion of simulations that significant segments harboring local genetic correlation are identified in null simulations. In these simulations, heritability for the j -th SNP is proportional to $[f_j * (1 - f_j)]^{0.75} * w_j$, where f_j is the MAF and w_j is the weight computed by the LDAK software³. Two cohorts did not have any overlapping samples. Simulations under different parameters scenarios are repeated 100 times.

Trait heritability	Type I error rates
0.05	0.03
0.04	0.01
0.03	0.02
0.02	0.01
0.01	0.

Figure 3 (Supplementary Figure 8 in the revised manuscript). Assessment of statistical power under a LDAK model. The Y-axis shows the statistical power assessed by three different metrics: (A) signal points detection rate measures sensitivity at the SNP level, (B) signal segments detection rate measures sensitivity at the segment level, and (C) G-score jointly measures specificity and sensitivity. Significance cutoffs for gwas-pw are adjusted so that the empirical type I error rate is controlled at 0.05.

Taken together, these additional simulations demonstrate that LOGODetect can obtain robust results under mis-specified models with MAF- and LD-dependent genetic effects. The discussions above have been incorporated into the revised manuscript (page 5, lines 128-129, and page 7, lines 174-175), **Supplementary Table 3**, and **Supplementary Figure 8**.

Collectively (1-8), the treatment of LD and its related parameter theta remains to be a major technical weakness of the present study. However, the authors did not elaborate this key caveat when discussing study limitations (Line 406-431).

Response:

Thank you for these constructive comments and suggestions. We agree that the treatment of LD and parameter θ is crucial in practice, which is why we have added additional simulations and updated all previous simulations and real data analyses to evaluate the type-I error rate, statistical power, and robustness of our approach. Specifically, we have made several changes to address the problem. First, we have added explanations and discussions about the rationale underlying the adaptive strategy to select θ (lines 101-104). Second, we have re-implemented the simulations and real data analysis using the adaptive θ approach. We have also appended the simulations to the LDAK model with LD- and MAF-dependent effect sizes. In all simulations, LOGODetect showed well-controlled type I error. Our method achieved superior statistical power compared to other methods. These results under various settings are consistent with the conclusions we made in the previous submission, demonstrating the robustness and effectiveness of the adaptive θ approach. We hope these additional analyses and discussions could improve our manuscript and address your concerns. Finally, in the Discussion section, we have added texts to state that the treatment of LD and the proposed approach to select parameter θ is empirical (Page 15, lines 422-427).

Below are comments on other parts.

1. LOGODetect is built on the scan statistic $Q(R)$, a special topic that many people are not familiar with. The authors did provide some high-level description in Line 89-94, but I don't think it adequately gives readers the motivation of using this statistic and the key intuition. In addition, it is unclear how $Q(R)$ "extends the scan statistic proposed for single trait analysis" (REF 26, 27).

Response:

We appreciate this comment. We have proposed the scan statistic as follows, to search for regions harboring local genetic correlation:

$$Q(R) = \frac{\sum_{i \in R} z_{1i} z_{2i}}{(\sum_{i \in R} l_i)^\theta}.$$

As mentioned in the manuscript, this is a generalization of the scan statistic proposed in Jeng et al⁴ and Li et al⁵, which detects signal regions associated with a single trait. The scan procedure in Jeng et al was based on the mean of the marginal test statistics in a candidate region. We refer to this as the mean scan procedure. The scan statistic for a given region R is defined as $\frac{\sum_{i \in R} z_i}{\sqrt{|R|}}$, where $|R|$ denotes number of SNPs in region R . The authors showed that the mean scan procedure is asymptotically optimal as long as SNPs are independent and the SNPs in the signal region have the same effect size. In our paper, we extend the mean scan framework to the context of detecting local genetic correlation. We replace $|R| = \sum_{i \in R} 1$ in the mean scan procedure, with $\sum_{i \in R} l_i$, to account for LD. Notably, under the case that all the SNPs have the same LD effect, e.g. l_i are the same for all SNPs, $\sum_{i \in R} l_i$ just reduces to $|R|$ up to a multiple constant. Following Li et al⁵, we relax the choice of power θ from 0.5 to a numeric in $[0,1]$ to ensure that the scan statistic is comparable for different region sizes.

Second, the scan statistics we proposed has good genetics interpretation. It has been shown that under the polygenic model that per-SNP genetic covariance is the same for all SNPs, and if two GWASs are independent, then the following equation holds²:

$$E[z_{1i} z_{2i}] = \frac{\sqrt{N_1 N_2} \rho_g}{M} l_i,$$

where the N_1, N_2 represents the sample size for two GWASs, M is the SNP count and ρ_g is the (global) genetic covariance. If we let θ be 1 and R be the whole genome, then the expectation of the scan statistic $Q(R)$ reduces to $\frac{\sqrt{N_1 N_2} \rho_g}{M}$, which is exactly the (global) genetic covariance ρ_g multiplied by a constant. This implicates that when the candidate region is the whole genome, the scan statistic actually represents the strength of (global) genetic covariance. In our framework, we do not assume that per-SNP genetic covariance is the same for all SNPs across genome, but assume that genetic covariance is localized in some small genome regions. Therefore, we use the scan statistic in a local region, as a metric to detect significant

local genetic sharing.

The discussions above have been incorporated into the revised manuscript (page 4, lines 95-98) and **Supplementary Notes 1**.

2. The distinction between LOGODetect and rho-HESS (both weakness and strength of each method) should be made clear in the main text. This is important for readers to decide which method to use in their own research. Line 408-415 are along this line, but I hope the authors could expand this discussion in a fair way, as they did in the rebuttal letter (page 2).

Response:

Thank you for the comments. We note that the deflation of type I error observed for ρ -HESS is not contradictory to results published in ρ -HESS paper⁶. ρ -HESS was formulated as an estimation problem instead of the hypothesis testing problem in our manuscript. In their paper, they have shown simulation results to demonstrate that the local genetic correlation can be accurately estimated when the true parameter is 0. However, the evidence shown in the ρ -HESS paper could not rule out deflation when the method is used for inference. The discussions above have been incorporated into the revised manuscript (page 15, lines 411-416).

3. The term "dense, simulated genotype data" is vague (Line 126, 169), and corresponding details are missing in the main text (Line 597). Because this newly added simulation study corresponds to a set of more realistic scenarios (in terms of per-SNP heritability; rebuttal letter page 2), the authors should provide a concise description in the main text.

Response:

Thank you for this comment. Here are the technical details of the simulation scenario. Based on 503 individuals with European ancestry from the 1000 Genomes Project Phase 3 data, we simulated genotype data for 100,000 individuals with minor allele frequency (MAF) greater than 5% on chromosome 1 using HAPGEN2¹. 336,532 variants remained in the dataset after removing strand ambiguous SNPs. All the simulations have been updated using these simulated dense genotype data. The discussions above have been incorporated into the revised manuscript (page 5, lines 121-123, and page 22, lines 638-642) and **Supplementary Notes 2**.

4. Comparison with existing methods (*rho*-HESS, coloc, gwas-pw) is a key part of this study, and thus the authors should briefly describe how each of these methods was applied to simulated and real datasets in the main text.

Response:

Thank you for this suggestion. We used ldetect⁷ to pre-specify 1,703 approximately LD-independent blocks (spanning 1.6 Mb on average) as candidate genomic regions, as suggested by ρ -HESS and gwas-pw. We also used these LD-independent blocks as candidate genomic regions for coloc. In simulation studies, we used 133 approximately LD-independent regions in chromosome 1 as the pre-specified genomic regions for ρ -HESS, coloc, and gwas-pw. For ρ -HESS, the 1000 Genomes Project Phase 3 data⁸ was used as the reference panel, and the number of eigenvectors used in the truncated-SVD for LD matrix inversion is determined as 50 by default, and the minimum eigenvalue cut off in truncated-SVD is determined as 1.0 by default, as suggested by the ρ -HESS software (<https://huwenboshi.github.io/hess/>). ρ -HESS reported the estimate and significance of local genetic correlation for each candidate genomic region, and we applied Benjamini-Hochberg procedure⁹ to control FDR with a cutoff of 0.05, accounting for the multiple testing problem concerning all genomic regions. Coloc (<https://CRAN.R-project.org/package=coloc>) and gwas-pw (<https://github.com/joepickrell/gwas-pw>) estimated the posterior probability that two traits shared at least one causal SNP for each genomic region, and those genomic regions with posterior probability above 0.95 are determined as identified regions. The discussions above have been incorporated into the revised manuscript (page 24, lines 689-704).

Reviewer #2 (Remarks to the Author):

I am happy with the authors' response to my previous set of comments. I have no further comments.

Response: We appreciate all the comments and suggestions. We are glad that you found our revisions sufficient for publication.

Reviewer #4 (Remarks to the Author):

R 3.1- Thank you to the authors for incorporating the definition, and discussion of additive genetic correlation in the discussion of the paper and highlighting differences in estimation methods depending on the available information. Note that definition is different than estimation methods.

R3.2- I appreciate the clarification, and addition of a discussion of lack of testing in scenarios with population structure. However, I agree with the authors that in a simulation they do not need to replicate every single scenario on a population (e.g. family vs unrelated data, population structure, etc.).

Response: Thank you for your positive and constructive comments which have helped improve our manuscript.

Reference

1. Su, Z., Marchini, J. & Donnelly, P. HAPGEN2: simulation of multiple disease SNPs. *Bioinformatics* **27**, 2304-2305 (2011).
2. Bulik-Sullivan, B. *et al.* An atlas of genetic correlations across human diseases and traits. *Nature genetics* **47**, 1236 (2015).
3. Speed, D., Cai, N., Johnson, M.R., Nejentsev, S. & Balding, D.J. Reevaluation of SNP heritability in complex human traits. *Nature genetics* **49**, 986-992 (2017).
4. Jeng, X.J., Cai, T.T. & Li, H. Optimal sparse segment identification with application in copy number variation analysis. *Journal of the American Statistical Association* **105**, 1156-1166 (2010).
5. Li, Z., Liu, Y. & Lin, X. Simultaneous Detection of Signal Regions Using Quadratic Scan Statistics With Applications in Whole Genome Association Studies. *arXiv preprint arXiv:1710.05021* (2017).
6. Shi, H., Mancuso, N., Spendlove, S. & Pasaniuc, B. Local genetic correlation gives insights into the shared genetic architecture of complex traits. *The American Journal of Human Genetics* **101**, 737-751 (2017).
7. Berisa, T. & Pickrell, J.K. Approximately independent linkage disequilibrium blocks in human populations. *Bioinformatics* **32**, 283 (2016).
8. Consortium, G.P. A global reference for human genetic variation. *Nature* **526**, 68-74 (2015).
9. Benjamini, Y. & Hochberg, Y. Controlling the false discovery rate: a practical and powerful approach to multiple testing. *Journal of the Royal statistical society: series B*

(Methodological) **57**, 289-300 (1995).

REVIEWERS' COMMENTS

Reviewer #1 (Remarks to the Author):

I appreciate the additional revisions that the authors made to address the referees' comments. I think the authors have provided sufficient information for readers to interpret and assess the LD modelling strategy in the proposed method. I do not have any major comments on this key issue.

It seems that LDSC plays an important role in the implementation and application of the proposed method (e.g., Line 547, Page 20; Line 625, Page 24). Hence, I wonder if the authors could concisely describe how LDSC was used in the main text. I am confident that the authors can address this minor comment with the assistance of the editorial team, and thus there is no need for me to read the revised text again.

Response to Reviewer #1:

I appreciate the additional revisions that the authors made to address the referees' comments. I think the authors have provided sufficient information for readers to interpret and assess the LD modelling strategy in the proposed method. I do not have any major comments on this key issue.

It seems that LDSC plays an important role in the implementation and application of the proposed method (e.g., Line 547, Page 20; Line 625, Page 24). Hence, I wonder if the authors could concisely describe how LDSC was used in the main text. I am confident that the authors can address this minor comment with the assistance of the editorial team, and thus there is no need for me to read the revised text again.

Response:

We appreciate all the comments and suggestions. We are glad that you found our revisions sufficient for publication. In the following, we described how LDSC was used.

We used LDSC (<https://github.com/bulik/ldsc>) to estimate heritability in each chromosome. Stratified-LDSC was used to estimate genetic covariance of the identified regions. In detail, we manually created two annotations: the identified regions and the remaining genome, then we ran the standard LDSC software to calculate the genetic covariance and the proportion of genetic covariance of each annotation. For both LDSC and stratified-LDSC, LD scores were computed with the standard LDSC software from 503 individuals with European ancestry from the 1000 Genomes Project Phase 3 data. Both methods were applied with an unconstrained intercept, using all SNPs as observations in the dependent variable and LD scores as regression weights.

These details have been added into the revised manuscript (page 20, lines 654-661).